Subject Areas:
cybernetics/systems theory/complexity

Keywords:
Campbell's Law, Goodhart's Law, agent-based model, proxy, multitasking, replicability crisis, climate inaction

Author for correspondence:
Oliver Braganza
e-mail: oliver.braganza@ukbonn.de

# Proxyeconomics, a theory and model of proxy-based competition and cultural evolution

Oliver Braganza[1,2]

[1]Institute for Experimental Epileptology and Cognition Research, and [2]Center for Science and Thought, University of Bonn, Bonn, Germany

OB, 0000-0001-8508-1070

Competitive societal systems by necessity rely on imperfect proxy measures. For instance, profit is used to measure value to consumers, patient volumes to measure hospital performance, or the journal impact factor to measure scientific value. While there are numerous reasons why proxies will deviate from the underlying societal goals, they will nevertheless determine the selection of cultural practices and guide individual decisions. These considerations suggest that the study of proxy-based competition requires the integration of cultural evolution theory and economics or decision theory. Here, we attempt such an integration in two ways. First, we describe an agent-based simulation model, combining methods and insights from these disciplines. The model suggests that an individual intrinsic incentive can constrain a cultural evolutionary pressure, which would otherwise enforce fully proxy-oriented practices. The emergent outcome is distinct from that with either the isolated economic or evolutionary mechanism. It reflects what we term *lock-in*, where competitive pressure can undermine the ability of agents to pursue the shared social goal. Second, we elaborate the broader context, outlining the system-theoretic foundations as well as some philosophical and practical implications, towards a broader theory. Overall, we suggest such a theory may offer an explanatory and predictive framework for diverse subjects, ranging from scientific replicability to climate inaction, and outlining strategies for diagnosis and mitigation.

## 1. Introduction

### 1.1. Competitive societal systems

Modern information-driven societies increasingly rely on competitive systems and the proxy measures they require. However, there are

**Table 1.** Illustrative examples of goals, proxies and corruption claims in proxy-based competititive systems.

|  | science | medicine | education | politics | markets |
|---|---|---|---|---|---|
| (purported) societal goal | true and relevant research | patient health | knowledge, skills | voter representation | subjective and objective well-being |
| proxy measure | publication count, impact factor | patient numbers, profit | standardized test scores | publicity, votes | income, profit, GDP |
| corruption claims | replicability crisis [1–6] | bad patient care, overtreatment [7,8] | teaching to the test [9,10] | populism, lobbycracy [11,12] | financial crises, global warming [13–18] |

For any competitive societal system to achieve an abstract goal, it must rely on proxy measures. However, any proxy measure becomes a target for the competing individuals (or groups). Campbell's and Goodhart's Laws state that this will promote corruption of the measures [19–21]. Note that the purported 'societal goal' can also be seen as a mere public justification, a position discussed in §§3.1 and 4.4.

prominent arguments and compelling evidence suggesting that the use of proxy measures can corrupt such systems (table 1 [1–18]).

For instance, in academia a substantial fraction of scientific research appears not to be replicable, which, alongside subjective researcher assessments, suggests an excessive proxy orientation [1,4]. Similarly, there are arguments that rich economies as a whole have become excessively oriented towards the proxy measures used in market competition, such as profit [14,15,22–25]. Such arguments question if market proxies necessarily serve their purported social goal, namely welfare (e.g. [18], also see §4.6). Excessively proxy-oriented systems can maintain stability despite widespread acknowledgement of the arising problems [4,15,24,26], a type of systematic inertia we term *lock-in*. For instance, despite widespread dissatisfaction with standardized tests to measure educational outcomes, or the journal impact factor to measure research value, these proxy systems continue in force. The persistent inaction to combat anthropogenic climate change, despite overwhelming evidence of the likely catastrophic long-term consequences, can be seen as another example [16,27].

While such arguments are not new, they remain both highly relevant and highly controversial. Here, we argue that they must be taken seriously, because proxy-induced corruption, or *lock-in* states, arise systematically due to the very use of proxy measures to mediate competition. The reason is best known as Goodhart's Law: *When a measure becomes a target, it ceases to be a good measure* [20,21]. To be clear, the present argument is *not* that proxy measures should, or can, be avoided. Indeed, it is precisely because they can frequently not be avoided, that the arising dangers merit special attention. However, the study of proxy-based competition, and the scientific appraisal of issues such as those raised in table 1, has been hampered by traditional disciplinary boundaries.

In particular, we argue that an understanding of proxy-based competition requires synthesizing economic approaches with insights and methods from cultural evolutionary theory [28–30]. This requires confronting incongruent methodological assumptions between the two disciplines [31]. In economic models, outcomes are typically determined fully by individual decisions, where the decision makers are typically highly rational, knowledgeable and may pursue complex goals. In particular, economists have used so-called 'multitasking' models to investigate the combined impact of intrinsic (e.g. moral) and extrinsic (e.g. competitive) goals on individual decision making (e.g. [32,33]). By contrast, cultural evolutionary models focus on the cultural determinants of individual action and social outcomes, which are thought to in large part supersede individual knowledge and agency (e.g. [1]). Such cultural evolutionary approaches typically exclude the very notion of purpose or a shared social goal.

Here, we develop an agent-based model which integrates these incompatible traditions by combining (A) an economic 'multitasking' model with (B) a mechanism of cultural evolution. While the economic decision model allows for individual agency and an individual motivation towards a shared social goal, the cultural evolution mechanism plays out proxy-based selection at the system level. The resulting model describes a potential mechanism underlying system *lock-in*, and allows to explore its determinants and implications. Crucially, both the emergent dynamics and equilibrium outcomes are

distinct from those resulting from either the economic or evolutionary mechanism alone. In addition to the model, we develop the conceptual foundations and potential implications of a prospective broader theory of proxy-based competition, tentatively labelled *proxyeconomics*.

The remaining paper will be structured as followed:

§1 **Introduction.** First, we will summarize the central informational arguments for an empirical and theoretical focus on proxy measures (§1.2). We will then introduce the general background of the present work in cultural evolution theory (§1.3) and the mostly economic literature on Goodhart's Law and incentive misalignment (§1.4).

§2 **Model.** Next, we present the agent-based model, including

  §2.1 rationale,
  §2.2 technical description,
  §2.3 results,
  §2.4 discussion and
  §2.5 conclusion.

§3 **General discussion.** Next, we will further develop the fundamental information theoretic concepts underlying the present theory referring to systems theory (§3.1), optimization theory (§3.2) and cultural evolution theory (§3.3).

§4 **Future perspectives.** Then we will present future perspectives and potential implications across a surprising range of additional qualitative and quantitative disciplines (§4.1–4.7), outlining open questions requiring further research and illustrating the full potential scope of the present theory.

§5 **Conclusion.**

Given its transdisciplinary nature, the present paper presents particular challenges to a presentation which is both concise, yet legible to scholars with varying backgrounds. We have attempted to combine a mostly modular structure with cross-references, such that readers may be able to skip to the sections most relevant to them. Readers most interested in the agent-based model may skip directly to §2, referring to specific subsections of the general introduction (§1) or general discussion (§3) only as needed. Readers more interested in the broader theory and its potential social, political or ethical implications may want to focus on §§1, 3 and 4.

## 1.2. Why *proxy*economics?

The term *proxyeconomics* is meant to suggest *the proxy* as a useful focus point in the theoretical and empirical investigation of proxy-based competition. It is based on two premises

  (i) Competitive societal systems are designed to serve, and justified by, an abstract purpose, termed *the goal* (table 1).
  (ii) The practical mechanisms mediating actual competition can only approximately capture this goal, and their combined effect can be summarized in a scalar metric termed *the proxy*.

To clarify what we mean by this, we will now first define the term proxy, as used here. We will then clarify what we mean by societal goal and elaborate why proxy and goal will generally differ.

### 1.2.1. The proxy

We can most easily think of familiar proxy measures, such as the journal impact factor, test scores, or quarterly profits (table 1). Such proxy measures are intended to capture some underlying goal (e.g. scientific quality or consumer value). To do so, they condense complex information into simple *scalar metrics* (i.e. magnitudes). For instance, the journal impact factor captures complex information through the editorial and peer-review process. The fundamental reason for the use of proxies is that decisions require ranking options on some single *scale*. Accordingly, the creation of proxy measures is common in any system requiring ranking. Such proxies often have a considerable impact on factual competitive outcomes, and they are convenient means to build intuition about proxy-based competition. In particular, the inspection of how exactly such proxies are created reveals the potential imperfections in their approximation of the underlying goal.

Here, we extend this intuition, defining 'the proxy' as a *summary metric of competitiveness* within a given societal system. This summary metric may incorporate multiple familiar proxy measures improving its information content, but it will ultimately remain imperfect: the proxy reflects the best attempt to

capture a given abstract goal with practically feasible societal mechanisms. To the degree that a competitive system produces consistent rankings, the proxy can be described by a scalar metric.[1]

When societal competition entails some form of *selection* (of e.g. companies or academics or their practices), this conceptualization of *the proxy* allows us to explicitly equate it with the concept of *fitness* in cultural evolution theory [1,29,37,38]. Indeed, as we will argue below, the competitive societal system can be seen as an attempt to translate an abstract societal goal into a cultural selection mechanism, or 'constructed cultural niche' (§1.3).

### 1.2.2. The goal

Most competitive societal systems are designed to serve, or justified by, some abstract societal goal (table 1). For instance, the purpose of academia may be conceived of as producing true and relevant research [1]. Similarly, markets are generally justified as means to serve the goal of maximizing the subjective and objective well-being of its participants [18]. While the precise definition of such societal goals is often extremely challenging [31,39], we here simply posit that there is *some* arbitrary goal. Arguably, such a goal is implied by the very fact that society maintains an artificial competitive system in the first place (see also §§4.4 and 4.5). We also note that avoiding to theorize a societal goal, even because of valid epistemic concerns (§4.1), may itself be a practice conducive to corruption (§4.2). The reason is that in order to even conceptualize Goodhart's Law, we must first posit a goal. Avoiding to theorize a goal, however difficult it may be, blinds us to the potential dangers of Goodhart's Law by default. The societal goal is thus defined as an arbitrary societal agreement on what a competitive system should achieve.

In practice, the goal may be partially defined in relation to the proxy, i.e. by outlining where the two diverge (e.g. research which cannot be replicated hampers the goal of scientific progress). In the present model, we define the societal goal as a shared goal, i.e. something that individual agents have a genuine intrinsic interest in. For instance, scientists may be assumed to share a motivation to produce knowledge or relevant research (even if this may sometimes entail compromising on what would optimally serve academic success). Indeed, there are numerous reports of individuals in competitive systems who state that competition is impeding their ability to act according to the societal goal (see [4,5,13] for examples from science and banking). Similarly, a substantial fraction of the general population personally feels that competitive markets force them to do work which they feel does not contribute to overall societal welfare (the presumed goal of competitive markets) [26].

### 1.2.3. Why proxy and goal differ

It is important to make clear why a proxy is unlikely to perfectly reflect any underlying societal goal. As mentioned above, proxies generally reflect our best attempt at capturing an abstract goal given practical constraints. For instance, proxies, particularly if they are intended to mediate competition, have to be created within limited time. Information about longer term outcomes, which may still be relevant to the societal goal, cannot be captured in a proxy mediating ongoing competition. Ideally, journals would know if a study will be replicable before acceptance, but this information is simply not available in the short term [40], and can thus not enter publication-record-based faculty decisions [1]. Similarly, the proxy must be aggregated with finite resources, limiting the information it can contain [10]. For instance, the number of test items in a standardized test, or the number of reviewers in peer review is necessarily limited. In practice, proxies must thus generally rely on correlations, heuristics and sampling (consider any example in table 1).

Additionally, there are mechanisms that will tend to actively undermine the information content of the proxy: we, to our knowledge for the first time, note that *any proxy measure in a competitive societal system becomes a target for the competing individuals (or groups)*. This suggests that *any* competitive societal system is susceptible to Goodhart's Law, which states that 'When a measure becomes a target, it ceases to be a good measure' ([19–21], see §1.4). A fundamental insight of recent research has been that this occurs not only via the types of mechanisms traditionally studied by economists [13,32,41] or psychologists [42,43], but also via more abstract statistical mechanisms [36,44–46]. One such statistical mechanism is the above-mentioned cultural evolution [1]. Finally, note that in contrast to traditional

---

[1]Why and when a consistent ranking can be described by a scalar has been developed within the economic revealed preference approach [34,35] (in the present context *revealed competitive rankings*). We will further develop the information theoretic concept in section §3.2, relating it to an objective function in machine learning [36]). The degree to which systems produce inconsistent competitive rankings can most easily be conceptualized as a random noise term, and is not further considered here.

signals as studied in economics or ecology [47,48], there is no automatic constraint on the veracity of the proxy (as discussed in §§3.1 and 3.3). We thus suggest that a social system's behaviour under imperfect proxies should be the principal subject of study, with perfect proxies only interesting as a theoretical boundary case.

### 1.2.4. The proxy as focus point of theoretical and empirical inquiry

We propose a theory of *proxyeconomics* should centre on the informational content of competitive proxies and their consequences.

The *proxy* is a promising focus of inquiry precisely because its underlying competitive mechanisms can be empirically dissected and used to predict problems. For instance, an analysis of the academic publication process reveals the phenomenon of publication bias, i.e. a systematically increased probability that positive as opposed to negative findings are published. This informational idiosyncrasy leads to robust distortions of the published literature [40] and individual publication records (i.e. the proxy). It allows empirically verifiable predictions about potential differences between proxy and goal ([49], see §4.7). Similarly, quarterly earnings reports of a company will lead to a predictable focus on outcomes realizing within this specific time frame. The potential resulting short-termism may undermine the underlying goal in predictable ways. In each case, empirical idiosyncrasies in the mechanisms which generate the proxy, allow predictions about exactly which types of information will be lost, and which types of problems should be expected.

Another important factor which can be empirically analysed is how proxy measures psychologically affect the decisions of the competitors within a system. For instance, the journal impact factor elicits a rapid and reliable neural reward signal among neuroscientists [50], suggesting a powerful effect on decisions. Indeed, the ways in which competing agents perceive proxies are likely to affect psychological decision mechanisms in a number of highly systematic ways (further discussed in §4.2).

Finally, supraindividual mechanisms such as cultural evolution need to be taken into consideration (§1.3). Such system-level statistical forces may be determined predominantly by emergent proxies, leading to a form of Goodhart's Law [1].

In sum, we believe a focus on *the proxy* and its relation to the underlying *goal* can drive (i) the abstract conceptual investigation of proxy-based competition (as undertaken here, §2), (ii) experimental investigations of the relevant mechanisms (§§3.2, 3.3 and 4.2), and (iii) empirical investigations of real systems including the design of mitigation strategies (see §§2.4.4 and 4.7). Additionally, as the present agent-based model will illustrate, the proxy can help to integrate separate disciplinary approaches, such as economics and cultural evolution theory into a coherent framework.

## 1.3. Cultural evolution

Cultural evolution theory posits that human behaviour is in large part determined by culture in addition to individual cognition ([30,38,51], see [52] for an excellent introduction). The fundamental insight is that our actions are intricately shaped by cultural forces such as norms, or because we have learned it from a supervisor, role model, or peer. From this perspective, the transmission and selection of cultural content, such as scientific [1] or corporate [30] practices, is thought to resemble biological evolution. Indeed, substantial empirical evidence suggests that cultural evolution affects almost all aspects of cognition and behaviour, as is forcefully evidenced in, for instance, the languages we speak [52–54].

Accordingly, multiple scholars have suggested that cultural evolution theory will prove crucial to understand and address the pressing societal challenges of our time [28,55]. The reason is that the selection of cultural practices adds a system-level dynamic, which may be difficult to capture with any other theoretical toolbox. We propose the *proxy* as the cultural evolutionary equivalent of *fitness*, in a given competitive societal system. It presents a summary metric describing which cultural practices or entities will be selected in the competitive system (as in [1]).

In this paper, we will use the term 'practice' to denote the cultural evolutionary equivalent of a gene. A practice may be anything that determines human (or organizational) behaviour and is at least partially culturally transmitted, such as professional and organizational procedures and technologies [30]. Notably, this includes many antecedents of cognition and behaviour, such as perception, language, norms and moral valuations, all of which have been shown to be shaped by culture [52,54]. The 'fitness' of a practice describes its ability to reproduce because it is 'selected' within the competitive societal system. For instance, a successful academic could train the next generation of academics, transmitting her scientific practices [1]. A successful company could grow and produce spin-offs or

imitators, transmitting business model, corporate culture, organizational procedures and technology [30,56]. Assuming that this learning or imitation procedure is imperfect, or complemented by individual creativity, introduces variability into practices. This provides the final necessary ingredient for an evolutionary framework. Practices selected by the respective competitive system, and its implied *proxy*, will come to dominate and form the basis of the next generation of practices. The resulting notion of cumulative cultural evolution suggests that individual (or organizational) practices are built on an immense body of 'cumulated' cultural information, which goes far beyond what any individual can understand [38].

However, recently, Mesoudi & Thornton [38] have noted that the concept of fitness in cumulative cultural evolution theory remains 'undertheorized', raising the question: cumulation towards which goal or with respect to which measure? The predominant 'goal' within the cultural evolutionary literature to date seems to be 'inclusive genetic fitness' [52,57]. However, while this is plausible in ancestral societies, it is unsatisfactory in the present, both from an explanatory [38,58] and a normative perspective [31,55]. Indeed, cultural evolutionary forces appear to actively undermine genetic fitness, as evidenced in low fertility rates throughout modern societies [58–60]. For most cultural competitive systems (e.g. academia), it seems absurd to posit their goal as maximizing anyone's genetic fitness. Yet any other notion of a goal is typically excluded from cultural evolutionary model specifications (see e.g. [1,57]). Indeed, the rejection of abstract goals or purpose in cultural evolutionary models, may be described as a central methodological tenet. It is justified by the plausible assumption that selection is fundamentally agnostic to any such potential goal. However, it is difficult to make sense of the complex societal systems listed in table 1 without positing some underlying goal or purpose.

A recent exception is a study by Smaldino & McElreath [1], who model the 'natural selection of bad science'. The authors develop a notion of rigorous, replicable science as the goal of the academic system. They then model how culturally transmitted research methods should be expected to evolve in the light of academic competition based on the proxy measure of 'publication output'. Given the central assumption, that there will always be cultural practices which increase output by compromising on 'rigour', they find that less rigorous methods will come to dominate a population under most plausible circumstances. Importantly, their model 'does not assume that anyone actively alters their methods in response to incentives'. Instead, 'selection pressures *at the institutional level*' led to the evolution of fully proxy-oriented practices, which lack rigour and thus entail poor scientific quality (ibid., emphasis from the original). While the underlying assumptions are highly plausible, and their overall case powerful, Smaldino & McElreath acknowledge that they may be missing some countervailing forces. Indeed, they do not incorporate any mechanism whereby individual agents could perceive, or be influenced by the abstract goal. This is in line with the general cultural evolutionary tradition, in which any such mechanism might be seen as highly dubious. Accordingly, Smaldino & McElreath are left with no other option than to point to a vague possible impact of researchers' personal incentives and goals during their qualitative discussion.

Here, we build on Smaldino & McElreath and the more general cultural evolutionary literature in two ways.

— First, we extend the notion of proxy-based cultural evolution to proxy-based competitive societal systems in general, dramatically expanding the scope and relevance of Smaldino & McElreath's findings (table 1). In this view, the competitive societal system reflects an attempt to translate a shared abstract goal into a cultural selection mechanism. To use ecological terminology, the competitive system represents a *constructed cultural niche*, i.e. a socially constructed selection environment [38,61]. The proxy is a convenient summary description of the selection pressures within the niche. Since this selection environment is constructed to serve a social goal, the proxy can also be conceived as a summary 'signal'[2] of the goal, on which selection is based.

— Second, we introduce individual agency as a mechanism for the abstract goal to influence the system dynamic. This deviates from previous cultural evolutionary literature, in that it not only allows the reintroduction of purpose into cultural evolution, but also explores a mechanism whereby this purpose can affect system dynamics. While we come to the details below, it should here be stated clearly that our implementation respects the foundational tenet that cultural selection is agnostic to the abstract purpose. Instead, the effect of a shared social purpose is introduced via individual-level agency.

---

[2]Note that the proxy is distinct from traditional signals in evolutionary (and economic) theory in several ways (see §§3.1, 3.3), most importantly in that it cannot be assumed to necessarily be informative.

## 1.4. Goodhart's Law, incentive misalignment and multitasking

In their contribution, Smaldino & McElreath [1] describe their cultural evolutionary mechanism as an instance of Goodhart's or Campbell's Law. It is worth noting that, while the underlying principle may be best known by the economist Charles Goodhart's name, his precedence as the originator of the eponymous law is disputed [62]. In the following, we will first briefly outline some historical context, including recent developments, concerning Goodhart's Law and its analogues. We will then come to the economic literature on incentive misalignment and multitasking, which we draw on to reintroduce agency and purpose into the cultural evolutionary model.

As early as 1936, Robert K. Merton coined his eponymous *law of unintended consequences*, noting that 'In some one of its numerous forms, the problem of unanticipated consequences of purposive action has been treated by virtually every substantial contributor to the long history of social thought' [63]. While Merton's Law captures the general difficulty of control in complex systems, most of the economic literature examines a specific approach to control, namely the provision of incentives contingent on some measured performance. An influential early contribution within the business literature is the classic paper by Steven Kerr: 'On the folly of rewarding A, while hoping for B' [41]. In such cases, the unintended consequence is, quite predictably, that agents will tend to do what they are rewarded for, instead of pursuing the original goal. We will return to the economic literature below, but first want to outline some of the numerous related concepts from other disciplines. These appear to have independently discovered variations of the driving phenomenon, some preceding both Goodhart & Kerr [62]. Most famously, the sociologist Donald T. Campbell coined the eponymous law: 'The more any quantitative social indicator is used for social decision-making, the more subject it will be to corruption pressures and the more apt it will be to distort and corrupt the social processes it is intended to monitor' [1,8–10,19]. Campbell pioneered the study of the corruptive effects of high-stakes standardized testing in education. Additional related concepts are too numerous to list (e.g. [64–66]). Yet the most pithy, and best-known phrasing of the law, probably furnishing Goodhart's claim to its name, was coined by Marylin Strathern (within the context of university auditing [21]). Most recently, it has been invoked in academic publication culture [3,67], within artificial intelligence research [36,68,69] and even ecology [44]. The formulations by Campbell and Goodhart draw attention to the central role of quantitative proxy measures, i.e. the measures used to link an abstract goal or purpose to incentives. They start with the simple insight that no measure is perfect, as outlined above (§1.2.3). More problematically, they assert that the use of the measure for control purposes will tend to corrupt it. A recent surge of attention to Goodhart's Law in artificial intelligence research is making clear that it should be expected to occur not only through human behaviour (e.g. because of incentive misalignment [32]), but via statistical effects in any optimization mechanism [36,45]. The cultural evolutionary mechanism explored in [1] reflects exactly such a mechanism. This is important because it may override the effects of individual decisions at the system level, but it has not been considered in economic treatments [32].

Next, we introduce the rich economic literature on incentive misalignment, and specifically multitasking, on which our model draws [13,32,70–73]. Since the proxy is likely to act as an incentive, which is misaligned with the underlying social goal, this body of research is highly relevant in the present context. In particular, the *multitasking* framework [33] provides an elegant formalization to allow for both an intrinsic (or moral) and an extrinsic (or competitive) incentive to affect individual decisions. In the present context, we may for instance assume that scientists share a genuine intrinsic incentive towards the goal of producing 'good science', while simultaneously being subject to extrinsic competitive incentives. Economists typically consider which type of pay scheme an employer should use to optimally structure the extrinsic incentive for employees when performance cannot be perfectly measured. The canonical result of such models is that poorer performance measures should be met with weaker incentive contracts (paradigmatically a lower optimal slope of a linear incentive contract) [32]. In other words, the less well a proxy measure reflects an underlying goal, the weaker should the incentives based upon the proxy be. In the extreme, i.e. if proxies are useless, employers may be best served by resorting to fixed-wage contracts, i.e. removing all performance-contingent pay. Interestingly, one of the main puzzles driving the creation of this literature has been the prevalence of flat-wage contracts in the real economy [74], suggesting poor proxies are ubiquitous in business.

We will draw on the underlying formalization below, to reintroduce the notion of a shared social goal into our combined economic-evolutionary model. However, one important note is in order, which distinguishes the 'organizational goals', typically considered in the economic literature, and the more abstract goals of competitive social systems discussed here (table 1). Briefly, economists often

implicitly assume that only 'optimal' incentive contracts, i.e. those which optimally serve e.g. corporate organizational goals, will survive market selection (e.g. [32]). In other words higher-level market selection is assumed to provide a natural constraint on the potential adverse effects of poor proxies or inappropriately strong extrinsic incentives. It is precisely this reasoning that underlies the inference that fixed-wage contracts must frequently be optimal, from the observation that they are common in the real economy. Such an assumption seems highly dubious in the present context, since it is unclear what would constitute the 'higher-level selection' in the context of whole societal systems. This subtle, but immensely important point is further discussed in §3.1. For now, we simply want to note that it suggests Goodhart-like phenomena, and resultant incentive misalignments, may be substantially more important in the context of competitive societal systems than in business.

In sum, as noted above, *any proxy measure in a competitive societal system becomes a target for the competitors*, fostering Goodhart's Law. This suggests proxy and goal incentives may generally be misaligned to some degree within competitive systems. This in turn suggests that the mechanisms investigated within the economic multitasking literature can inform the analysis of proxy-based competitive societal systems in general (table 1). However, there are important caveats. Firstly, system-level statistical mechanisms, such as cultural evolution must be taken into account, as they may override the effects of individual decisions. Secondly, economists generally focus on incentive misalignment and the related topic of signalling within corporations or between market participants, relating it to a presumed ground truth, which is revealed and enforced by markets [75]. In the present case, it is questionable if such assumptions are warranted. Instead, market selection itself is conceptualized as reflecting an imperfect proxy. In other words, we propose to recast the *a priori* assumption that market proxies perfectly reflect an underlying social goal [18,75] as a central empirical question. The final sections of this paper will attempt to outline how this question can be systematically addressed in practice (§§4.6 and 4.7).

# 2. An agent-based model of proxy-based competition

## 2.1. Model rationale and questions

We model a small society of agents [76,77] in a competitive system, where competition is mediated by an imperfect proxy measure of a shared social goal. As reviewed above, there are currently several separate literatures and modelling traditions capturing different aspects of proxy-based competition and Goodhart's Law, most prominently (A) economic and (B) evolutionary approaches.

(A) Economic approaches capture many positive effects of competition, such as effort incentivization and screening for talent ([78], §1.4). Additionally, so called multitasking models allow to incorporate the combined effect of intrinsic (e.g. moral) and extrinsic (e.g. competitive) incentives on effort [13,32,33,74]. Here, we draw on such a multitasking paradigm to allow an 'intrinsic' incentive to affect the decisions of individual agents in parallel to an 'extrinsic' proxy incentive. Such a combined effect is well grounded in empirical research [42,43,79,80].

(B) Cultural evolutionary approaches do not typically allow for individual agency or the notion that cultural selection serves a purpose or shared social goal (e.g. [1,57]). Instead, they model a cultural dynamic which is independent of individual agency and essentially undirected (§1.3). Here, we draw on a cultural evolutionary agent-based model of Goodhart's Law by Smaldino & McElreath [1]. The authors model a system in which cultural practices are selected based on a competitive proxy measure, finding a robust evolution towards fully proxy-oriented practices. However, their model does not allow to incorporate the effect of individual agency, a shared social goal, or even effort incentivization. It thus neglects many positive effects of competition, as well as the potential constraining effect of individual morality on the evolution towards full proxy orientation.

We combine these two paradigms, in order to explore the interactions between the economic (A) and evolutionary (B) mechanisms, while capturing both positive and negative effects of competition. This relates the proxy to the evolutionary and economic concepts of *signalling* (further discussed in §§1.3 and 3.3). Beneficial signalling/screening (for talent) interacts with detrimental signalling/ screening (Goodhart's Law), such that the ultimate value of the signal is an emergent system property.

In order to combine (A) and (B) it is necessary to formalize (i) proxy fidelity to the goal and (ii) proxy-based competition, in a way that captures their expected respective influences. Specifically, (i) proxy fidelity should directly affect only the economic mechanism, because the abstract social goal is

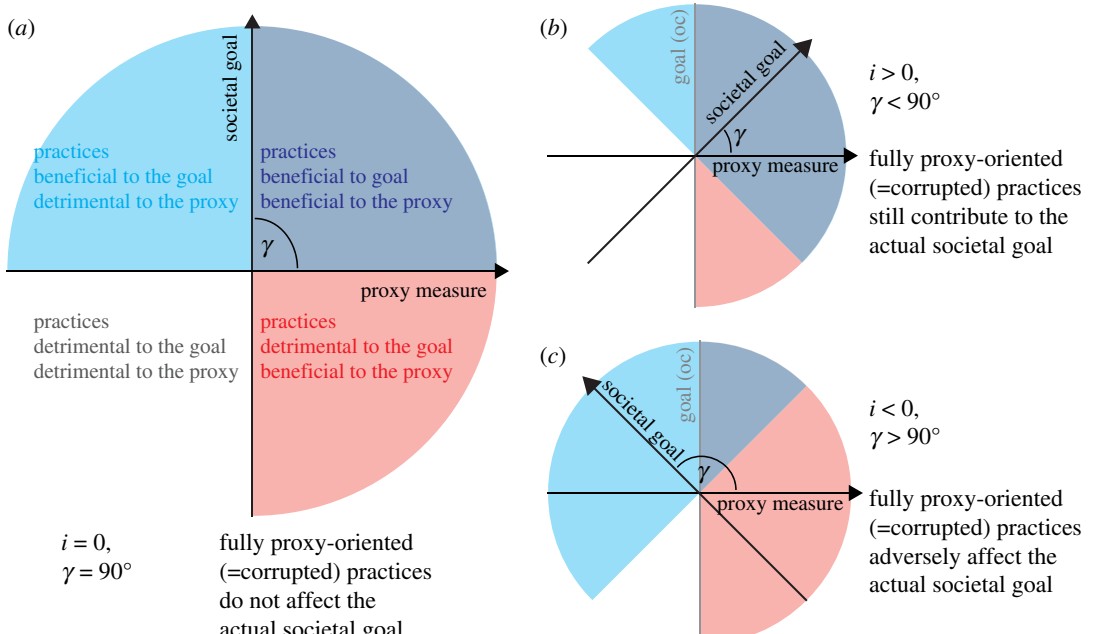

**Figure 1.** The practice space, Individual practices may contribute to different degrees to the proxy measure and the societal goal, motivating the mapping of all practices to a two-dimensional practice space. Sections of the practice space which are beneficial/ detrimental to the proxy/goal are colour coded (see coloured labels in (*a*)). The degree to which the proxy captures true and incorruptible information about the societal goal can then be represented as the angle between the main axes, denoted goal angle ($\gamma$) with information ($i = \cos(\gamma)$). Exclusively proxy-oriented practices are located on the horizontal axis (proxy measure). Accordingly, when such (corrupted) practices are neither beneficial nor detrimental to the societal goal, then $\gamma = 90°$ and $i = 0$ (*a*). Presumably, in most cases, the proxy captures sufficient incorruptible information about the goal such that even fully proxy-oriented practices contribute to the societal goal ($0° < \gamma < 90°$; $1 > i > 0$ (*b*). However, full proxy orientation may also lead to negative outcomes for the actual societal goal ($90° < \gamma < 180°$; $0 > i > -1$ (*c*). When $\gamma \neq 90°$ the goal component orthogonal to the proxy is labelled 'goal (oc)'.

invisible to the evolutionary mechanism. By contrast (ii) proxy-based competition should affect both individual incentivization and cultural evolutionary selection. Individual agents both want to succeed in competition and may be selected out of the system if their proxy performance is insufficient. To this end, we defined (i) a *practice space*, capturing the fidelity with which the proxy reflects the goal and (ii) a competitive survival threshold, relating the psychological and cultural evolutionary effects of competition. We will now introduce each in turn.

### 2.1.1. The practice space—proxy fidelity to the goal

The practice space defines the relative contributions to proxy and goal, for all possible cultural practices. A cultural practice is defined as any specific pattern of actions that can be learned, i.e. culturally transmitted. Four fundamental groups of practices can be intuitively differentiated based on the degree to which they are beneficial or detrimental to the proxy and goal (figure 1*a*). For instance, in any specific societal context, we may consider if practices exist that exclusively contribute to either the proxy measure or the societal goal. The existence of such practices motivates the dimensional reduction of the practice space to two dimensions representing the proxy and the goal. The degree to which the proxy captures true and incorruptible information about the societal goal can then be represented as the angle between the main axes (goal angle; $\gamma \in [0°, 180°]$). Similar formalizations are routinely used in economics [13,72,74]), but have also been used to describe practices in scientific [1] as well as political [12] competition. A convenient way to describe proxy fidelity is by an information measure $i \in [-1, 1]$, given simply by $i = \cos(\gamma)$.

Intuitively, $i$ measures the effect of fully proxy-oriented practices on the actual societal goal. A good proxy ($0 < i < 1$) captures sufficient information about the societal goal, such that even fully proxy-oriented practices produce positive outcomes for the goal (figure 1*b*). By contrast, when full proxy orientation produces negative externalities which outweigh the beneficial effects for the societal

goal, this is captured by $-1 < i < 0$ and $90° < \gamma < 180°$ (figure 1c). Externalities beyond the dimensions described above, i.e. that are independent of both proxy and goal (e.g. leisure time), are not considered in the current model. Note that small $i$ does not imply the proxy is useless, since with a given set of practices it may still correlate highly with the societal goal. However, small $i$ implies the proxy is corruptible (or distortable [32]), as alternative practices which contribute less to the societal goal may more efficiently optimize the proxy.

### 2.1.2. Proxy-based competition

In order to relate the cultural evolutionary and psychological sides of competition, we define a competitive *survival threshold* separating 'winning' from 'losing' agents. 'Losing' in the cultural evolutionary context means to be at risk of being selected out of the system due to insufficient performance. 'Winning' means potentially reproducing (i.e. passing on one's cultural practices, see §1.3). Intriguingly, the perhaps most prominent empirical theory describing individual utility, namely prospect theory, has been explicitly related to evolutionary selection [81,82]. Prospect theory describes an empirically validated [83–85], s-shaped utility function, which is aligned to some salient reference point (see §2.2.3 for details). Agents below this salient reference point are in a 'loss frame', and are powerfully incentivized to supersede it. This (as well as the overall curvature) of the prospect function suggest that its salient reference point is related to a survival threshold [81,82]. Indeed, this notion has received empirical support in experimental economic contests [86–88]. Additionally, such an s-shaped, reference-dependent utility function allows to specify a neuroscientifically plausible [89], concrete effort choice heuristic (§§2.2.3 and 2.2.4).

We thus define competition ($c \in [0, 1]$) as the relative proxy performance defining the *survival threshold*. For instance, intense competition ($c = 0.9$) means that only the top 10% of proxy performers (A) have positive utility from competition (§2.2.3) and (B) are protected from being selected out of the system (while becoming eligible to reproduce, §2.2.5). Note that the *survival threshold*, thus defined, depends on the emergent population distribution of proxy performances. Curiously, this very simple and empirically driven way to compute competitive effort incentivization appears not to have been explored in the economic literature so far, perhaps due to a predominant methodological tradition of analytic equilibrium modelling [90]. To capture that more intense competition entails more selection, $c$ also modifies the probability of selection events (see §2.2.5). Finally, to capture that evolution represents a slower, stochastic process as compared with economic decisions, we introduce a parameter determining the relative speed and probability of selection, namely selection pressure ($p \in [0, 1]$).

These operational choices allowed to model the expected complex feedback loops within and between individual and system level. The informational divergence between proxy and goal at the system level will affect individual decisions. The competition experienced by individuals relates to the competitive selection at the system level. Finally, and most importantly, proxy performances affect outcomes primarily as a relative measure, i.e. by their relation to other agents' proxy performances, via the emergent *survival threshold*. The latter implies feedback loops between individual agents for (i) psychologically driven proxy performance, (ii) selection-driven proxy performance and (iii) between the two processes. In other words an agent (i) knows she needs to supersede her peers, (ii) may be selected away if she does not supersede her peers, and (iii) will find that the proxy performances of her peers, and thus her own, must reflect both the above.

Given these operational choices, we were interested in the following main questions:

— First: how will the positive and negative effects of competition play out at both the individual and the system level? In particular, how will (i) effort incentivization, (ii) screening for talent, and (iii) screening for proxy orientation trade off for different competitive intensities ($c \in [0.1, 0.9]$) Does an optimal level of competition arise with respect to the shared social goal? What will be the emergent distributions of individual agents underlying the outcome? And finally, how will these outcomes develop dynamically, given the dual time courses of economic and evolutionary processes?

— Second: will the system-level evolutionary selection mechanism override the effect of individual decisions, as tacitly suggested by [1]? The underlying reasoning is, that goal-oriented agents might simply be selected out of the system, quite independent of their individual decisions. This may undermine any potential impact of an intrinsic incentive on system dynamics.

## 2.2. Model description

We simulate a societal system, in which $N$ utility-maximizing agents repeatedly compete with each other based on individual proxy performances. Individual performances result from both the individual's effort and the orientation of their practice towards the proxy. The system is characterized by the following: the fidelity of the proxy, parametrized as the *practice space* by the goal angle ($\gamma$); competition ($c$), parametrized as the fraction of losers; and cultural evolution, parametrized through a practice mutation amplitude ($m$) and selection pressure ($p$). Individual utility-maximizing agents are characterized by talent ($t$) and a parameter determining the power of an intrinsic moral incentive towards the societal goal ($g$). Finally, the individual agent's properties of effort ($e$) and practice orientation ($\theta$) represent the main outcomes of interest, informing on overall productivity and the direction of that productivity. An overview of the model's structure is presented as pseudocode 1.

---

**Pseudocode 1.** Model structure.

**initialize world and agents**
— create practice space ($\gamma$, §2.1.1)
— set competition intensity ($c$, §2.1.2)
— set selection pressure ($p$) and practice mutation amplitude ($m$), (§2.2.5)
— create $N$ agents, each with random talent ($t$) and practice angle ($\theta$), and initial effort ($e = 0$)

**step Model**
— economic decision phase: draw each agent in random order and **step Agent**
— evolution phase: recompute proxy rank and implement *evolution*

**step Agent**
— *optimize effort()*, (§§2.2.2 and 2.2.3):
   choose optimal effort ($e$) given utility from both goal ($U^g$) and proxy ($U^p$): $U^* = \arg\max_e f(U^g, U^p, t, e)$ where $U^g = f(g, \gamma, \theta, e)$ and $U^p(c, \theta, e, \bar{P})$, $g$ defines the strength of an intrinsic incentive towards the goal, and $\bar{P}$ represents the distribution of proxy performances of all other agents

*evolution()* (§2.2.5)
— identify potential 'winners' and 'losers' based on recomputed *survival-threshold*($c, \bar{P}$)
— with probability $c \cdot p \cdot N$, initiate *selection-event()*
   *selection-event()*: replace random 'loser' by random 'winner' passing on $\theta$ with noise: $\theta_{offspring} = \theta_{parent} + \mathcal{N}(0, m)$, and drawing new random talent ($t$)

---

### 2.2.1. Model time course

The model proceeds in time steps, each of which is subdivided into a decision phase and an evolution phase. In the decision phase, all agents are drawn in random order to adjust their effort ($e$) to maximize utility given their individual properties ($\theta$, $t$, see below) and the *currently* observed competitive rank (§2.2.2 and 2.2.3).[3] In the evolution phase, losing proxy performers are subject to stochastic death, and are instantaneously replaced by offspring from winning proxy performers, such that the population size stays constant (§2.2.5). To model the slower and stochastic nature of the evolutionary mechanisms, selection events only occur with some low probability ($p$), such that $p$ determines the approximate number of steps one can afford to be a loser before being actually removed. Together, this models a competitive system in which agents make their decisions based on the observed proxy performances of all other agents, but are unsure about the exact time frame of competition, i.e. to which degree other agents might still change their proxy performance before a selection event might occur.

---

[3]Currently, here, means at the time of the random draw, to model uncertainty about if or when other agents may react. However, note that the model outcomes were indistinguishable if proxy rank was calculated based on the previous time step (not shown).

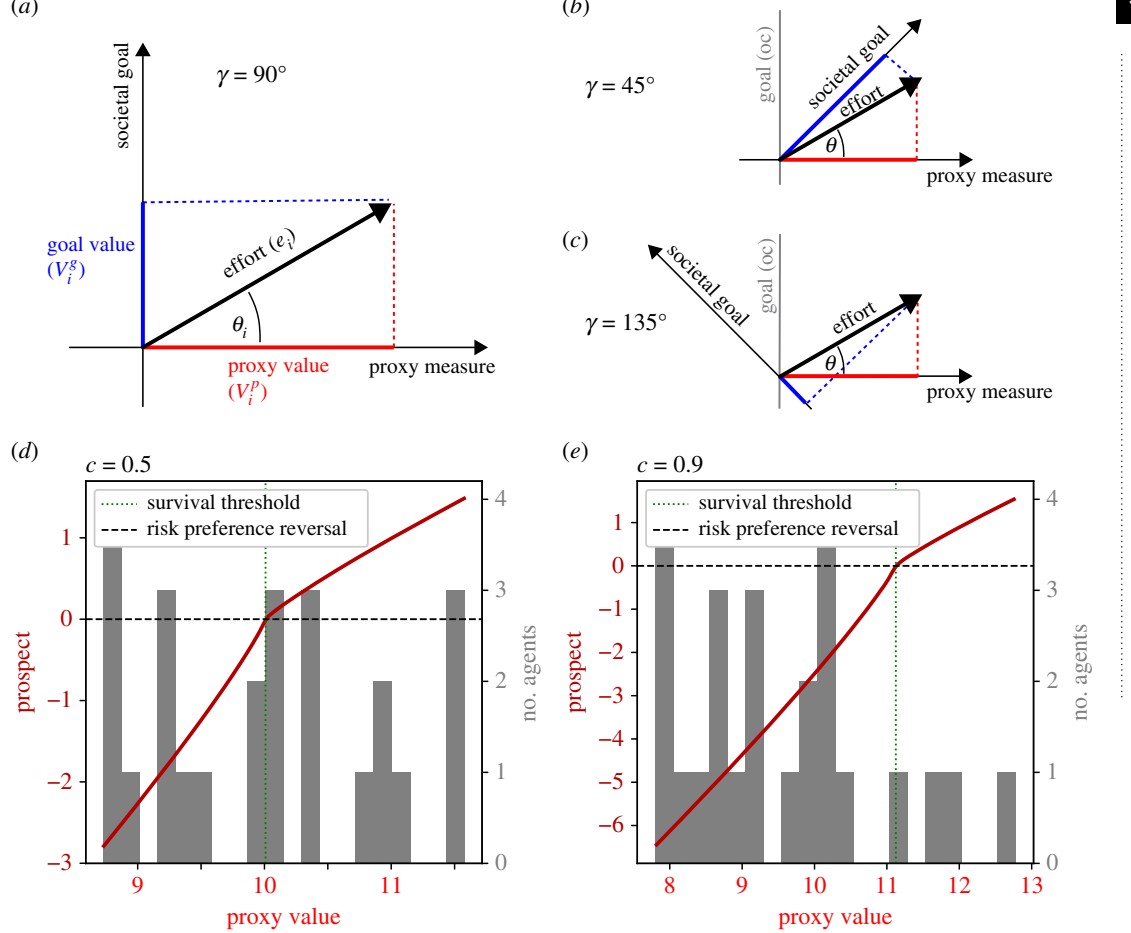

**Figure 2.** Agent decision model. Agents derive utility both from contributing to the societal goal and performing well according to the proxy measure (blue, red, respectively). An individual agent (index $i$) produces goal value ($V_i^g$) and proxy value ($V_i^p$) as determined by the practice angle ($\theta_i$) and effort ($e_i$), simply as the projection of the effort vector onto the main axes (equations (2.1) and (2.2); (a–c)), where the goal angle ($\gamma$) is the angle between the main axes. The utility derived in the proxy dimension (denoted prospect) depends not on the absolute proxy value but on the relative ranking of proxy values and the survival threshold according to equations (2.4) and (2.5). Panels (d,e) show illustrative prospect functions (dark red) for competition $c = 0.5$ and $c = 0.9$, respectively. The survival threshold (vertical spotted line in (d,e)) is the salient reference proxy value separating losers from winners, where competition ($c$) denotes the fraction of losers. Accordingly, for $c = 0.5$ or $c = 0.9$, the survival threshold is the 50th or the 90th percentile of the population distribution of proxy values respectively (grey histograms in (d,e), respectively).

## 2.2.2. Economic decision phase: multitasking

The system-level definition of the practice space (§2.1.1) allows us to describe the proxy orientation of a given agent's practices by their 'practice angle' ($\theta$) within this space (figure 2a–c). The agent's effort ($e$) then, is defined as the length of a vector along this practice angle. The resulting contributions to proxy and goal value are simply the projections onto the main axes. To allow both, the intrinsic (goal) and a competitive (proxy) incentive to elicit effort, we adapt an economic multitasking model (figure 2a). Note, however, that in contrast to traditional multitasking models, practice angles are primarily established through a process of cultural evolution. An additional robustness analysis shows how this differs from their establishment through individual choice, i.e. *practice agency* (figure 8d–f).

Accordingly, the effort of the $i$th agent $e_i \in \mathbb{R}_{\geq 0}$ is modelled as the length of the vector with orientation $\theta_i$ and the resulting proxy and goal values ($V_i^p$ and $V_i^g$ respectively) are the projections onto the main axes capturing the practice-dependent trade-off (equations (2.1) and (2.2)).

$$V_i^g(\theta_i, e_i) = \cos(\gamma - \theta_i) \cdot e_i \tag{2.1}$$

and

$$V_i^p(\theta_i, e_i) = \cos(\theta_i) \cdot e_i. \tag{2.2}$$

The utility derived from the goal performance ($U_i^g$) is simply the goal value multiplied by a constant ($g$), determining the psychological strength of an intrinsic, moral incentive towards the societal goal (equation (2.3)).

$$U_i^g(V_i^g) = g \cdot V_i^g. \tag{2.3}$$

### 2.2.3. Economic decision phase: competition as incentive

The utility derived from the proxy value ($U_i^p$) is determined via Kahneman and Tversky's cumulative prospect function [91], given the emergent competitive survival threshold (ST; equation (2.4)). Specifically, it depends on the difference between the own proxy value $V_i^p$ and ST (figure 2$d$,$e$).[4]

$$U_i^p(V_i^p, ST, \epsilon) = (V_i^p - ST)^{0.88}. \tag{2.4}$$

Additionally, as in prospect theory agents are loss averse, i.e. negative prospects are multiplied by 2.25 (equation (2.5), [91]).

$$U_i^p \mapsto \begin{cases} U_i^p, & \text{if } U_i^p \geq 0 \\ U_i^p \cdot 2.25, & \text{otherwise} \end{cases} \tag{2.5}$$

For sensitivity and robustness analysis, we additionally probed a simple step function as prospect function $[-1, 1]$ (figure 8$a$–$c$), or linear prospect function (electronic supplementary material, figures S3 and S4).

### 2.2.4. Economic decision phase: effort choice heuristic

To compute which effort choice leads to maximal utility (equation (2.6)), each agent ($A_i$) samples a small range of effort adjustments (see §2.2.6). This respects the limited computational capacity of both real humans and our computational agents.

$$U_i(U_i^g, U_i^p, \text{cost}) = U_i^g + U_i^p - \text{cost}(e_i, t_i), \tag{2.6}$$

where effort cost is given by

$$\text{cost}(e_i, t_i) = \frac{e_i^2}{t_i}. \tag{2.7}$$

Here, $t_i$ is the talent of the agent, i.e. a constant determining the relative cost of effort independent of $\theta$. In a subset of models (figure 8$d$–$f$), practice agency was introduced by letting agents maximize utility by adjusting both $e$ and $\theta$ during the economic decision phase.

### 2.2.5. Evolution phase

In the evolution phase, the ranking of proxy performances is reassessed to define 'winners' and 'losers' given the emergent survival threshold (ST, §2.2.3). A selection event is initiated with probability $c \cdot p \cdot N$, such that each potential loser ($V_i^p < ST$) is subject to professional death with probability = selection pressure ($p \in [0, 1]$). Upon each death a position opens up, which allows a randomly chosen 'winning' proxy performer ($V_i^p > ST$) to professionally reproduce, passing only her practice angle ($\theta$) on to her offspring (talent is randomly drawn).

During practice inheritance, $\theta$ mutates stochastically with normal distribution, such that $\theta_{\text{offspring}} = \theta_{\text{parent}} + \mathcal{N}(0, m)$. For robustness analysis, we also implemented a fitness-proportionate selection algorithm, where winners are chosen proportional to their proxy performance, and losers are chosen randomly (electronic supplementary material, figures S2–S4). As above, the

---

[4]Note that in the first version of this model we have considered a related s-shaped prospect function based on a Gaussian uncertainty distribution around the ST (based on [82]). Though the main results were similar (see manuscript history [92]), this function was not scale invariant with respect to proxy value.

**Table 2.** Parameter space.

| parameter | base value(s), range | description | main effect (if parameter is increased) |
|---|---|---|---|
| equilibrium determining parameters | | | |
| goal angle ($\gamma$) | **0, 45, 90**, _135_, 0–180 | angle [°] defining the amount of corruptible proxy information | hill-shaped effect on equilibrium corruption, the optimal level of competition decreases |
| goal scale ($g$) | **0,1**, _2_, 0–10 | scaling factor of psychological goal valuation (relative to experienced prospect value) | increased effort, decreased equilibrium corruption |
| dynamics determining parameters | | | |
| competition ($c$) | _0.3_, **0.6, 0.9**, 0.1–0.9 | competitive pressure, i.e. the fraction of potential losers per round | complex effects on effort, evolution and utility (see main text), increased speed of convergence to equilibria |
| selection pressure ($p$) | _0.001_, **0.1**, 0.001–1 | probability of death for each losing agent in each round | increased speed of convergence to equilibria |
| parameters affecting variability | | | |
| talent standard deviation ($t_{sd}$) | **1**, 0–6 | standard deviation of talent within the population | increased effort spread |
| population size ($N$) | **100**, 10–500 | number of agents in the system | decreased variability over models |
| practice mutation rate ($m$) | **2**, 0–30 | standard deviation (°) of practice angle mutations during inheritance | increased practice variability and, if the equilibrium practice is outside the initialization range, speed of convergence to equilibria |

Overview of the parameter space. During sensitivity analysis individual parameters were varied in the specified ranges holding all other parameters at the base values. Each range was probed with 4–9 increments (see model code or data points in figures 7 and 8). Base values from presented data are in boldface (figures 3–8). Base values in italics were additionally probed during sensitivity/robustness analyses.

frequency of selection events (i.e. the number of agents selected at each round) is proportional to $c$ (specifically $c \cdot p \cdot N$).

### 2.2.6. Implementation and parameter choice

The model was implemented in the agent-based modelling framework 'Mesa0.8.6' in Python3.6 and run on a standard Windows 7, 64 bit operating system. The full code to run the model and generate figures is attached as electronic supplementary material and is available at https://github.com/oliverbraganza/Proxyeconomics_original.

Table 2 shows an overview of the explored parameter space with the parameters shown in the main figures highlighted in boldface (also see figure 7). A number of additional parameters were varied to probe robustness, but led to no change, and will be reported at appropriate locations throughout the paper. The model was initialized with population size $N$ (typically 100), where every agent received a practice angle $\theta$ drawn randomly from a uniform distribution between proxy and goal axes $\mathcal{U}(0, \gamma)$ and initial effort 0. In one sensitivity analysis, $\theta$ was initialized as $\theta = \gamma = 45°$, implying all initial practices are completely goal-oriented (figure 7e,f). Model runs were typically repeated 10 times for each level of competition to obtain measures of the mean and spread (standard deviation) of system behaviour. In each model run, all agents in a population compete against each other via their proxy performances as described above.

In order to practically compute the optimal effort (§2.2.4), each agent probed the following effort adjustment list $[-10, -5, -1, -0.5, -0.1, 0, 0.1, 0.5, 1, 5, 10]$, computing their expected utility for each imagined effort change. Note, that alternative effort choice lists (e.g. −10 to 10 in 0.1 increments) did not change model outcomes but dramatically increased the computational burden of agents (and model). Furthermore, the specific range (−10 to 10) simply covers the magnitude of plausible effort changes in the system arising from the arbitrary mean talent choice of 10, and choosing a larger range did not change outcomes. In robustness analyses including angle agency, the utility effect of effort adjustments were calculated for each of a range of practice adjustments in an angle list of $[-5°, -1°, 0°, 1°, 5°]$, following the same rationale.

## 2.3. Model results

The combination of the economic and the evolutionary mechanism in our agent-based model of proxy-based competition yields four main results (R1–R4). We will begin with an overview to illustrate how R1 to R4 build on each other, culminating in the most interesting R4:

(R1)  Individual agents
 (R1.1)  display complex heterogeneous effort choices depending on $\theta$ and $t$, as well as other agents' proxy performances (a mostly positive effort-incentivization effect of competition),
 (R1.2)  are screened for talent (a positive effect of competition), and
 (R1.3)  are screened for proxy orientation (a negative effect of competition).
(R2)  The combination of R1.1 to R1.3 led to an optimal level of competition. Beyond this level the system displays (i) more proxy-oriented practices, (ii) unhappier agents, and (iii) less overall effort.
(R3)  The shared social goal is better served with (i) better proxies (smaller $\gamma$) or (ii) a stronger intrinsic motivation towards the goal (higher $g$).
(R4)  In the long term, the intrinsic motivation can bound the evolution to full proxy orientation leading to an equilibrium distinct from that resulting from either economic or evolutionary mechanism in isolation. The optimal level of competition is shifted to lower intensity. Beyond this optimal level, the equilibrium entails (i) similar practices, (ii) unhappier agents and (iii) less overall effort.

We will now present each result in turn, concluding with a number of sensitivity and robustness analyses.

### 2.3.1. Competitive effort incentivization (R1.1)

To introduce the model, we consider a system with an imperfect but informative proxy ($\gamma = 45°$; $i \approx 0.7$). Goal scale ($g$) is set to $g = 0$ in order to first explore the system without an intrinsic incentive. Selection and mutation are mild ($p = 0.1$, $m = 2°$), implying a 10% probability of death for losers at each time step and small changes of practice orientation during inheritance. At initialization, each agent receives a random (uniformly distributed) practice angle $\theta = \mathcal{U}(0, \gamma)$ and random (normally distributed) talent $t = \mathcal{N}(10, 1)$. We will now first describe how the s-shaped prospect function gives rise to realistic dynamic effort patterns, including (i) powerful incentivization, (ii) an emergent discouragement effect, and (iii) a resulting bimodal distribution of effort. We will then describe how the emergent effort choices of individual agents relate to their respective proxy orientations ($\theta$), revealing complex competition-dependent patterns.

The dynamical effort incentivization effect can best be illustrated by inspecting individual effort trajectories for different competitive intensities (figure 3a, four exemplary model runs ranging from very low to very intense competition). At each time step, agents adjust their effort level to optimize the utility, given the other agents' proxy performances. In higher competition, individual agents are forced to increase effort more to cross a higher survival threshold. This in turn affects the population distribution of proxy values leading to a positive feedback loop and higher effort levels in higher competition. When marginal effort cost begins to outweigh the expected utility gain, individual agents may stop increasing or begin to decrease effort. Indeed, the s-shaped form of the prospect function (figure 2d,e) leads to a 'discouragement' effect when the observed survival threshold is far beyond the own proxy performance. In other words the diminishing returns (i.e. flattening) of the prospect function induces agents with no current hope of winning to decrease effort (figure 3a, e.g. black arrows). Discouragement becomes more frequent in more competitive systems, leading to an eventual reversal of the competition to effort relation for some agents, as observed empirically [93–95]. Increasing prevalence of discouragement with higher competition also entails an increase in variance

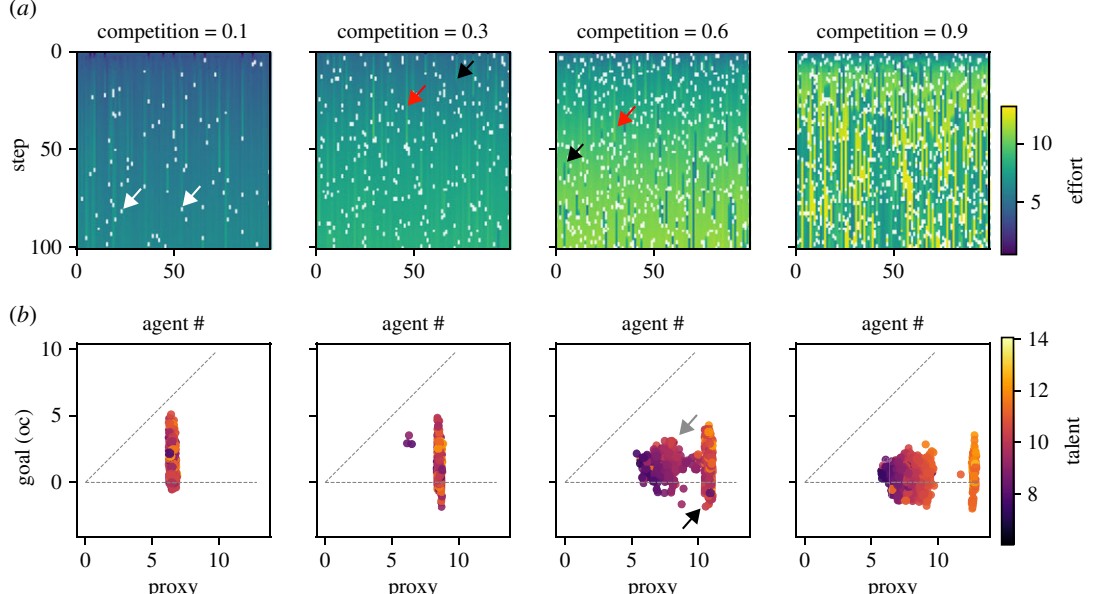

**Figure 3.** Agent dynamics (R1) (100 time steps), with $N = 100$, $\gamma = 45°$, $\theta_{\text{initialization}} = \mathcal{U}(0°, \gamma)$, $g = 0$, $t = \mathcal{N}(10, 1)$, $p = 0.1$, $m = 2°$. Data are collected from 10 model runs per competition level, $c$. Data for four levels of competition are shown as major columns. (a) Effort dynamics, each sub-panel shows effort over time for each of the 100 agents (random run for given $c$). Columns within sub-panels represent individual agents or positions. White squares indicate death/birth events (two examples highlighted with white arrows). Black arrows indicate examples of agents becoming discouraged by competition. The red arrow indicates an example of an agent increasing effort to gain the prospect of winning. (b) Agent performance in the practice space, Realized proxy and goal values of each agent at the 100th time step. Values are the result of the current chosen effort given the individual agents practice angle (compare figure 2b). Each data point represents an agent (agents accumulated from 10 runs per panel). Talent is colour coded. The black and grey arrows indicate highly proxy or goal-oriented agents, respectively. Note that the agents indicated by the grey arrow are outperforming those indicated by the black arrow concerning their goal performance, but not proxy performance.

of effort at the population level. At the individual level, agents similarly show more variable effort over time in higher competition. Note how some agents progressively increase effort in order to compete, but eventually become discouraged (figure 3a, black arrows). The resulting decrease in the survival threshold may in turn provide other agents with a prospect of winning, such that they can gain utility by increasing effort (figure 3a, red arrow).

The population distribution of individual effort levels can be more clearly seen by projecting the realized practice-effort vectors of individual agents back into the practice space (figure 3b). Here, each data point represents the end point of the effort vector (as in figure 2b) of an individual agent at time step 100. The resulting goal and proxy performances correspond to the projections onto the main axes, as shown in figure 2b. Across all levels of competition, we observe a dominant effect of the competitive incentive on realized effort levels, as outcomes tend to organize in vertical lines, i.e. they cluster around a specific proxy value. Further analysis showed that the vertical line corresponded to proxy values just above the emergent survival threshold for a given level of competition. Agents just below this threshold had either increased effort attempting to enter the win domain or decreased effort further, to save effort cost. This leads to a bimodal distribution of effort, as observed in experimental contests [96]. Accordingly, the s-shaped prospect function, including loss aversion, leads to the emergence of a bimodal distribution of effort.[5]

Inspecting practice-effort vectors of individual agents within the practice space reveals another interesting emergent phenomenon. Among all agents of a population, those with the highest practice angles are most motivated in low competition, but are among the least motivated in high competition. In low competition (figure 3b, $c = 0.1$, leftmost panel), agents with high practice angles are forced to put in extra effort, but are still able to compete, leading to particularly high relative contributions to the societal goal. By contrast, when competition is intense (figure 3b, $c = 0.9$, rightmost panel) agents with more goal-oriented practices can no longer compete on the proxy scale and are preferentially

[5]These effects disappear with a linear prospect function (electronic supplementary material, figure S3).

discouraged, even if they have high talent. Note that the emergent practice-effort realizations cover several qualitatively distinct domains across the practice space, where observable proxy performance only partially predicts unobservable goal performance. Some agents are peak proxy performers, while only moderately contributing to the actual societal goal (figure 3b, black arrow). Simultaneously, some highly talented agents are 'losing' in proxy competition, while actually outperforming many 'winners' regarding the actual societal goal (figure 3b, grey arrow). More generally, the model predicts that for many observed proxy performances and at many time points, there is a mix of agents with highly varying degrees of proxy orientation.

Accordingly, the model captures a powerful, yet overall complex and realistic, effort incentivization effect of competition. Additionally, we observe novel but plausible emergent patterns of relative motivation changes depending on the relative goal orientation of individual agents. Note that all these economic/psychological dynamics are independent of the evolutionary mechanism and occur similarly when the latter is disabled ($p = 0$; electronic supplementary material, figure S1).

### 2.3.2. Simultaneous beneficial and detrimental signalling/screening (R1.2–R1.3)

Additional to these psychologically driven effects, competition determines selection. Each agent with insufficient proxy performance (loser) is subject to stochastic professional death with probability $p = 0.1$, where death means permanent elimination from the competitive system (indicated by white rectangles; figure 3a, e.g. white arrow). The relative ability of individual agents to compete, i.e. avoid being selected out of the system, depends on their personal parameters, namely talent ($t$) and practice orientation ($\theta$). Both can affect selection only via the proxy. Accordingly, the proxy may mediate signalling/screening for both $t$ and $\theta$.[6]

Across all levels of competition, but most prominently for high competition, we observe a beneficial signalling/screening effect of the proxy as higher talent translates to higher proxy performance. This can be seen by inspecting the effort vectors of agents with varying talent in the practice space (where talent is colour-coded, figure 3b). In particular, more talented agents (more yellow data points) are consistently at the higher end of the population distribution of proxy performances, meaning they are less likely to be selected out of the system. At the same time, however, there are fewer agents with high ($\theta$) in more competitive systems (compare leftmost and rightmost panels in figure 3b). This reflects screening for proxy orientation (i.e. lower practice angles), which is more intense in higher competition. Accordingly, our model, when viewed at the level of individual agents, suggests simultaneous signalling/screening effects for talent and proxy orientation. Note, however, that individual talent cannot be inherited, such that screening for talent works to retain talented agents in the system, but cannot unfold the same cumulative evolutionary dynamic as screening for proxy orientation.

### 2.3.3. Moderate levels of competition are optimal (R2)

We next explore the emergent system-level dynamics, specifically, the population means of proxy performance, goal performance and practice orientation (figure 4a,b). Mean proxy and goal performances at the system level show distinct competition dependent trajectories, reflecting the complex interplay of the forces described above (figure 4a).

First, note an initial phase of effort equilibration (within the first 10 steps), visible as a steep rise in both proxy and goal value. During this phase, effort adjusts to an initialization independent level,[7] while selection effects are still negligible. Subsequently, value may progressively increase (figure 4a, $c = 0.1$ to 0.6) or decrease (figure 4a, $c = 0.9$) due to feedback loops between individual effort and survival threshold. Additionally, more intense competition leads to an increasing divergence between proxy and goal value (figure 4a). One driver of this effect is the preferential discouragement of agents with high goal orientation (R1.1), implying an overall effort redistribution towards the proxy. A second potential driver is a selective removal of goal-oriented agents due to inferior proxy performance (R1.3). The degree to which the latter plays a role can be assessed by viewing the dynamics of the population mean practice angle, which is independent of effort (figure 4b). With a goal angle $\gamma$ of $45°$, the initial uniform distribution of practices between proxy and goal leads to an

---

[6]Note that the terms 'signalling' and 'screening' are used in a more general sense than usual, i.e. without assumptions about who moves first (collapsing the distinction between signalling and screening) or that an existing signal is necessarily honest (as further discussed in §§3.1 and 3.3).

[7]Models initialized with effort levels of 0, 1, 10 or 20 all converged to the same values during this initial phase

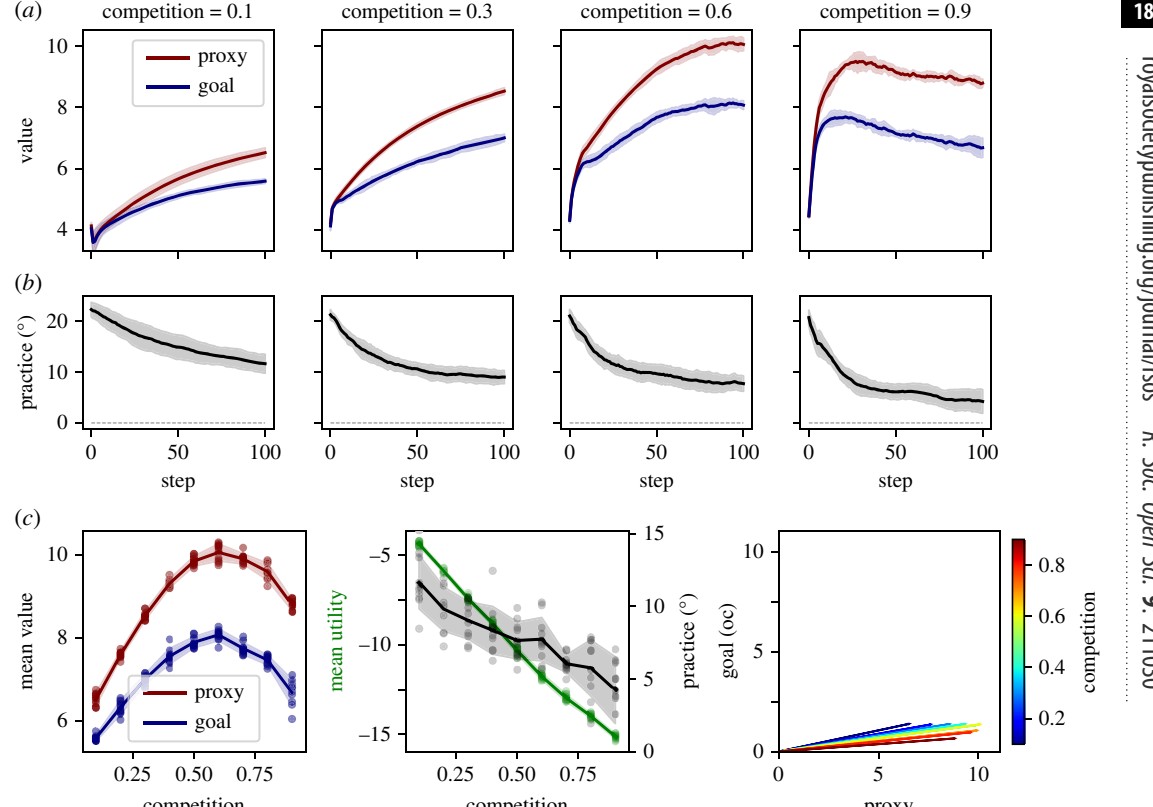

**Figure 4.** System dynamics (R2) (100 time steps), with $N = 100$, $\gamma = 45°$, $\theta_{\text{initialization}} = \mathcal{U}(0°, \gamma)$, $g = 0$, $t = \mathcal{N}(10, 1)$, $p = 0.1$, $m = 2°$. Data are collected from 10 model runs per competition level, $c$. Data for four levels of competition are shown as major columns in ($a$) and ($b$). All data are shown as population means and standard deviation over model runs. ($a$) Population dynamic of proxy and goal performance. ($b$) Population dynamic of the practice angle ($\theta$). ($c$) Outcomes plotted for the final time step (step 100), where each data point represents a run. (left) Mean proxy and goal values as a function of competition. (middle) Mean utility (green) and practice angle (grey) as a function of competition. (right) Data on mean proxy and goal values projected back into the practice space (as in figure 2$b$).

initial population mean practice of $\theta = \gamma/2 = 22.5°$. However, as selection removes agents with high practice angles, replacing them with offspring from agents with lower angles, the population mean evolves towards the proxy. As expected, the speed of evolution towards the proxy is proportional to the intensity of competition since the quantity of selection events drives population change of practices. Thus, our model reproduces a central finding by Smaldino & McElreath [1], namely the powerful corruptive force of proxy-based cultural evolution.

To obtain a more complete view of proxy and goal performances as a function of competition, we next plot the mean final values at time step 100 as a function of competition (figure 4$c$, left). For both proxy and goal value, there is an emergent optimal level of competition ($c^* \approx 0.6$). Very low competition implies low effort incentivization and low screening for talent, leading to low value creation in both proxy and goal domains. As competition increases, there is increasing effort incentivization and screening for talent. However, simultaneously proxy and goal values begin to diverge, due to the dual process of selective discouragement and cultural evolution described above. Finally, at very high competition, an increasing fraction of discouraged agents leads to decreasing mean overall effort. Note that an optimal level of competition similarly arises when selection is disabled ($p = 0$, electronic supplementary material, figure S1), albeit at a slightly higher level ($c^* \approx 0.7$). This shows that the pure economic effects of discouragement, arising from the s-shaped prospect function already leads to a hill-shaped competition-to-effort relation, consistent with empirical results [96].

Another negative effect of higher competition is lower mean utility, i.e. 'unhappier' agents (figure 4$c$, middle). Though each agent individually maximizes utility, the necessity of being relatively better than other agents pushes the system to relatively high effort levels (and effort cost), as agents continuously attempt to break out of the loss frame [97]. Since the proportion of agents in the loss frame, however, ultimately remains fixed by competition, this leads to decreasing utility. For instance, in $c = 0.9$ up to

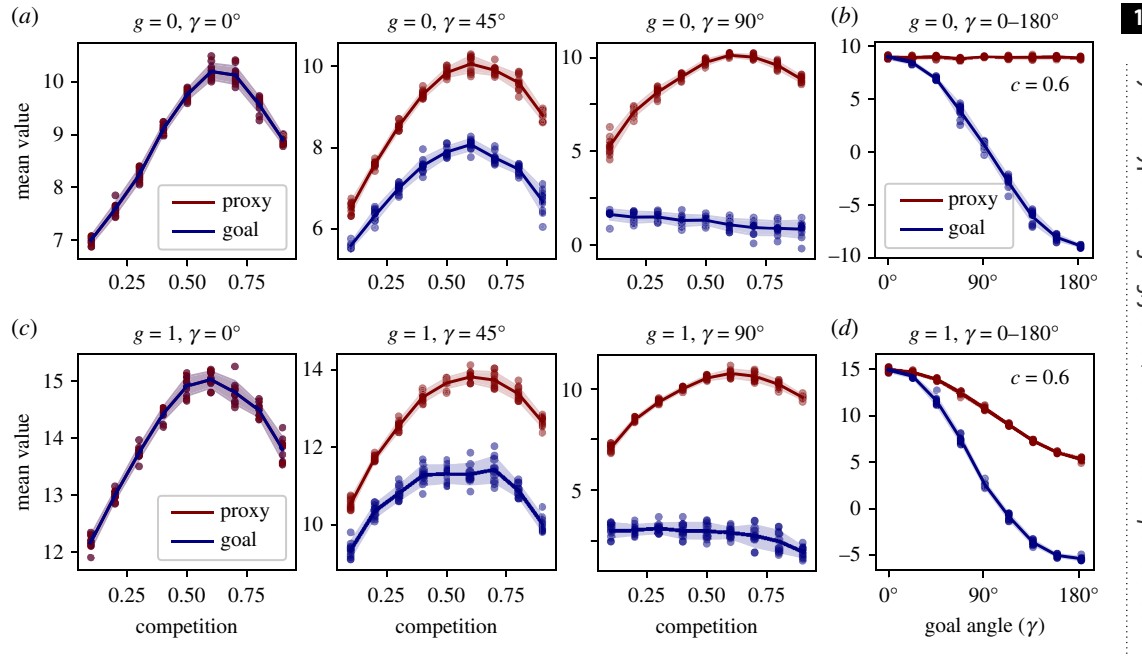

**Figure 5.** Effect of $\gamma$ and $g$ (R3), 100 time steps, $N = 100$, $\gamma = [0°, 45°, 90°]$, $\theta_{\text{initialization}} = \mathcal{U}(0°, \gamma)$, $g = [0, 1]$, $t = \mathcal{N}(10, 1)$, $p = 0.1$, $m = 2°$.). Outcomes are plotted for the final time step (step 100). (a) Mean proxy and goal values as a function of competition for $g = 0$ and $\gamma = 0°$, 45° or 90°. (b) Mean proxy and goal values as a function of $\gamma$ for $g = 0$ and $c = 0.6$ (c) Mean proxy and goal values as a function of competition for $g = 1$ and $\gamma = 0°$, 45° or 90°. (d) Mean proxy and goal values as a function of $\gamma$ for $g = 1$ and $c = 0.6$.

90% of agents may be in the loss frame, rendering the mean utility in the population negative. Note that we have not modelled positive utility from a flat wage component, since it does not affect the utility maxima of individual agents and thus system behaviour. Adding a flat-wage utility (10 or 20 utils), increased utility by a constant independent of competition, linearly shifting the utility distribution upward (not shown), but not otherwise affecting model outcomes. Also note, however, that we feel it is not necessary to assume a participation constraint in the form of a flat wage, as our agents model trained professionals and may accept substantial disutility rather than switching to a different profession, which may involve additional costly training. Instead, agents may become discouraged and cease participating due to a selection event (an endogenous participation constraint). Accordingly, low mean utility may be interpreted as either necessitating a higher flat-wage compensation or entailing a distressed agent population. Finally, we directly visualize the population mean practice-effort combinations within the practice space (figure 4c, right). Competition is colour coded, such that the effect on both mean effort (vector length) and practice orientation (vector angle) are directly visualized (compare figure 2b).

In sum, in the short term, competition beyond a certain intensity produces a system with (i) more corrupt practices, (ii) unhappier agents, and (iii) less overall effort. Additionally, the speed of corruption is faster in higher competition. Thus the system level bears out the notion of an optimal intensity of competition.

### 2.3.4. Effect of $\gamma$ and $g$ (R3)

Next, we investigate the effect of proxy fidelity ($\gamma$). As expected, the goal is best served by a proxy which perfectly reflects it (figure 5a, left; $\gamma = 0°$, $i = 1$), indicating the model works as intended. Note that the optimal level of competition ($c^* \approx 0.6$) is robust, implying it derives primarily from the increasing discouragement effect at higher competition (§2.3.1). By contrast, in extremely complex systems there may be many practices which contribute exclusively the proxy or the goal. In such systems, full proxy orientation may entail practices that, in sum, neither serve nor harm this goal (figure 5a, right; $\gamma = 90°$, $i = 0$). In this case, an observer with faith in the proxy may still infer optimal system performance at ($c^* \approx 0.6$), but the actual performance with respect to the shared goal is in fact optimal at much lower competition. The case of an imperfect, but informative proxy described above (figure 4) reflects an

intermediate case (figure 5a, middle; $\gamma = 45°$, $i \approx 0.7$). Exploring the whole range of $\gamma$ for $c = 0.6$ (figure 5b, $\gamma \in [0°, 180°]$), we observe that proxy-based competition can even substantially undermine the actual shared social goal (if $\gamma > 90°$, $i < 1$).

Next, we add a moderate intrinsic incentive ($g = 1$) towards the goal (figure 5c,d). This increases overall motivation, leading to higher performance for the shared goal for most proxy fidelities. For good proxies ($\gamma < 90°$), the goal incentive acts synergistically with the proxy, leading to similarly higher proxy performance (figure 5c). However, when proxies are bad ($\gamma > 90°$), the added incentive can actually decrease proxy performance, increasing the proxies' overall relation to the goal (figure 5d). This leads to the paradoxical situation in which an intrinsic incentive which is unequivocally beneficial to the social goal may be interpreted as detrimental based on the proxy.

In sum, both better proxies and a stronger intrinsic incentive are beneficial to the shared social goal. Furthermore, the intrinsic incentive increases the degree with which the performance of even an imperfect proxy reflects the actual goal performance.

## 2.3.5. Long-term dynamics (R4)

So far, we have explored the model in the short term (100 time steps). This timescale is most relevant to judge the dynamic behaviour of the system, i.e. it informs without presupposing that any equilibrium is ever reached. Indeed, additional mechanisms, such as delayed identification and removal of corrupted practices, or a generally continuously changing system might render these short-term dynamics dominant in determining system behaviour. However, our model also provides the opportunity to examine the long-term system behaviour without such additional mechanisms, i.e. resulting only from effort choice dynamics and cultural evolution (figure 6, 10 000 time steps). In a previous similar study, practices were found to evolve to full corruption [1]. Our model allows to test if an intrinsic incentive towards a shared goal has the power to bound this detrimental evolution.

To test this, we return to our initial model with an imperfect but informative proxy ($\gamma = 45°$), but set the intrinsic incentive to a moderate level ($g = 1$). We find that, in the long term, this intrinsic incentive constrains the evolution towards fully proxy-oriented practices (figure 6d). Specifically, an equilibrium $\theta$ is reached, beyond which the average practice no longer becomes more corrupted. Notably, this equilibrium level of corruption was similar for all levels of competition, differing only in the speed with which it was reached (figure 6d). At equilibrium corruption, the optimal level of competition was shifted to lower intensities ($c \approx 0.2$, figure 6e, left). Note that this emergent long-term equilibrium resembles a *lock-in* state. Specifically, the system orientation remains stable although individual agents value the societal goal and know their comparatively low contributions towards this goal. Indeed, the bounding occurs because agents who mutate towards even lower $\theta$ are, on average, insufficiently motivated to survive competition. In other words, their individually known low contribution to the shared social goal provides insufficient incentive to muster competitive effort levels. Note that this happens even though their relative effort cost to compete (i.e. to produce the proxy) is lower than for more goal-oriented agents. Finally, as in the short term, more competition leads to a larger fraction of agents in the loss frame, i.e. lower average utility at the population level (figure 6e, middle).

In sum, in the long term, competition beyond a certain intensity produces an equilibrium with (i) similar practices, (ii) unhappier agents, and (iii) less overall effort. Thus our model demonstrates that an intrinsic motivation to work towards the societal goal can bound the evolution towards fully proxy-oriented practices, giving rise to the paradoxical situation we have termed *lock-in*.

## 2.3.6. Sensitivity/robustness analysis

Next, we tested the sensitivity of these results to parameter variations. Since a full combinatorial exploration of the parameter space is infeasible, we followed a three-stage strategy: we first systematically varied one parameter at a time holding the other parameters constant (figure 7; base value in boldface), monitoring the effect on equilibrium $\theta$. We found that parameters fell into three main groups: (i) equilibrium determining parameters (figure 7a; $\gamma$, $g$), (ii) dynamics determining parameters (figure 7c,d; $c$, $p$, $m$), and (iii) parameters affecting predominantly system variability (figure 7b; $t_{sd}$, $N$). We then attempted to find deviations from this initial classification by repeating one-at-a-time analysis with a range of alternative base values (table 2, base values in italics). In over 100 additional model runs, we failed to find such deviations, suggesting our mapping of the main parameter effects was robust.

Across all model specifications equilibrium corruption was determined predominantly by the informational quality of the proxy ($\gamma$) and the psychological power of the intrinsic incentive towards

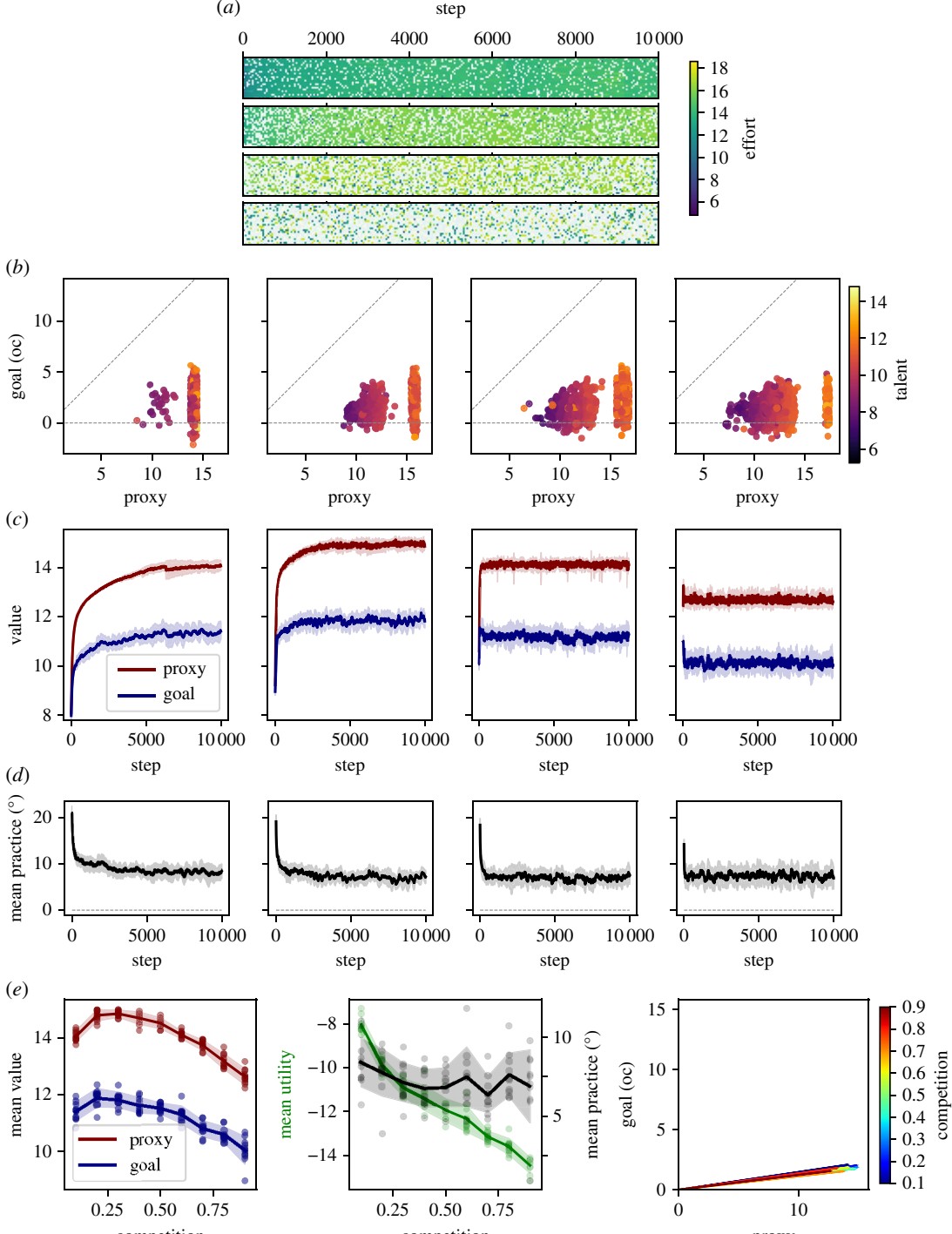

**Figure 6.** Long-term dynamics, (10 000 time steps), with $N = 100$, $\gamma = 45°$, $\theta_{\text{initialization}} = \mathcal{U}(0°, \gamma)$, $g = 1$, $t = \mathcal{N}(10, 1)$, $p = 0.1$, $m = 2°$. For panel descriptions please refer to figures 3 and 4. In (a), only a subset of agents and every 10th time step is depicted.

the societal goal ($g$; figure 7a). In the main model (i.e. without practice agency, see below), the dynamics of corruption was predominantly determined by the level of competition ($c$) and the degree to which competition was realized in actual selection events ($p$; figure 7c,d). Practice mutation rate ($m$) had no notable effect on equilibrium or dynamics of corruption for these standard models (figure 7d,f, top), but did contribute to the dynamics of corruption when the equilibrium practice was outside the initialization range (figure 7d,f, bottom). In this additional sensitivity analysis, we initialized all practices as completely goal-oriented ($\theta_{\text{initialization}} = \gamma$), and find that practice mutation amplitude ($m$) then increases the speed of corruption. By contrast, the number of agents in competition ($N$) and talent spread ($t_{\text{sd}}$) affected mainly system variability (figure 7b).

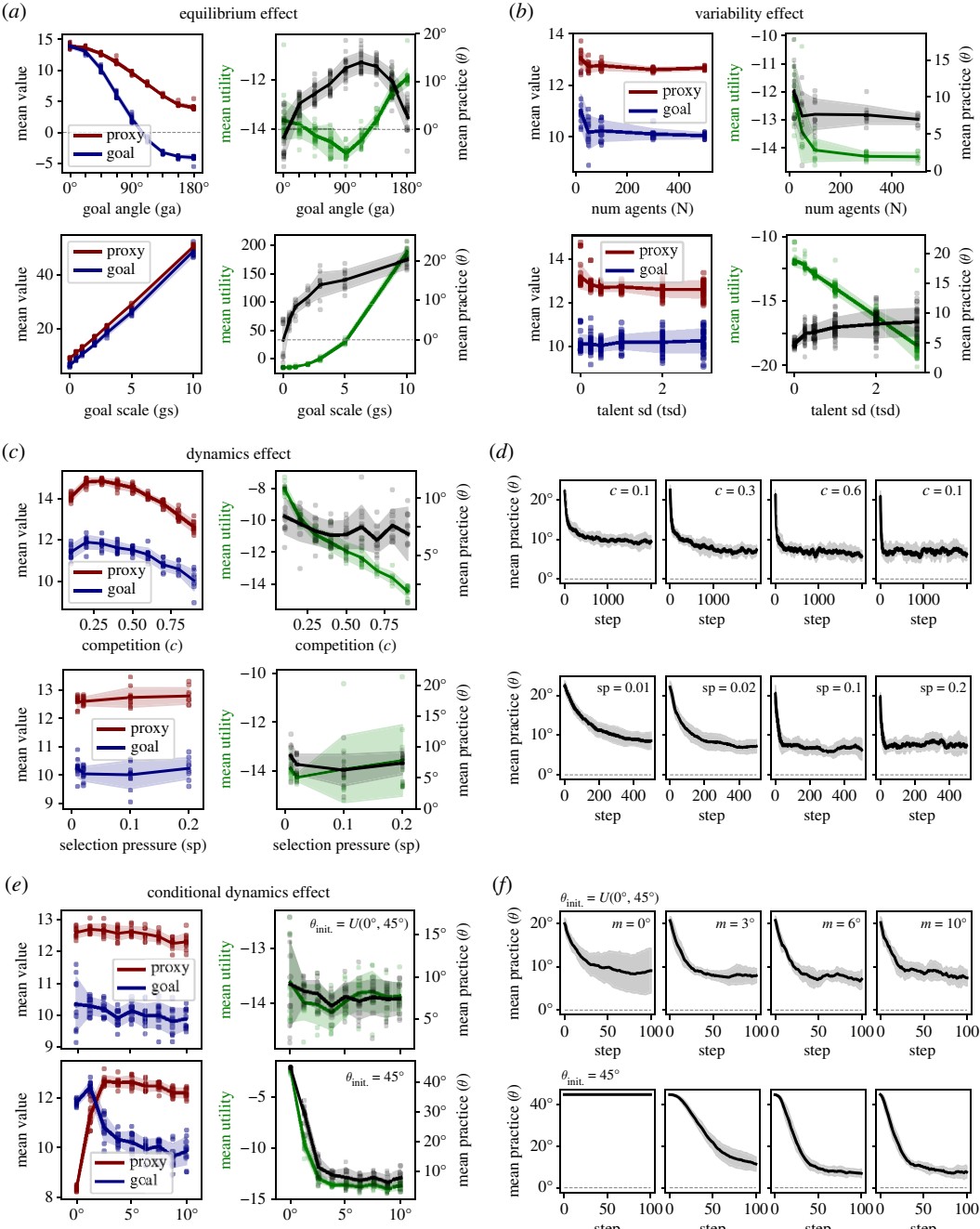

**Figure 7.** Sensitivity analysis, parameter base values: $N = 100$, $\gamma = 45°$, $\theta_{\text{init.}} = \mathcal{U}(0°, \gamma)$, $g = 1$, $t = \mathcal{N}(10, 1)$, $p = 0.1$, $m = 2°$. One parameter at a time was varied while leaving the other parameters at their base values. Overview plots (a–c,e) show mean model outcomes at equilibrium/$\theta$ (step numbers were increased until convergence to equilibrium was confirmed for each parameter). Data points represent mean population outcomes of individual runs and lines represent mean ± s.d. over runs. (a) Parameters controlling equilibrium corruption: goal angle ($\gamma$), top; goal scale ($g$), bottom. (b) Parameters controlling system variability: number of agents ($N$), top; talent standard deviation ($t_sd$), bottom. (c,d) Parameters controlling the speed of convergence: competition ($c$), top; selection pressure ($p$), bottom. ($p = 1$ produced the same equilibrium, but is not shown because the dynamics could no longer be resolved.) (e,f) Parameter controlling system dynamics if equilibrium $\theta$ is outside the initialization range. For our standard practice initialization range ($\theta_{\text{init}} = U(0°, 45°)$) $m$ had no notable effect (top). If all practices were initialized at $\theta_{\text{init}} = \gamma = 45°$, $m$ governed the dynamics of convergence, and in the case of $m = 0°$ precluded convergence (bottom).

Thus, our sensitivity analysis confirms that equilibrium corruption is determined primarily by proxy fidelity ($\gamma$) and the intrinsic incentive towards the social goal ($g$). Additionally, the pressure towards this equilibrium corruption is governed by the intensity of competition ($c, p$) and potentially the practice mutation rate ($m$).

Next, we tested the robustness of these main findings for two major model modifications, a step-prospect function and practice agency. In a first additional set of models, we replaced the prospect function by a simple step function (prospect = −1 for losers, 1 for winners; figure 8a–c). While overall effort and value creation was lower (given the lower maximal prospect differential), the respective effects of the parameters remained robust. This is important, given that the Kahnemann/Tversky type prospect, considered above, may apply to individual agents, but may not hold when larger entities, such as laboratories or companies, are considered as agents. In a second additional set of models, we introduced agency over the practice angle by letting agents choose not only effort, but also practice orientation, to maximize utility at every time step. Remarkably, the main determinants of equilibrium corruption, i.e. $\gamma$ and $g$, again followed the same pattern (figure 8d,e). However, introducing practice agency led to a dramatically increased speed of convergence and decreased variability of practices within a population. Indeed, our specification of practice agency was so potent, that it effectively overrode the cultural evolution mechanism and related parameters. Interestingly, it also markedly increased the ability of $g$ to counteract corruption, leading to approximately twofold higher equilibrium $\theta$ (compare figures 7a and 8d). Thus our main finding, that there is an equilibrium corruption level which is determined by $\gamma$ and $g$, was highly robust, while the dynamics determining effects of $c$, $p$ and $m$ become less relevant when agents continuously choose their practice orientation to maximize utility. These results further suggest that increased agency over practices may help counteract corruption given sufficient intrinsic goal valuation ($g$).

Next, we tested the dependence of the main model results on an alternative selection mechanism, namely fitness-proportionate selection (electronic supplementary material, figure S2). Briefly, winning agents are selected with a probability proportional to their fitness, i.e. proxy performance, replacing a randomly chosen agent. As in the previous simulations, the frequency of such selection events is proportional to competition, allowing to see the effect of competitive intensity. All main model results (R1–R4) remain robust (compare figures 3 to 6).

Finally, we combined fitness-proportionate selection with a linear prospect function (electronic supplementary material, figure S3). As expected, this removed the meso-level phenomena of effort bifurcation and hill-shaped effort to competitivity relation (R1–R3). Higher competition still increased the speed of corruption, also as expected. We probed if the main model result, namely the ability of an intrinsic economic incentive to constrain the evolutionary pressure towards full corruption (R4), was robust within the fitness-proportionate selection model with linear prospect function (electronic supplementary material, figure S4). Indeed, the evolution towards full proxy orientation was still constrained and the equilibrium corruption level was comparable to that seen in the original implementation (figure 6).

## 2.4. Model discussion

We have presented an agent-based computational model, sketching central components and mechanisms of proxy-based competition. Specifically, we defined a population of agents which are both incentivized, and are subject to selection, based on some proxy measure. We used an economic multitasking model to define the proxy measure as an imperfect approximation of a shared social goal, which agents may value independently of the proxy. Crucially, agent practices can vary with respect to their orientation between proxy and goal, and are determined via a process of cultural evolution driven by proxy-based selection.

The resulting model displays realistic effort choice patterns and, to our knowledge for the first time, captures the simultaneous screening for talent as well as proxy orientation. It further displays complex patterns of heterogeneous agent behaviour involving simultaneously existing sub-populations of relatively proxy-oriented and goal-oriented agents each of which may contain discouraged as well as motivated agents (R1). At the system level, this leads to the robust emergence of an optimal level of competition, which is moderate and subject to slow dynamic change (R2). Parameter analyses reveal that the primary determinants of how well the shared social goal is served are (i) the quality of the proxy and (ii) the strength of the intrinsic incentive towards the shared goal (R3). Finally, in the long term, the intrinsic incentive can constrain a progressive evolution towards fully proxy-oriented practices (R4). In general, increasing competition beyond the optimal level led to (i) worse or similar practices, (ii) unhappier agents, and (iii) less overall effort (R2 and R4).

In the following, we will first discuss our adaptation of the economic multitasking model highlighting the similarities and differences to previous studies (§2.4.1). Next, we will discuss the iterative effort choice algorithm (§2.4.2), and the cultural evolution mechanism (§2.4.3). Finally, we discuss the implications of the model for mitigation strategies (§2.4.4), and draw conclusions from the agent-based model (§2.5).

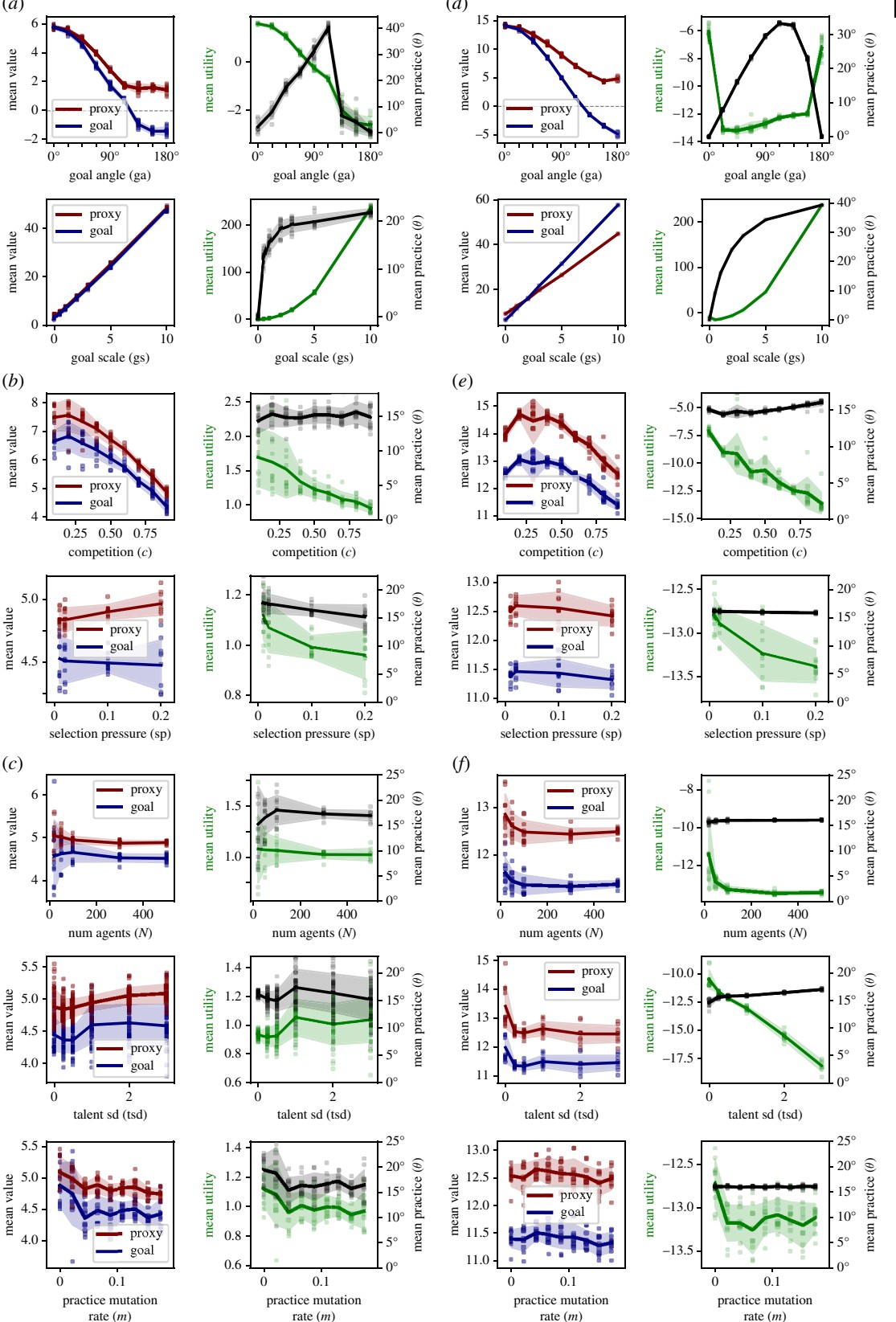

**Figure 8.** Robustness, parameter base values: $N = 100$, $\gamma = 45°$, $\theta_{\text{init.}} = \mathcal{U}(0°, \gamma)$, $g = 1$, $t = \mathcal{N}(10, 1)$, $p = 0.1$, $m = 2°$. One parameter at a time sensitivity analyses was repeated for two alternative models: One containing a step-function prospect $[-1, 1]$ (a–c) and one containing agency over the practice angle (d–f).

### 2.4.1. Multitasking and moral hazard

A rich economic literature has addressed the problem of incentive misalignment, specifically the question of how to formulate optimal incentive contracts given imperfect proxies, rational agents, and standard equilibrium assumptions [13,32,70–72,98]. The central result of this literature is that worse proxies lead to a lower optimal slope of a linear incentive contract, or more generally, a weaker optimal incentive strength.

This is consistent with the present findings, where higher competition entails faster corruption and intermediate levels of competition are generally optimal. However, note that the underlying mechanisms differ substantially (see e.g. §2.4.2) and that the actual equilibria were affected by both the evolutionary and the economic mechanism. In the following, we will first highlight some differences between the present model and previous multitasking models regarding operationalization and then concerning results. Some additional conceptual differences to traditional economic models, such as signalling models, are discussed in §§3.1 and 3.3.

In terms of set-up, the present model borrows from the economic multitasking literature by assuming two task dimensions (proxy and goal), an interest congruence parameter (the goal angle), and utility-maximizing agents with quadratic effort cost [13,32,70,72]. While direct experimental justification for these operational choices is still scarce [78], they represent an elegant and parsimonious way to capture the effects of both intrinsic and extrinsic incentives on effort. Beside this commonality in operationalization, there are a range of general differences resulting from the contrast between analytic equilibrium models and computational agent-based models [28,77]. Briefly, at the cost of mathematical elegance, agent-based models relax some assumptions about 'rational agents' and 'standard-economic equilibrium' and allow to investigate heterogeneous agent behaviour and system dynamics. The agents in the present model have to individually compute their effort choice, given only the imperfect information and limited computational power available to them. Furthermore, they make these choices not synchronously, but stochastically and sequentially. Finally, agents are allowed to be heterogeneous in terms of their traits and choices. This gives rise to the interesting individual-level and system-level dynamics and distributions, detailed in §§2.3.1 and 2.3.2.

As a result, the present model, to our knowledge for the first time, allowed to simultaneously capture the signalling/screening effect on talent but also proxy orientation. As noted above, we use the terms signalling and screening here somewhat loosely, since we make no assumption about who moves first (collapsing signalling and screening), or that signals must be necessarily 'informative' or 'honest', ([47,99], discussed further in §§3.1 and 3.3). Instead, we simply observe the simultaneous signalling/screening for talent and proxy orientation. The model suggests that observed high performers are likely to generally represent a mix of true high performers (on the goal scale) and highly proxy-oriented agents, i.e. deservedly and fraudulently successful agents, respectively. Simultaneously, there exist groups of agents which are deservedly or undeservedly losing in proxy-based competition (according to their goal performance). Notably, this occurred simply due to chance, even to agents with identical practices and talent. While the simultaneous existence of such diverse groups is intuitive, it is important to note that they were to the best of our knowledge not captured in previous models. Furthermore, they contradict simplistic accounts of meritocracy and kleptocracy, suggesting both narratives are probably partially correct in most proxy-based competitions.

A particularly interesting set of studies in this context are investigations of contests with the possibility of sabotage [93,100]. A mixed strategy containing productive effort as well as sabotage is analogous to a low practice angle in our model. Indeed, increased competition (a smaller fraction of winners or increased prize spread) led to a redistribution of effort towards sabotage, consistent with our results. Notably, studies addressing contest incentives in explicit multitask settings appear to be lacking and thus remain an important avenue for future research [78].

In the following two subsections, we will discuss additional emergent behaviours at the individual and the system level, resulting from our iterative effort choice algorithm (§2.4.2) as well as the dynamic interaction of these mechanisms with cultural evolution (§2.4.3).

### 2.4.2. Agent-based contest algorithm

The relation of our model to the economic contest literature (see [78] for an overview) deserves particular mention, since to the best of our knowledge, we are the first to implement a dynamic effort choice heuristic based on cumulative prospect theory. It thus adds to this literature by providing a mechanistic account of how equilibrium effort is approached (see also [101]) and recreates a remarkable range of empirical

patterns. Our simple prospect-based contest heuristic is neuroeconomically plausible in that it contains (i) an empirically validated representation of utility, (ii) multiple sources of input to this utility, and (iii) reference dependence of this utility [89]. The emergent behaviour reproduced many findings from the experimental contest literature, the origins of which are still poorly understood [78,96]. Note that these findings were independent of the evolutionary mechanism, but depend on the s-shaped prospect function (electronic supplementary material, figures S1 and S3, respectively).

Firstly, effort incentivization is usually optimal at intermediate levels of competition, consistent with experimental and field observations [93–95]. Secondly, agents performing far below the survival threshold display a 'discouragement effect' [96]. While traditionally discouragement is thought to result from agent heterogeneity, our model predicts it even in cases without agent heterogeneity, simply due to chance and the emergent distribution of proxy performances. The discouragement effect leads to a bimodal distribution of effort, again a well documented, but incompletely understood, empirical phenomenon [96].

Finally, our iterative effort choice algorithm makes predictions about the temporal evolution of effort expenditure, particularly its path of convergence to potential equilibria. In this, it resembles previous experimental contest models based on reinforcement learning [101] or belief-updating [102]. Another specific prediction is variability of individual effort over time and over the population, which is proportional to the level of competition (qualitatively matching data presented in [93,103]).

Accordingly, we specified what is to our knowledge the first agent-based iterative effort choice algorithm. This generative agent-based algorithm reproduced a remarkable range of effort choice patterns observed in the experimental contest literature, which were not considered during model design [78,96]. Embedding this algorithm within a multitasking framework further allowed us to intuitively model the complex screening effect of competition on observed (proxy) as well as unobserved (goal) performance (see §2.4.1).

### 2.4.3. The evolution towards bad practices

A central question we asked of our model, was whether the individual-level decision model could alter a simultaneously acting system-level mechanism of cultural evolution. A previous agent-based evolutionary model of cultural practices had reported robust evolution towards the worst possible practices [1]. The central unifying element with the present model is the existence of a *practice variable* ($\theta$ in the present study), which alters the relation between true (goal) and selected (proxy) performance, formalizing the assumption that there exist practices which contribute differentially to proxy and goal. Furthermore, this variable can be imperfectly inherited, formalizing the assumption that professional practices are complex and transmitted from senior to junior professionals imperfectly. In our view, these assumptions are nearly self-evident, and the resulting detrimental evolution becomes a powerful prediction. However, even the authors note that this prediction seems overly pessimistic [1].

Here, we have modelled an intrinsic incentive and observe that it is able to bound the evolution to fully proxy-oriented practices. Specifically, our model led to an equilibrium corruption level which was substantial, but still markedly different from full proxy orientation. This result is intuitively appealing and, in fact, better matches the results of a meta-analysis of sample sizes within the same study [1]. The result was also highly robust to varying parameters (figure 7), changing implementation details of the prospect function (figure 8) or changing the evolutionary selection algorithm (electronic supplementary material, figures S2–S4). The mean proxy orientation at equilibrium was determined primarily by the informational quality of the proxy ($\gamma$) and the relative psychological strength of an intrinsic incentive towards the societal goal ($g$). How both parameters may be interpreted and altered in practice will be further discussed below (§2.4.4). Notably, the emergent equilibrium displays several characteristics of what we have termed *lock-in*. Namely, the system remains stable, even though all individual agents are aware of its imperfect service of the goal. Indeed, the bounding of corruption occurs precisely at the point where the resulting disincentivization of agents becomes so large that they are no longer sufficiently motivated to compete. Note that this aggregate picture does not preclude the existence of complex distributions of individual agents, where some may be genuine free-riders but others manage to reconcile competitive pressure with the service of the goal (e.g. figures 3*b* and 6*b*). Yet it predicts that, at equilibrium, substantial portions of the population may feel that competitive pressures are undermining their ability to pursue the shared social goal, as suggested by empirical findings [4,26].

Notably, our robustness analyses suggest that increased agency over the practice orientation can further decrease the level of corruption. This is interesting because the assumed level of agency over practices is a principal difference between traditional economic and cultural evolutionary approaches. In the absence of practice agency, the dynamics of corruption in our model were governed by competition ($c$ and $p$) and potentially complexity ($m$). Adding practice agency let the model converge to equilibrium nearly instantaneously, effectively occluding the dynamic effect of $c$, $p$ and $m$. However, in our implementation, practice agency is substantial (with a similar speed and range as effort agency). In real systems practices may be determined by a mix of social processes and individual choice, suggesting outcomes may lie between the models with and without practice agency. The interaction between the evolutionary process and an active practice choice could be investigated in more detail by, for instance, probing models where practice agency is allowed only stochastically. Indeed, exploratory analysis in such a model suggests that, in such cases, the level of competition can mediate between the equilibria with and without practice agency.

Finally, numerous additional social and individual factors could be addressed within our modelling framework. For instance, the network structure between agents and the precise formulation of information transmission between them. In our model, all agents observe the noisy proxy performances of all other agents, implying full network connectivity. Though this may seem unrealistic, the idea that professionals are generally aware of their approximate competitive standing seems justified.[8] Future models could further explore system dynamics if practice angles are influenced by social forces such as the formation of cultural norms [104]. While, cultural-norm transmission is implied in the hereditary transmission of practices modelled here, they may also be determined more directly through neighbouring agents (see e.g. [105]). This is likely to introduce additional nonlinear effects in space and time, as locally normative practices emerge or collapse. There are of course numerous additional individual-level factors, such as potentially differing talent for proxy- and goal-oriented practices between agents, which could be investigated. While our model thus addresses an, in our opinion, crucial force in proxy-based competition, namely cultural evolution, the scope for additional exploration of social mechanisms and inter-individual variability is substantial.

### 2.4.4. Implications for mitigating proxy-induced corruption

Our model suggests several principal ways to improve system outcomes, namely to increase proxy fidelity ($\gamma$), to adjust competitivity ($c$), or to attempt fostering the intrinsic motivation towards the social goal ($g$). As we will see, these map readily to existing mitigation strategies, organizing them within a general framework. We will now briefly discuss each in turn.

The most robust way to improve system outcomes is by improving the proxy (decreasing $\gamma$). In practice, this can for instance be done by identifying particular weak spots of a proxy and specifically addressing them (see §4.7). It may be done directly, by modifying the processes by which competitive proxies emerge (e.g. improving peer review or statistical practices), or indirectly by penalizing specific identified corrupt practices. Improving proxies is the most intuitive and most widely used mitigation strategy, and should be pursued wherever possible.

It is, however, important to reiterate that *perfect* proxies are often impossible in principle, e.g. when our goals encompass the future. Furthermore, they will generally not be cost-effective (further discussed in §3.2). Within our model, $\gamma$ will thus generally be some positive value. Indeed, we posit that more complex systems will generally have more corruptible proxies (i.e. larger $\gamma$). The system-theoretic reason is that the number of 'failure modes' increases dramatically with system complexity [106]. Intuitively, there will be more possibilities to 'game the system' in more complex contexts. Firstly, there may simply be more opportunities, i.e. possible practices, overall. Secondly, the relative effects of any act or practice on proxy and goal are far more likely to behave idiosyncratically. This may be because (i) more complex systems will generally feature more nonlinear interactions and (ii) possible action to outcome mappings will undergo a combinatorial explosion with increasing complexity [106]. That said, social or socio-ecological systems are highly complex, and thus $\gamma$ may often be closer to 90° than 0°.

While it is thus valuable to continuously consider how individual proxies may be improved, it will generally also be necessary to map instances where they cannot be improved, and other approaches have to be taken to ameliorate the adverse consequences. The first such approach is similarly suggested by the

---

[8]To verify robustness against the full connectivity assumption, we probed whether restricting the sampled proxy performances to 9 or 22 neighbouring agents altered outcomes. This had no effects other than increasing variability (not shown).

present model, namely to reduce the competitive pressure. This will decrease the speed of corruption, potentially providing more time for measures to improve proxy fidelity in a dynamic setting. Moreover, in systems where proxy fidelity cannot be improved, very low levels of competition may simply be optimal with respect to the shared social goal (even when proxy measures suggest otherwise, see e.g. figure 5). Additionally, our model suggests that lower competition will allow substantially higher levels of average individual utility, or 'happiness', for the competitors within a system.

The final general approach to improve outcomes suggested by our model is to increase the intrinsic incentive towards a shared purpose ($g$). Calls to this end have become more prominent in recent times, for instance in the context of business [107] or science [108]. The strength of this incentive can be seen as a product of (i) goal valuation and (ii) individual knowledge about the consequences of one's own actions for this goal. Part of the solution could thus be education or awareness campaigns. More generally, we would argue that this result supports the recurring call to re-emphasize training in the humanities, nurturing individuals who consider value beyond quantitative or competitive proxy measures [109,110]. Overall, the study of how intrinsic motivations are shaped at both the individual and social level is a burgeoning field of research, upon which we can only touch here (e.g. [42,100], §4.2). While such intrinsic motivations may be fickle, and vary substantially between individuals, the empirical evidence suggests that they are systematically underestimated by both economic and cultural evolutionary research traditions.

Overall, our model thus supports the diverse previous proposals, and indeed ongoing efforts, throughout various societal systems. Unfortunately, none of these is a silver bullet. The present theory may help to further inform in which situations which type of proposal is most promising, and help to organize them into a coherent framework. It also illuminates the fundamental information theoretic reasons why a single-minded focus on any proxy or proxy system is dangerous.

## 2.5. Model conclusion

In sum, our agent-based simulation combines two of the central mechanisms, expected to arise in proxy-based competitive societal systems, namely an individual-level decision mechanism and a system-level cultural evolution mechanism. Cultural evolution is based on an imperfect proxy of the actual goal of the competitive system. Individual decisions are determined by both an intrinsic incentive towards this goal as well as a proxy-based competitive incentive.

The model, to our knowledge for the first time, captures simultaneous screening for talent and proxy-oriented practices, producing complex patterns of motivated and discouraged agents with variably corrupted practices. These patterns arose through the complex feedback loops between the individual decision mechanism and cultural evolution and are amenable to further experimental verification.

The model suggests that outcomes are typically optimal at moderate levels of competition. This holds in the short term, but particularly in the long term or when proxy measures are poor. Outcomes for the shared goal can be improved by (i) increasing the information captured in the proxy, or (ii) promoting the strength of the intrinsic incentive towards the societal goal. A principal way to decrease the corruptive pressure in a dynamic setting is to decrease the intensity of competition.

Finally, the model suggests that the intrinsic incentive towards the goal can constrain a progressive cultural evolution, which would otherwise enforce fully proxy-oriented practices. The emergent equilibrium resembles a *lock-in* state, where competitive pressures undermine the agents' ability to serve the shared social goal. In *lock-in*, competition beyond the optimal level leads to (i) similar practices, (ii) unhappier agents, and (iii) less overall effort.

# 3. General discussion

The combination of an economic and a cultural evolutionary mechanism in our model thus captures several key positive and negative dynamics, which should be expected in proxy-based competition. It demonstrates how a focus on proxies may help to integrate different disciplinary methodologies and concepts. The underlying information theoretic and system-theoretic reasoning has already been briefly introduced above (§1.2). The next three sections will place the model and underlying theory more firmly into a framework of *optimization in complex adaptive systems*. First, §3.1 will develop some important system theoretic insights relevant for proxy-based competition, namely (i) the role of the proxy in fostering stable social institutions and (ii) the consequential difference between proxies in

corporations versus whole societal systems. Second, §3.2 will draw attention to the striking similarities between the concepts presently discussed and the literature addressing optimization in artificial intelligence or machine learning, suggesting there are truly fundamental informational mechanisms at work. Third, §3.3 will (i) more closely consider the relation to economic and biological signalling theories and runaway selection and (ii) develop the novel notion of directed cultural evolution.

## 3.1. Complex systems perspective

The overarching scientific framework of the present theory and model is that of *complex adaptive systems* (CAS, see e.g. [111] for an excellent introduction, or [28] for a recent evolutionary economic treatment). Complex adaptive systems theory is perhaps most briefly characterized as the study of how structure emerges from the dynamic interactions of smaller components, for instance, how interacting molecules give rise to cells or how human behaviour gives rise to institutions. Competitive societal systems represent just such institutions, which maintain their properties due to the myriad of actions of the individuals (and things) composing them. The proxy can be understood as an emergent property of such institutions [112]. In the following, we will point towards two insights from CAS theory that we believe are essential for the understanding of proxy-based competitive societal systems, namely (i) the role of the proxy in system stability and (ii) the natural constraints on minimal proxy fidelity. We will treat each in turn.

First, we would argue that the proxy typically plays an essential role in the process of auto-catalysis (i.e. self-reproduction) of competitive societal systems. As mentioned above, the question how structure can emerge and self-reproduce at larger scales is foundational in complex systems theory. The proxy probably plays a crucial role in this process because the criteria of competition are typically determined by 'winners' of the respective competition. What constitutes good science is determined in large part by successful scientists. In other words, the allocation of power to define the proxy (i.e. the criterion of competitive success) is itself strongly impacted by the proxy. A more provocative way to phrase this, would be that competitive societal systems can be viewed as self-reproducing power structures [113]. In this view, the *societal goal* may be viewed primarily as a *public justification* of the system, aiding in auto-catalysis, but serving no other societal purpose. While the implications of assuming, or not assuming, a societal goal are further discussed in §§3.3–4.5, we here want to emphasize that a function as *public justification* does not exclude a function as *societal goal*. Indeed, we would argue that the *public justification* should be simply equated with the societal goal. More importantly, the auto-catalysis of institutions is a necessary requirement for their existence. Accordingly, it might be argued that any contribution towards an abstract societal goal requires the self-reproducing power structures engendered in proxy measures. The degree to which this societal goal then becomes an actual guiding force for the institutions, rather than a justifying façade, depends on the degree to which the problems motivating the present paper are taken seriously.

Second, an important factor determining the minimal informativity of a proxy is the distinction between systems that are, or are not, subject to higher-level selection (CAS1 and CAS2, respectively, [114]). We posit that competitive societal systems are generally not subject to higher-level selection, i.e. they are CAS2, and this implies proxies cannot be *a priori* assumed to be minimally informative. This dramatically increases the societal relevance of the present theory compared with previous economic or evolutionary treatments of CAS1. Within these, any proxy that exists is assumed to be minimally informative (e.g. [32,47,57]). While the reasoning underlying this assumption is not typically explicitly discussed, it seems to rely on higher-level selection. CAS1 systems, such as corporations, can be thought to 'adapt' to whatever mediates this higher-level selection, e.g. profitability or firm value [32,70]. This can then be viewed as the 'organizational goal' of the corporation. Importantly, higher-level selection will act to optimize this organizational goal without any need to consider decision mechanisms within the corporation. In other words, the market constrains the ability of corporations to deviate from their organizational goal, as unprofitable firms will eventually be eliminated. Proxies, which are not informative would undermine the organizational goal (by misdirecting resources). It is reasonable to think that they would be eliminated from the system by the higher-level selection, along with the organizations employing them.

Indeed, the view that any practices within an *existing* firm are thus by definition 'optimal' is implicitly borne out by the traditional economic literature (e.g. [32,70]). Similarly, economic signalling theories focus on signals which reduce information asymmetries, again implicitly relying on the assumption that only these would survive the market [47]. For instance, in the seminal paper by Spence [99], the 'critical assumption' that the cost of a signal must be negatively correlated with productivity/talent,

since otherwise they would not be persistently informative, is quite explicitly posited as the basis of the proposed model. Why only 'informative' signals are worthy of study is not further discussed, but presumably derives from the foundational assumption that 'non-informative' signals would be eliminated by market mechanisms. Interestingly, this renders the study of Goodhart's Law within companies (CAS1) a topic which is important for management scholars, but of limited relevance to society at large: positing the market as revealing a ground truth, which is automatically enforced via market selection, guarantees that *existing* firms employ only informative proxies and serve their organizational goals well.

By contrast, we posit that competitive societal systems, such as the market or academia, are attempts to translate an abstract societal goal into a selection mechanism. In our view, the assumption that this constructed selection mechanism automatically represents a ground truth, is simply not warranted [75]. Because the whole societal system is not selected, proxy-based competition is better understood as CAS2, constrained only by our abstract goals. We have captured such an emergent constraint within our model: the goal affected outcomes only via the decisions of individual agents and no actions of an assumed 'principal' were required.

Finally, note that there is no reason to believe the lack of higher-level selection renders a Goodhart mechanism within the system any less relevant. This may appear so, because the goal of a competitive societal system will generally be more abstract and difficult to define than a goal of an organization, and thus any Goodhart phenomenon will be similarly difficult to conceptualize (see §4.1). However, the resulting difficulty to detect and describe Goodhart phenomena probably renders them *more powerful and prevalent* rather than less (see e.g. §4.2). Notably, a set of efficient lower-level CAS1 could be the selected entities in a corrupted higher-level CAS2 (§4.6 further explores the ideas of nested and interlocking proxy-based competitions).

Thus, CAS theory suggests that (i) proxies *per se* may be an integral part of maintaining competitive societal systems but also (ii) the study of proxy corruption is probably far more relevant in societal systems as opposed to corporations. While any corruptive effect of proxies will be naturally constrained in a corporation (or in natural evolution, see §3.3), there is no such natural constraint for societal systems. Emergent proxies may thus be expected to be the default determinants of a societal system's evolution, rendering the question to which degree abstract social goals can shape these proxies a pressing social and philosophical issue (see §4.1).

## 3.2. Optimization perspective

Closely related to the complex adaptive systems perspective is the optimization perspective, most extensively explored in a recent literature on machine learning and artificial intelligence (for brevity AI) [28,36,45,106,115,116]. These literatures study how complex entities or systems can be controlled (or categorized), generally via an optimization process. Specifically, designers of AI systems typically have to create some *objective function*, which defines the goal of the AI in concrete code. To serve the goal, the AI then chooses whichever options maximize this objective function. In other words, the AI will transform complex inputs into a scalar measure, which serves as a proxy for an abstract goal.

*The proxy* as defined here (§1.2.1) is analogous to this scalar outcome of an objective function. The competitive societal system can be conceptualized as an information processing device, analogous to the AI system. Competitive pressure corresponds to optimization pressure. These analogies are extremely helpful, because the AI literature is, for the first time, beginning to systematically dissect the statistical and informational reasons for the occurrence of Goodhart's law in proxy-based optimization [36,45,69]. More generally, we suggest that all the principal types of challenges that occur during AI optimization also apply in competitive societal systems. To illustrate this point, consider the three canonical AI-safety problems of (i) reward hacking, (ii) negative side effects, and (iii) scalable oversight [116].

— Reward hacking describes when an AI 'hacks' its own objective function, for instance by cheating in a game or choosing other strategies which undermine the programmers original intention. It can be understood as an instance of Goodhart's Law, e.g. when individuals game a competitive proxy, thereby undermining the original goal of a social system. It is what we have considered in our model.
— Negative side effects describe undesirable outcomes which do not directly relate to the intended goal of the AI. In competitive societal systems this may include negative environmental or social side effects (e.g. [5]). Within our model, this could be thought of as additional orthogonal dimensions to our practice space.

— Scalable oversight describes the process of balancing the cost of creating the proxy with the benefits from better proxy information. One main reason why proxy measures will imperfectly reflect the societal goal is because they need to be cost-efficient. For instance, the information contained in an impact factor could be arguably arbitrarily increased by adding more reviewers or having experiments replicated. Clearly, the cost of improving (or monitoring) the proxy needs to be balanced with the potential detrimental effects of an imperfect proxy.[9]

Accordingly, AI-optimization systems can be seen as a model of proxy-based competitive systems in several important ways. Perhaps most relevantly, it allows the rigorous study of the purely statistical drivers of Goodhart's Law [36], one of which has recently been dubbed the 'unethical optimization principle' [45]. Another crucial advantage is the explicitly codified nature of AI architectures. While AI-systems are frequently decried as black boxes, the mechanisms underlying their decisions are at least in principle 'written down'. This means the processes leading to undesired outcomes can be experimentally studied and reproduced. It is perhaps one of the reasons why Goodhart's Law, one such process, is receiving abundant interest in AI-research, but remains underappreciated in competitive societal systems.

The above considerations suggest AI as a valuable epistemic tool, when trying to understand proxy-based competitive societal systems. One final observation is in order, which instead concerns the potential practical impact of using AI within such social systems. Specifically, the use of AIs may compound proxy-induced corruption for two reasons. Firstly, AIs may be used to increase the power of proxy optimization. In a system in which the proxy does not serve the goal well, this would amplify undesired effects. For instance, recently Salvador Pueyo [117] considered the potential effect of deploying powerful AIs to serve current profitability proxies, concluding this would probably compound problems such as environmental destruction and climate change. Secondly, an AI may lack the type of moral constraints or an intuitive understanding of the societal goal we invoked in the present model. While AIs can thus increase transparency in principle, this requires supervision and research by humans knowledgeable of the underlying goal (or more general moral standards). Without such supervision, AIs may pursue clearly unethical strategies with greater probability than if humans were involved, and without anyone noticing.

In summary, the burgeoning AI-safety literature is likely to drive the statistical understanding of proxy-based optimization. It already suggests three central conclusions, which we argue are fully translatable social systems: (i) there is an ideal optimization pressure which crucially depends proxy quality; (ii) Goodhart's Law should be understood as a near-inevitable system-level force arising from proxy-based optimization; (iii) this necessitates continuous higher-level monitoring to assess both where the original goal may be undermined or other negative side effects may occur. At the same time, the use of increasingly powerful AIs to optimize proxies, may supercharge adverse consequences within existing competitive social systems.

## 3.3. Cultural evolution perspective

A cultural evolutionary perspective is also closely related to the complex systems and the information processing perspectives outlined above [28]. It focuses on the mechanistic forces shaping system-level outcomes and dynamics, rather than the impact of individual agency and goals at equilibrium [29,30,52,118]. Furthermore, the evolutionary mechanisms studied can be seen as a particularly powerful optimization algorithm (§3.2). In this subsection, we first want to (i) consider the relation to economic and biological signalling theories and runaway selection and (ii) the novel notion of directed cultural evolution.

The proxy as described here is a signal by the competitor, to whomever or whatever implements competition. This relates proxyeconomics to economic or biological signalling theories (e.g. [47,48]). However, the present theory has a different focus from previous signalling theories, because an abstract societal goal is fundamentally different from a goal enforced by some higher-level selection (such as market or genetic selection, see §3.1). For instance, economic signalling theory explicitly focuses on the conditions leading to 'honest' signals, implicitly assuming that only these can survive market selection [47,99]. Here, the market is assumed to reflect a ground truth, i.e. to automatically

---

[9]Note that both the cost of improving the proxy and the potential detrimental effects of an imperfect proxy are likely to interact with competition. More frequent competitive evaluations may be *more costly to perform* while simultaneously incurring a *higher unobserved cost due to corruption pressure* [6].

correspond to the abstract goal, which the signal serves. Similarly, biological or anthropological signalling theories have focused on the conditions under which signals may become 'dishonest' or 'run away' [48,57], only within the constraint of genetic selection.

A runaway mechanism describes a situation in which a signal becomes statistically associated with its selection mechanism, leading to a positive feedback loop. For instance, so-called Fisherian runaway signalling in sexual selection describes a positive feedback loop between e.g. the size of a peacock's tail and a peahen's preference for large tails [48]. Similarly, 'runaway cultural niche construction' describes a situation in which the practices selected by a niche and those creating the niche become associated [57]. The 'niche' in our context corresponds to the competitive societal system (see §1.3). Such a dynamic is plausible in the present context, if past proxy winners determine current selection.[10] For instance successful academics who were selected based on a particular methodological approach may now select academics with similar approaches. In this sense, any proxy-based competition, in which the terms of competition are defined by past proxy winners, is reminiscent of runaway cultural niche construction.

While we thus believe the concept of 'runaway cultural niche construction' captures the phenomenon of self-referential proxy measures well, the concept becomes particularly relevant in circumstances where higher-level selection does not clearly apply. Within the literature cited above, runaway signalling is automatically related to and constrained by genetic selection. For instance, runaway sexual selection can only proceed as long as it supports, or at least does not undermine, the inclusive genetic fitness of the signalling animals. The seminal 'runaway cultural niche construction' paper by Rendell *et al.* [57] similarly focuses on the interaction of cultural and genetic selection. While a traditional signal can thus be either honest or dishonest/runaway, both are grounded in a higher-level selection: honest signals are underwritten by genetic fitness, dishonest ones constrained by it. As a result when dishonest or runaway signalling occurs, it is usually treated as random and undirected, since it has no clear relation to genetic fitness.

Here, we have suggested that competitive societal institutions can be considered as attempts to actively direct cultural evolution (see table 1 for examples). We can now more specifically posit that societal systems can be seen as runaway cultural niches, constructed to direct practices towards arbitrary societal goals. The notion that the concept of *runaway cultural niche construction* will prove crucial to understand current social developments has previously been suggested [55]. It has also been argued that the concept could help to reintegrate social and biological sciences [39], and that particularly its relation to agency needs to be considered [31]. We suggest the concept of the proxy provides a concrete path to implement this programme (see §4.7). Notably, further investigating the similarities and differences of various signalling and runaway mechanisms between biological and cultural systems promises novel insights in both domains. For instance, McCoy & Haig [44] have recently suggested that Goodhart mechanisms may be a fundamental driver of biological complexity, with interesting implications for the origins of social and economic complexity [28].

In summary, we have proposed that the *proxy* represents an attempt to translate an abstract societal goal into a cultural selection mechanism, which is itself not constrained by any higher-level selection. To date, most cultural evolutionary work has focused on ancient societies and gene-culture coevolution, a context in which the assumption that cultural evolution serves genetic evolution made sense [52,57]. By contrast, the focus of the present theory concerns the direction of purely cultural evolution in recent and current society, which may be much faster and more powerful [38,55] and merits the consideration of agency [31]. Our agent-based model showed how individual agency might shift the long-term system equilibrium away from full proxy orientation, towards a shared social goal. Future research might additionally model how the proxies themselves could be aligned to such goals, allowing to actively determine the direction of cultural evolution and runaway niche construction (see e.g. [108], §4.1).

# 4. Future directions

The remainder of the paper will outline a number of complementary research questions and implications in a diversity of both qualitative and quantitative disciplines (e.g. §§4.1–4.7) and conclude. While the individual sections in part venture well beyond the direct implications of the agent-based model, they all derive from the central concept of proxy-based competition. They thus help to convey how the

---

[10]This relates directly to the complex-system-theoretic notion of an auto-catalytic societal system, discussed above (§3.1).

model may be interpreted (and tested) in practice and illustrate the full potential scope and relevance of a broader transdisciplinary theory of proxy-based competition.

First, §4.1 will outline how the informational difference between proxy and goal gives rise to a fundamental epistemic gradient. This raises questions about how quantifiability interacts with Goodhart's Law, with important epistemological implications for the academic study of proxy-based competition. Second, §4.2 will draw on psychological literature, pointing to the most striking ways in which this epistemic gradient must be expected to affect individual decision makers. Third, §4.3 will suggest that proxy-based competition must be expected to systematically alter intuitive appraisals of what constitutes moral behaviour, in a way that is intricately related to the previous epistemic and psychological considerations. Fourth, §4.4 will suggest that the present theory directly links to prominent qualitative strands of sociological research. Fifth, §4.5 will suggest that it has systematic and relevant implications for policy research. Sixth, §4.6 will turn to the perhaps most controversial topic, namely the potential proxy orientation of whole economies. It will propose that a proxy-centric perspective both reproduces and informs the canonical Hayekian thesis, namely that markets are extremely efficient and accurate information processors [75,119]. Finally, §4.7 will outline how the empirical analysis of proxies can systematically guide the diagnosis and mitigation of excessive proxy orientation in real systems, by e.g. leveraging behavioural economic findings.

## 4.1. Epistemological implications

As outlined in the introduction, the proxy reflects a concrete operationalization of an abstract goal. This has an important epistemological implication, namely that there is a systematic epistemic gradient between proxy and goal. The proxy will, almost by definition, reflect our best attempt at quantifying the underlying goal. A goal may for instance include aspects about the future, which are fundamentally uncertain, or it may contain qualitative moral notions which are difficult to quantify.[11] The difference between proxy and goal is thus likely to systematically interact with quantifiability (or perhaps even more generally 'knowability'). This has important consequences for individual decision making, which we will come to below (§4.2). Here, we want to highlight that some broader philosophical issues must be considered.

First, the mere conceptualization of a goal requires the philosophical notion of purpose or goal-directedness [120]. The rejection of such a notion from e.g. cultural evolutionary research probably reflects implicit metaphysical assumptions, which must ultimately be addressed by philosophical reasoning [121]. Furthermore, stating a societal system can serve an abstract goal implicitly assumes such goals can somehow causally affect a physical system [120,122]. Both of these issues reflect ongoing philosophical debates concerning the nature of goals and causation [120,123]. The study of proxy-based competition explicitly depends on pushing such philosophical debates forward [123], as the very notions of proxy or Goodhart's Law are not well defined without goals.

Second, the practical definition of a social or moral goal requires qualitative research. What we strive for in society is an inevitably philosophical or political question. Pretending it does not require qualitative deliberation, because it is e.g. automatically given by some quantitative proxy measure, has been intensely criticized within the humanities for decades [109,110]. The present perspective supports this notion, by drawing attention to the information-theoretic reasons why proxies should generally not be assumed to perfectly reflect their underlying goals.

Third, qualitative aspects of the goal may be particularly susceptible to the processes of Goodhart's Law. The reason is that the proxy will generally be the best attempt to quantify, or operationalize, the societal goal. Yet not everything can be easily quantified, particularly in ongoing competition. Unfortunately, it will be precisely the aspects of the societal goal which are most 'knowable' (in any sense), which are most likely to be captured in the proxy (e.g. through the process of peer review). This directly biases corruption towards whatever is difficult to assess or quantify. Indeed, much of the evidence suggesting a corruption of practices tends to be qualitative, voiced in interviews, editorials and surveys [2,4,5,124]. While we would maintain it is important to complement such research with approaches yielding quantitatively falsifiable predictions (e.g. [49]), this does suggest fundamental limits to quantitative approaches with respect to the problems at hand. More specifically, it suggests an indispensable role of qualitative research in the ongoing assessment of real societal systems.

Combining the two latter observations, we suggest that the process of qualitatively critiquing proxies may itself be viewed as a social epistemic process helping us better define and debate our societal goals.

[11]Note that a bad proxy can nevertheless systematically undermine such aspects.

Conversely, avoiding to conceptualize goals as distinct from proxies is likely to foster Goodhart mechanisms and promote proxy capture. Importantly this holds both if the avoidance stems from warranted epistemic concerns or from metaphysical assumptions.

## 4.2. Psychological perspective

A central observation of our theory is that 'the proxy' will tend to become a target for individual decision makers. While this holds when 'decision makers' are conceptualized as insentient 'auto-catalytic systems' (§3.1), artificial intelligences (§3.2) or idealized 'rational agents', it is particularly interesting to consider realistic human beings. Specifically, a detailed analysis of a proxy (e.g. its time course, visibility, relation to the goal, implication for personal survival/success, etc.) is likely to reveal numerous interactions with particularities of the human decision mechanism (e.g. discounting, computational capacity, reference-dependence, loss-aversion, etc.). While the remainder of this subsection will touch upon just a few striking implications, a more general strategy to synthesize behavioural findings and an analysis of the proxy will follow below (§4.7).

First, there is overwhelming evidence that humans incorporate a moral component into their decisions [42,80,125,126]. While the economic multitasking model we adopted provides an elegant way to formalize this, it must be stated clearly, that the actual mechanisms are far from understood. For instance, it is unknown what the relative motivational power of moral and competitive incentives are, particularly for real professionals. Compared with laboratory settings both the moral incentive (e.g. treating a patient well) and the competitive incentive (actual professional survival), may be substantially more powerful.

Nevertheless, experimental investigations provide essential insight about the potential psychological mechanisms during such incentive conflicts [42,43], with systematic implication for proxy-based competition. For instance, the more goal-oriented option is likely to be associated with (i) *higher ambiguity*, (ii) *longer time frames*, and (iii) *less personal relevance* [127]. The proxy is almost by definition an attempt to make an abstract societal goal more concrete, immediate and personally relevant. Of course, this has unavoidable consequences for decisions, given the well-known phenomena of *ambiguity discounting, temporal discounting and social discounting* [128–130]. Moreover, it is likely to systematically affect the propensity for biased, or motivated, perception and cognition [131–133].

Another important question concerns the direct effect of competition on decisions. While a large body of experimental evidence demonstrates the remarkable power of competition as an incentive, the underlying mechanisms remain poorly understood, particularly in multitask settings [78]. However, neuroeconomic research is beginning to reveal the important role of loss-aversion in contests [86]. More generally, the fields of behavioural economics, psychology and neuroeconomics are beginning to reveal a set of actual, empirically validated, decision mechanisms. Specifically, they suggest a computationally bounded mechanism with multiple (moral/egoistic), potentially reference-dependent, valuation inputs converging into a single utility computation [89,134]. We have attempted to capture these confluent insights into our decision model in the simplest possible way, reproducing a remarkable set of empirical observations from the experimental contest literature (§2.4.2). However, the necessity for further elaboration and empirical validation is clearly vast.

Finally, there is substantial empirical evidence for inter-individual differences in moral/egoistic drive. For instance, there is direct experimental evidence for an increased propensity to sabotage in males than females [135], which may partially explain apparent productivity differences between the sexes [136]. Variable incentive strengths could be easily modelled as variations in goal scale (or prospect function exponent). Incorporating such variability could inform on the outcomes of empirically observed differences between genders or in 'Machiavellianism' scales [137,138].

## 4.3. Ethical implications

Another simple yet profound implication concerns the ethical structure of decision problems. So far, we have considered only an egoistic incentive, and an incentive towards an abstract societal goal. However, individual competitive success often also affects one's family, team or employees. In such cases, the 'egoistic' action may become a moral imperative. For instance, a scientist may have to weigh a questionable research practice against the social responsibility of securing funding for her employees. Similarly, a CEO may have to weigh her responsibility towards her employees and shareholders against environmental or social damage. In creating such a social responsibility, proxy-based

competition could be seen as making the corrupt option more ethical *per se*, especially if loyalty or other forms of in-group prioritization are viewed as ethical.

The full significance of this effect can only be appreciated by taking the epistemic gradient as well as its psychological implications into account (§4.1 and 4.2). Specifically, a 'corrupt' option will tend to entail a relatively uncertain, ambiguous and distant harm to society at large, which has to be weighed against a quite certain, clear and immediate harm to one's family, team-members, subordinates or compatriots. In such cases, many would arguably view the 'corrupt' option as clearly more ethical. It is worth returning to table 1 to consider how these psychological profiles recur across contexts.

Indeed, in practice, securing jobs is frequently invoked as a justification for otherwise problematic practices, such as selling arms, damaging the environment or economizing on worker safety. Such cases demonstrate that this is not a theoretical argument, but a central part of current public discourse. The present theory implies that this type of moral dilemma will tend to be automatically created by proxy-based competition, where it is likely to systematically affect decisions. Compounding the issue, the cultural evolution perspective (§3.3) suggests that proxy-based competition may in fact shape the ethical values we hold, selecting for e.g. loyalty as a moral priority.

## 4.4. Sociological perspective

Next, we want to draw attention to the link our theory provides to some traditions of qualitative sociological research. While a detailed discussion of these links is beyond the scope of this article (and the author's expertise), some similarities are conspicuous. A cultural (or professional) practice in the sense used here includes everything that is culturally transmitted, i.e. narratives, norms, language, valuations, etc. [5,52]. The detailed analysis of e.g. narratives or language in turn is the core focus of prominent qualitative research traditions (e.g. critical theory). Above (§3.1), we have further suggested that competitive societal systems are auto-catalytic (i.e. they are self-reinforcing power structures). Crucially, the process by which such societal systems auto-catalyse is often intricately linked to the proxy, since past (proxy) winners will tend to become those with most power to determine future proxies. This implies that the critical analysis of cultural practices (including language), with respect to their implications for societal power structures (i.e. the proxy), addresses necessarily arising frictions in proxy-based competition.

Our theory thus suggests that critical analysis of cultural practices should play a principal role in extricating the proxy, and associated self-referential power structures, from an underlying societal goal. Note that the theory further sketches when qualitative versus quantitative approaches are best suited and how both may interact in such an endeavour (§4.1). For instance, qualitative sociological research may identify subtle shifts in 'valuation practices' [5]. Modern computational sociological techniques may allow to further develop and test arising predictions, e.g. through the analysis of language patterns. Such a systematic integration of qualitative and quantitative approaches may prove most powerful to address the potential excessive proxy orientation of specific societal systems.

## 4.5. Political implications

While the anthropological and sociological approaches discussed above may avoid assuming that society can specify consensual 'goals', this is very much a foundational assumption of democratic political theory [139]. A principal purpose of the democratic political process, and many other forms of governance, is to continually determine and revise these goals. The present theory remains mostly agnostic about what these goals are (other than assuming that competition is not a goal in itself). Interestingly, however, many aspects of political ideology concern not the goals, but rather the means to achieve them [140].

For instance, one party may focus on inadequacies of the proxy, advocating costly improvement or regulation of specific proxy-oriented practices (e.g. environmental or social regulations). An opposing party may emphasize the correlation between proxy and goal, suggest that regulation or improving proxies will not be cost-effective, and point out that unilateral reorientation towards an ambiguous societal goal may result in (i) losing competition and (ii) no change at the system level. The present theory implies that both parties' arguments result from a deep intuitive understanding of proxy-based competition and that a confrontation of both views is probably necessary for an optimal regulation of a system. If, for instance, a specific informational weakness of the proxy cannot be cost-effectively improved, regulation of specific related practices can be considered and the cost of this can be weighed against the cost of simply accepting corruption. One specific, perhaps somewhat counterintuitive policy prescription to mitigate corruptive pressures, where the cost of improving the

proxy or regulating practices is too high, are partial lotteries. Intriguingly, these have been suggested for several domains independently such as politics and science [12,141].

Finally, it is important to again note that the institutions which create the proxy should at least partially be viewed as self-reinforcing power structures (§§3.1 and 4.4), because they are likely to be primarily designed by current proxy winners (professionals which have excelled within the current system). Accordingly, we might generally expect some inertia in societal systems where current proxy winners would decrease their power (or the valuation of their life legacy) by questioning the proxy. Notably, to the degree that the proxy captures the goal, this inertia (or conservatism) is valuable.

The present theory thus suggests that it is crucial to separate political debate about the societal goals from debate about the appropriate measures to achieve these goals. Establishing a consensus on the goals may help to overcome ideological investments in particular measures. The appropriateness of specific measures for a given goal can then be assessed by systematic and scientific analysis of the proxy and its impact in the specific system (see e.g. §4.7).

## 4.6. Proxyeconomics across scales: considering markets

So far, we have referred to the institutions/mechanisms which create the proxy as single coherent entities. However, this was an operational choice. In practice these institutions/mechanisms are often complex and may contain proxy-based competition themselves. For instance, the allocation of publication space in high-impact journals is determined through competition between authors, but also between journals. Such nested, or interlocked proxy-based competitions conceivably counteract or amplify corruption, probably depending on the overlap between informational deficits of the respective proxies. If both authors and journals can game their respective competitions by submitting/accepting more sensational content, corruption may be amplified. Generally, increasing the scale and complexity of a system is likely to increase the number of failure modes and thereby the scope for corruption [106]. However, if interlocking competitions largely counteract each others corruption potential, they may serve the goal with remarkable efficiency.

The market mechanism is the canonical example of a system of interlocked competitions which powerfully prevents many kinds of corruption, i.e. efficiently captures information about the presumed goal of maximizing welfare [75,119]. According to standard economic assumptions, optimizing the proxy measure of profit will automatically serve this societal goal [18].

We suggest that the informational efficiency argument first posited by Hayek [119], which underlies this standard assumption, is borne out and concretized by a proxy-based perspective. Specifically, Hayek's theoretical argument about the superior information processing capacity of markets can be recast in terms of proxy fidelity. Consider the proxy of profitability, which incorporates substantial information about consumer valuation as well as production cost, due to two interlocking competitive mechanisms. The distributed, continuous and direct (incentive-compatible) processes of markets are likely to prevent many sources of information loss, to which other mechanisms are susceptible. For instance, a system relying on representatives to infer peoples needs and valuations is unlikely to capture the same information as observing their actual choices. Accordingly, informational analysis suggests that market mechanisms should produce proxies, which capture the underlying goal far better than competing approaches (paradigmatically central planning by representatives). To see this most clearly, consider the competitive mechanisms underlying proxies in non-market-based systems, i.e. the mechanisms that determine their allocation of economic power.[12] In such systems, the proxy may be generated through e.g. the deliberations of individual planners, or perhaps the social and power relations within the governing body. It is difficult to conceive how this kind of mechanism could create proxies of similar quality as a market, given the goal of aggregate welfare [119,142].

Nevertheless, it in no way follows that market proxies are perfect, i.e. without systematic flaws. Mirowski & Nik-Khah [75] outline how the history of twentieth-century mainstream economics reflects the increasing insight that the precise mechanisms of markets (i.e. proxy aggregation) very much affect the obtained information, rather than reveal some ground truth. This opens up fundamentally new approaches to the study of markets and money [143]. Many such arguments suggest that whole market economies have become excessively proxy-oriented, i.e. that the pursuit of market proxies such as profit no longer necessarily serves the original goal of maximizing welfare [14,15,17,18,144]. Indeed, empirical research shows that, among rich nations, the positive welfare

---

[12]As discussed above (§1.2.1), any revealed ranking of agents, or revealed resource allocations, can be described by a proxy. Since resources also have to be allocated to agents in non-market systems, they also entail a proxy.

consequences of economic growth are quite challenging to even detect statistically, and are dwarfed by other sources of variability [23,145–147]. While this might be construed as an expected satiation effect, it does raise the question as to the purpose of exponentially increasing economic output in rich nations. Proxy orientation offers a simple, if provocative, answer to this question. It is consistent with the self-referentiality to be expected in proxy-based competition and a wealth of micro-level evidence (see §4.7). Additionally, widespread popular perception is consistent with the lock-in phenomenon introduced above, where individuals feel their economic activities do not serve general welfare [26].

Note that this argument does not even consider environmental externalities, such as $CO_2$ emissions, and the potentially dramatic negative long-term welfare consequences. Notably, $CO_2$ emissions show far less decoupling from economic growth than measures of well-being (the former relate roughly linear, while the latter relate roughly log-linear, see e.g. [148]). In our view, these considerations provide a strong case for excessive proxy orientation of entire economic systems [17,18,24,25,149], which merits a principled non-ideological discourse.

In summary, proxy-based competitive systems in practice are often complexly interlocked or nested. This may amplify or counteract corruption overall, rendering theoretical predictions difficult. Ultimately, it is probably necessary to empirically investigate the consequences of any given proxy to the purported underlying goal. Such an empirical assessment of the profit proxy, or more generally economic growth, raises the question how much economic activity is genuinely welfare enhancing and how much predominantly serves self-referential proxy measures [17,18,150].

## 4.7. Diagnosis and mitigation

Next, we want to elaborate how the present theory may guide diagnosis and intervention for real societal systems. We have already discussed the three general approaches to mitigation suggested directly by our model, namely to increase proxy fidelity, to adjust competitivity, or to foster the intrinsic motivation towards the social goal (§2.4.4). However, these do not further clarify how excessive proxy orientation can be diagnosed in the first place. Here, we want to outline a specific strategy towards diagnosis, which can subsequently be used for targeted mitigation.

Specifically, we propose to systematically investigate empirical idiosyncrasies of proxy-generation in order to predict meso-level patterns of corruption. Such predictions would allow the diagnosis of corruption within societal systems, as well as the design of mitigation strategies [1,13,49,151]. For instance, we have recently modelled how an informational idiosyncrasy of the academic publication process, namely positive publication bias, predicts specific 'corrupt' patterns of sample-size choices. Positive publication bias reflects an idiosyncrasy of the academic publication process, namely the phenomenon that 'positive' findings (i.e. findings of some non-null effect) are far more likely to get published than negative findings (i.e. null effects). Under plausible assumptions, this idiosyncrasy should give rise to a bimodal distribution of achieved statistical power, and relatively low overall replicability, even though both would undermine the actual goal of academic competition [49]. Yet both have been found empirically, supporting the notion of excessive proxy orientation [152–154]. Having identified not only a 'corrupt pattern', but also the informational idiosyncrasy at its source, now additionally allows the design of targeted mitigation strategies [49,155].

This approach can be generalized to market contexts. Specifically, we propose decision biases identified in behavioural economics can be systematically leveraged to predict patterns of corruption within societal systems. Excessive proxy orientation should be reflected in consumer choice architectures [156,157] which capitalize on known decision biases to increase profit [151,158]. Such choice architectures may e.g. foster addictive behaviours or withhold relevant information [158,159]. For instance, advertisements have recently been used to analyse behavioural market failure in the payday lending market [160]. Similarly, automated web-crawls have been used to identify corrupt, i.e. 'dark' patterns at a large scale [158]. Accordingly, behavioural economics can be used to derive fine-grained, falsifiable predictions of corruption patterns. Prevalent marketing and advertising practices could be readily analysed with respect to their implications for proxy and goal performance, given known decision biases. Behavioural economics can then be further used to devise choice architectures, which mitigate decision biases and therewith corruption [161]. In other words, the present theory provides a principled information theoretic motivation for targeted behavioural economic intervention. Crucially, its predictive accuracy can be assessed/validated by empirical findings at the behavioural and market level, rendering it systematically superior to purely theoretical arguments.

Finally, we want to emphasize a factor that may foster or inhibit proxy-induced corruption in multiple ways, namely the spread of 'narratives' [105]. We highlight this, because the present theory speaks

directly to a particular type of corruptive narrative, namely that 'a particular proxy perfectly reflects its underlying goal'. For instance, within economics this narrative takes the form that *'maximizing profit will also maximize social welfare'* [18,75]. The present theory takes a competing narrative as its foundational premise, namely that *proxies never perfectly reflect their underlying goals*. To reiterate, this is in no way meant to discourage the use of proxies *per se*, which are indispensable in practice and probably generally beneficial. However, any narrative which uncritically equates a proxy with its goal is likely to promote Goodhart's Law, leading to systems which revolve around proxy measures irrespective of the underlying goals.

# 5. Conclusion

We have outlined a transdisciplinary theory of proxy-based competition and cultural evolution, which applies whenever a societal system employs proxy measures to mediate competition in service of an abstract goal. The theory suggests that proxies are inevitable, but that a theoretical and empirical focus on the information content of the proxy, i.e. the question where it does and does not approximate its underlying goal, is essential. We further proposed that a focus on proxies will help to link inquiry across disciplines, in both theory and practice.

Our agent-based computational model synthesized an economic and a cultural evolutionary mechanism, thereby integrating several positive and negative effects of competition (including Goodhart's Law). It suggests a central role of competition on effort expenditure, individual utility, selection and the cultural evolution towards corrupt (i.e. proxy-oriented) practices. Accordingly, there may be an optimal level of competition, and a moderatable corruption pressure, depending on the complexity of the system and the preferences of the participating agents. The model shows that an economic decision mechanism, which includes an intrinsic goal motivation, can constrain the evolution to corrupt practices. It suggests a mechanistic account of how a system can remain *locked in* to a relatively proxy-oriented state, even if all individual agents know of, and value, the actual societal goal.

More generally, the theory provides a conceptual and predictive framework to empirically assess the degree to which cultural or societal systems may be wastefully or detrimentally oriented towards proxy measures. This may help to explain and address phenomena as diverse as the scientific replicability crisis or inaction to the threat of global warming.

Data accessibility. All code used in this study are openly accessible through https://github.com/oliverbraganza/Proxyeconomics_original. All data can be fully reproduced by running the provided code.

Competing interests. I declare I have no competing interests.

Funding. The project was funded through the VW-Foundation programme *Originalitaetsverdacht*.

Acknowledgements. I thank Heinz Beck for continuous support at his institute. I further thank Jonathan Ewell and Everard Braganza for detailed feedback on the manuscript. I also thank Klaus G. Troitzsch, Gerben Ter Riet, Susann Fiedler, Ashley Braganza, Markus Gabriel, Tony Kelly, Christina Selenz, several anonymous reviewers and the coordinators and participants of CCS2018, NRIN2018, BBS2019, NeuroPsychoEconomics2018 for the opportunity to present this research and/or invaluable feedback.

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
