## [Peer Review File · Royal Society Open Science]

Review History

RSOS-200971.R0 (Original submission)

Review form: Reviewer 1

Is the manuscript scientifically sound in its present form?

No

Are the interpretations and conclusions justified by the results?

No

Is the language acceptable?

Yes

Do you have any ethical concerns with this paper?

No

Have you any concerns about statistical analyses in this paper?

No

Recommendation?

Reject

Comments to the Author(s)

This paper though interesting, was difficult to review. The thrust of the paper is that competition and cultural evolution often occur not on the actual trait of interest, but on a proxy for this trait - hence "proxyeconomics". It presents a model to show how this might play out.

The paper draws on many disciplines to frame the model, but is patchy and some what haphazard in this. For example, the present work overlaps with signalling theory in both evolution and economics, which can lead to runaway selection (classic peacock's tail or tattooing in cultural evolution) or green beard type phenomenon. But none of this is discussed. To be most helpful, I've written my reactions to the paper as I read it:

The paper is about formally modeling Goodhart's Law (when a measure becomes a target, it's no longer a good measure) and the related Campbell's Law (that measure will be subject to corruption as people condition on the measure rather than the underlying trait). Table 1 is a nice illustration of this, though I'm less convinced by the politics and markets columns (that populism and lobbying is a good example in politics or markets are optimizing wellbeing).

The paper then makes the case for selection on the proxy, which becomes the measure of "fitness" being optimized. The make the case that the proxy can be corrupted, the author points out that the proxy cannot contain all information in the target and that there are mechanisms that undermine the information content of the proxy. This section is mostly asserted (rather than argued) with some citations, but was largely unclear. It was also unclear why the proxy (which is really a signal) doesn't eventually get recognized as a poor signal.

The final paragraph (p4 of 37) was highly abstract and more confusing. Proxyeconomics studies the "emergent information strengths and weaknesses" by investigating "time-courses and mechanisms of aggregation and use". Then the proxy's psychological impact and salience to neuroscientifically realistic individual decision-makers. All in all, I only had a vague sense for what the author was trying to achieve and the connections to other fields seemed superficial at best.

Sections 1.3 and 1.4 continued this trend and did a poor job of specifically situating this work in any literature.

Section 2 introduces the model, which is agent-based. The explanation for the model was again confusing, so I focused on the specifics. As a small aside, it might be better to subscript the variable names than use two letters which may be misunderstood as the product of two variables.

The model sets up the deviation between the proxy and the goal as an angle between two vectors, which seems reasonable. θ is an evolving angle for effort between these two. Selection pressure is adjusted via two variables - fraction of losers and probability a loser dies. I'm not sure why these weren't combined. An attempt is also made to incorporate a psychological goal scale (gs), though why this doesn't evolve was unclear, but it seems like some kind of individual decision to deviate from the norm.

Overall, like the paper in general, the model seemed to be trying to do too much at once, making it difficult to generalize the findings. Section 3 continued the trend of trying to connect the model to too much, but only doing so superficially. Unfortunately, I was not impressed and I recommend major revisions to more clearly explain the idea and connect it to specific existing models in any of the fields the author is trying to draw in. I don't want to be too harsh - I think

there is something here regarding how proxies can deviate from the underlying goal, but I think the question itself that the model is trying to shed light on needs to be better articulated and then more clearly presented.

Review form: Reviewer 2 (Maximilian Fochler)

Is the manuscript scientifically sound in its present form?

Yes

Are the interpretations and conclusions justified by the results?

Yes

Is the language acceptable?

Yes

Do you have any ethical concerns with this paper?

No

Have you any concerns about statistical analyses in this paper?

No

Recommendation?

Accept with minor revision (please list in comments)

Comments to the Author(s)

Dear Dr. Braganza,

thank you for the opportunity to read this paper.

Before I give my comments, allow me to situate the position from which I speak. I am a social scientist mainly working on the social dynamics of academia and their epistemic consequences. As such, your paper and its arguments touch my research, which is also highly concerned with the social and epistemic effects of competition. However, my expertise is mainly in qualitative methods. Thus I lack the expertise to fully judge the methodological details of your model.

Overall, this paper was a pleasure to read. Though as mentioned I cannot judge the methodological details, I was impressed with the clarity with which you describe your model and its implications. This allowed me as a more generalist than specialist reader to follow your argument and also scrutinize the underlying assumptions of your model to a degree that is appropriate for a scientific publication.

I highly appreciate the ambition of your paper to propose a general model that can stimulate thought and discussion on similar dynamics in very different social contexts and systems. In relation to the literature I have expertise on, on proxies and metrics in science, both your assumptions and your findings resonate very well with what has been described in social science analysis. However, your arguments also have the potential to add to this literature, on at least two levels. First, particularly your arguments about the respective capacities of qualitative and quantitative approaches are highly novel and valid. Second, the level of your theorizing which allows to compare dynamics to related phenomena in other social systems is highly inspiring. That said, as a qualitative social scientist oriented towards the micro, I could of course easily produce a long list of points along which your model is not fully realistic e.g. in its application to

science (e.g. talent to satisfy the proxy might be different from talent to meet the societal goal). But I recognize that it is the approach of every model to be reductive and to reduce to key elements of the dynamics addressed. Further studies might build on this, focus on more specific empirical domains and work with even more refined assumptions.

Some minor suggestions for improvement:

- For a social science readership, your paper would benefit from more clearly outlining its concept of cultural evolution earlier in the paper. As a reader, this was unclear to me for some time. And as there are very different concepts of cultural evolution, it is important to know what you mean precisely.

- In figure 3, I found it confusing that while lines A-D have the same column logic relating to different levels of competition, E does not follow this logic. I would recommend either visually flagging this more or moving E to a separate Figure.

In my reading, your paper has high potential to spark new interdisciplinary collaborations and conversations of high theoretical relevance to very important contemporary social phenomena. Thank you again for the opportunity to read it.

Decision letter (RSOS-200971.R0)

Dear Dr Braganza

The Editors assigned to your paper RSOS-200971 "Proxyeconomics, a theory and model of proxy-based competition and cultural evolution" have now received comments from reviewers and would like you to revise the paper in accordance with the reviewer comments and any comments from the Editors. Please note this decision does not guarantee eventual acceptance.

Please submit your revised manuscript and required files (see below) no later than 21 days from today's (ie 06-Oct-2020) date. Note: the ScholarOne system will 'lock' if submission of the revision is attempted 21 or more days after the deadline. If you do not think you will be able to meet this deadline please contact the editorial office immediately.

Please note article processing charges apply to papers accepted for publication in Royal Society Open Science (<https://royalsocietypublishing.org/rsos/charges>). Charges will also apply to papers transferred to the journal from other Royal Society Publishing journals, as well as papers

submitted as part of our collaboration with the Royal Society of Chemistry (<https://royalsocietypublishing.org/rsos/chemistry>). Fee waivers are available but must be requested when you submit your revision (<https://royalsocietypublishing.org/rsos/waivers>).

on behalf of Professor Bart de Moor (Associate Editor) and Marta Kwiatkowska (Subject Editor)
openscience@royalsociety.org

Reviewer comments to Author:

Reviewer: 1

Comments to the Author(s)

This paper though interesting, was difficult to review. The thrust of the paper is that competition and cultural evolution often occur not on the actual trait of interest, but on a proxy for this trait - hence "proxyeconomics". It presents a model to show how this might play out.

The paper draws on many disciplines to frame the model, but is patchy and some what haphazard in this. For example, the present work overlaps with signalling theory in both evolution and economics, which can lead to runaway selection (classic peacock's tail or tattooing in cultural evolution) or green beard type phenomenon. But none of this is discussed. To be most helpful, I've written my reactions to the paper as I read it:

The paper is about formally modeling Goodhart's Law (when a measure becomes a target, it's no longer a good measure) and the related Campbell's Law (that measure will be subject to corruption as people condition on the measure rather than the underlying trait). Table 1 is a nice illustration of this, though I'm less convinced by the politics and markets columns (that populism and lobbying is a good example in politics or markets are optimizing wellbeing).

The paper then makes the case for selection on the proxy, which becomes the measure of "fitness" being optimized. The make the case that the proxy can be corrupted, the author points out that the proxy cannot contain all information in the target and that there are mechanisms that undermine the information content of the proxy. This section is mostly asserted (rather than argued) with some citations, but was largely unclear. It was also unclear why the proxy (which is really a signal) doesn't eventually get recognized as a poor signal.

The final paragraph (p4 of 37) was highly abstract and more confusing. Proxyeconomics studies the "emergent information strengths and weaknesses" by investigating "time-courses and mechanisms of aggregation and use". Then the proxy's psychological impact and salience to neuroscientifically realistic individual decision-makers. All in all, I only had a vague sense for what the author was trying to achieve and the connections to other fields seemed superficial at best.

Sections 1.3 and 1.4 continued this trend and did a poor job of specifically situating this work in any literature.

Section 2 introduces the model, which is agent-based. The explanation for the model was again confusing, so I focused on the specifics. As a small aside, it might be better to subscript the variable names than use two letters which may be misunderstood as the product of two variables.

The model sets up the deviation between the proxy and the goal as an angle between two vectors, which seems reasonable. θ is an evolving angle for effort between these two. Selection pressure is adjusted via two variables – fraction of losers and probability a loser dies. I'm not sure why these weren't combined. An attempt is also made to incorporate a psychological goal scale (gs), though why this doesn't evolve was unclear, but it seems like some kind of individual decision to deviate from the norm.

Overall, like the paper in general, the model seemed to be trying to do too much at once, making it difficult to generalize the findings. Section 3 continued the trend of trying to connect the model to too much, but only doing so superficially. Unfortunately, I was not impressed and I recommend major revisions to more clearly explain the idea and connect it to specific existing models in any of the fields the author is trying to draw in. I don't want to be too harsh – I think there is something here regarding how proxies can deviate from the underlying goal, but I think the question itself that the model is trying to shed light on needs to be better articulated and then more clearly presented.

Reviewer: 2

Comments to the Author(s)

Dear Dr. Braganza,

thank you for the opportunity to read this paper.

Before I give my comments, allow me to situate the position from which I speak. I am a social scientist mainly working on the social dynamics of academia and their epistemic consequences. As such, your paper and its arguments touch my research, which is also highly concerned with the social and epistemic effects of competition. However, my expertise is mainly in qualitative methods. Thus I lack the expertise to fully judge the methodological details of your model.

Overall, this paper was a pleasure to read. Though as mentioned I cannot judge the methodological details, I was impressed with the clarity with which you describe your model and its implications. This allowed me as a more generalist than specialist reader to follow your argument and also scrutinize the underlying assumptions of your model to a degree that is appropriate for a scientific publication.

I highly appreciate the ambition of your paper to propose a general model that can stimulate thought and discussion on similar dynamics in very different social contexts and systems. In relation to the literature I have expertise on, on proxies and metrics in science, both your assumptions and your findings resonate very well with what has been described in social science analysis. However, your arguments also have the potential to add to this literature, on at least two levels. First, particularly your arguments about the respective capacities of qualitative and quantitative approaches are highly novel and valid. Second, the level of your theorizing which allows to compare dynamics to related phenomena in other social systems is highly inspiring. That said, as a qualitative social scientist oriented towards the micro, I could of course easily produce a long list of points along which your model is not fully realistic e.g. in its application to science (e.g. talent to satisfy the proxy might be different from talent to meet the societal goal). But I recognize that it is the approach of every model to be reductive and to reduce to key elements of the dynamics addressed. Further studies might build on this, focus on more specific empirical domains and work with even more refined assumptions.

Some minor suggestions for improvement:

- For a social science readership, your paper would benefit from more clearly outlining its concept of cultural evolution earlier in the paper. As a reader, this was unclear to me for some time. And as there are very different concepts of cultural evolution, it is important to know what you mean precisely.

- In figure 3, I found it confusing that while lines A-D have the same column logic relating to different levels of competition, E does not follow this logic. I would recommend either visually flagging this more or moving E to a separate Figure.

In my reading, your paper has high potential to spark new interdisciplinary collaborations and conversations of high theoretical relevance to very important contemporary social phenomena. Thank you again for the opportunity to read it.

===PREPARING YOUR MANUSCRIPT===

===PREPARING YOUR REVISION IN SCHOLARONE===

Author's Response to Decision Letter for (RSOS-200971.R0)

See Appendix A.

RSOS-200971.R1 (Revision)

Review form: Reviewer 2 (Maximilian Fochler)

Is the manuscript scientifically sound in its present form?

Yes

Are the interpretations and conclusions justified by the results?

Yes

Is the language acceptable?

Yes

Do you have any ethical concerns with this paper?

No

Have you any concerns about statistical analyses in this paper?

No

Recommendation?

Accept as is

Comments to the Author(s)

Dear Dr. Braganza,

thank you for your detailed response and revisions. They address the points I have raised very well. Again, I highly appreciate the interdisciplinary quality and ambition of your paper, and would be very happy to see it in print in this present form.

Review form: Reviewer 3

Is the manuscript scientifically sound in its present form?

No

Are the interpretations and conclusions justified by the results?

No

Is the language acceptable?

Yes

Do you have any ethical concerns with this paper?

No

Have you any concerns about statistical analyses in this paper?

No

Recommendation?

Reject

Comments to the Author(s)

This paper proposed a model of proxy-based competition and cultural evolution, particularly focusing on a differentiation of the population into two groups: motivated and discouraged groups.

Before the detailed discussions and relationships with interdisciplinary topics, I doubt that this is merely due to a particular type of cultural selection algorithm adopted in this model. In the model, a fixed prop. of the (motivated) best agents are secured and survive, and the remaining (discouraged) agents have randomly chosen with a fixed probability and replaced with a copy (with a mutation) of a randomly chosen agent in the former group. This extreme selection algorithm (i.e., elitism of adaptive agents + neutral selection on others) appears to be a major reason why the population evolved to be two different types of agents.

However, surprisingly, there are no clear discussions on the validity of using this type of algorithm to discuss this kind of phenomenon. I believe that this is necessary to understand the significance of the findings. After that, the authors need to show why such a phenomenon obtained from such a strong assumption is significant in this context and not merely due to artifacts caused by this algorithm. In other words, if a fitness proportional and stochastic selection (i.e., so-called roulette wheel selection), used often in evolutionary computations, is adopted, how the results will change and what does it mean in this context?

Another factor is the difference in the talents among agents. This appears to simply determine whether the agents will be fall into the motivated or discouraged groups, under this extreme selection pressure. This is not surprising to me.

The description of the model is not clear and the important procedures are described verbally. Thus, I recommend the authors to describe the whole algorithm by using a pseudo-code. I also recommend them to use capital letters to represent a parameter (i.e., a fixed value through a trial), which is one of the standard ways to discriminate between the fixed parameters and changeable variables, as far as I know.

In the results, I do not understand why the authors do not directly show the changes in γ and e , which are the actually evolving and changing variables. The authors need to show this and discuss what happened through the trial more clearly.

In addition, while other reviewers pointed out the relationship between this model and signaling theory and sexual selection, I do not understand why this is related to them. No signal is used in the proposed model in that agents do not exchange information about others. Also, a runaway process never occurs because there is no sexual selection in the model.

Due to these points, I do not understand the significance of the model and results, and how the finding can be related to the interdisciplinary phenomena discussed in this paper, from the current manuscript.

Minor:

Line 455: "In a subset of models": What do you mean by "subset"?

Decision letter (RSOS-200971.R1)

Dear Dr Braganza

The Editors assigned to your paper RSOS-200971.R1 "Proxyeconomics, a theory and model of proxy-based competition and cultural evolution" have made a decision based on their reading of the paper and any comments received from reviewers.

Regrettably, in view of the reports received, the manuscript has been rejected in its current form. However, a new manuscript may be submitted which takes into consideration these comments.

We invite you to respond to the comments supplied below and prepare a resubmission of your manuscript. Below the referees' and Editors' comments (where applicable) we provide additional requirements. We provide guidance below to help you prepare your revision.

Please note that resubmitting your manuscript does not guarantee eventual acceptance, and we do not generally allow multiple rounds of revision and resubmission, so we urge you to make every effort to fully address all of the comments at this stage. If deemed necessary by the Editors, your manuscript will be sent back to one or more of the original reviewers for assessment. If the original reviewers are not available, we may invite new reviewers.

Please resubmit your revised manuscript and required files (see below) no later than 09-Nov-2021. Note: the ScholarOne system will 'lock' if resubmission is attempted on or after this deadline. If you do not think you will be able to meet this deadline, please contact the editorial office immediately.

Please note article processing charges apply to papers accepted for publication in Royal Society Open Science (<https://royalsocietypublishing.org/rsos/charges>). Charges will also apply to papers transferred to the journal from other Royal Society Publishing journals, as well as papers submitted as part of our collaboration with the Royal Society of Chemistry (<https://royalsocietypublishing.org/rsos/chemistry>). Fee waivers are available but must be requested when you submit your manuscript (<https://royalsocietypublishing.org/rsos/waivers>).

Thank you for submitting your manuscript to Royal Society Open Science and we look forward to receiving your resubmission. If you have any questions at all, please do not hesitate to get in touch.

on behalf of Professor Bart de Moor (Associate Editor) and Marta Kwiatkowska (Subject Editor)
openscience@royalsociety.org

Associate Editor Comments to Author (Professor Bart de Moor):

Comments to the Author:

Regrettably, one of the reviewers sought has found substantial concerns with the paper - given the journal's general policy of only permitting one round of major revision, we must decline the paper at this stage; however, if you wished to submit a re-worked version of the paper for consideration for publication, you would be most welcome to do so. If, however, you felt your efforts may be better rewarded elsewhere, this decision allows you this flexibility (though, as I'm sure you can imagine, we hope you'll resubmit to RSOS!).

Reviewer comments to Author:

Reviewer: 2

Comments to the Author(s)

Dear Dr. Braganza,

thank you for your detailed response and revisions. They address the points I have raised very well. Again, I highly appreciate the interdisciplinary quality and ambition of your paper, and would be very happy to see it in print in this present form.

Reviewer: 3

Comments to the Author(s)

This paper proposed a model of proxy-based competition and cultural evolution, particularly focusing on a differentiation of the population into two groups: motivated and discouraged groups.

Before the detailed discussions and relationships with interdisciplinary topics, I doubt that this is merely due to a particular type of cultural selection algorithm adopted in this model. In the model, a fixed prop. of the (motivated) best agents are secured and survive, and the remaining (discouraged) agents have randomly chosen with a fixed probability and replaced with a copy (with a mutation) of a randomly chosen agent in the former group. This extreme selection algorithm (i.e., elitism of adaptive agents + neutral selection on others) appears to be a major reason why the population evolved to be two different types of agents.

However, surprisingly, there are no clear discussions on the validity of using this type of algorithm to discuss this kind of phenomenon. I believe that this is necessary to understand the significance of the findings. After that, the authors need to show why such a phenomenon obtained from such a strong assumption is significant in this context and not merely due to artifacts caused by this algorithm. In other words, if a fitness proportional and stochastic selection (i.e., so-called roulette wheel selection), used often in evolutionary computations, is adopted, how the results will change and what does it mean in this context?

Another factor is the difference in the talents among agents. This appears to simply determine whether the agents will be fall into the motivated or discouraged groups, under this extreme selection pressure. This is not surprising to me.

The description of the model is not clear and the important procedures are described verbally. Thus, I recommend the authors to describe the whole algorithm by using a pseudo-code. I also recommend them to use capital letters to represent a parameter (i.e., a fixed value through a trial), which is one of the standard ways to discriminate between the fixed parameters and changeable variables, as far as I know.

In the results, I do not understand why the authors do not directly show the changes in γ and e , which are the actually evolving and changing variables. The authors need to show this and discuss what happened through the trial more clearly.

In addition, while other reviewers pointed out the relationship between this model and signaling theory and sexual selection, I do not understand why this is related to them. No signal is used in the proposed model in that agents do not exchange information about others. Also, a runaway process never occurs because there is no sexual selection in the model.

Due to these points, I do not understand the significance of the model and results, and how the finding can be related to the interdisciplinary phenomena discussed in this paper, from the current manuscript.

Minor:

Line 455: "In a subset of models": What do you mean by "subset"?

===PREPARING YOUR MANUSCRIPT===

===PREPARING YOUR REVISION IN SCHOLARONE===

Author's Response to Decision Letter for (RSOS-200971.R1)

See Appendix B.

RSOS-211030.R2 (Revision)

Review form: Reviewer 3

Is the manuscript scientifically sound in its present form?

Yes

Are the interpretations and conclusions justified by the results?

Yes

Is the language acceptable?

Yes

Do you have any ethical concerns with this paper?

No

Have you any concerns about statistical analyses in this paper?

No

Recommendation?

Accept as is

Comments to the Author(s)

Thank you very much for the detailed responses to the comments. Now I fully understood that the results are robust to the selection algorithms, and additional comments on the signaling theory and runaway process made sense to me. Instead of providing a pseudo-code, the author provided the source codes of the model, which is very nice. However, I still believe that providing an overview of the whole model structure in the paper itself is informative to readers, which should make them much easier to understand the results and findings as well as the model definition itself.

Review form: Reviewer 4

Is the manuscript scientifically sound in its present form?

Yes

Are the interpretations and conclusions justified by the results?

Yes

Is the language acceptable?

Yes

Do you have any ethical concerns with this paper?

No

Have you any concerns about statistical analyses in this paper?

No

Recommendation?

Accept with minor revision (please list in comments)

Comments to the Author(s)

Agent-based models are difficult to do well, and this paper presents a strikingly good one. The paper models a competitive social system by combining (1) evolutionary selection on rank according to some measure with (2) a multi-tasking utility function for the agents that defines this measure as an imperfect proxy of some shared social goal. This is a major contribution in that it synthesises top approaches from several disciplines into a coherent whole; the resulting model is simple and yet it explains a lot.

Partially in response to prior reviewer comments, the author has included in the paper two implementations for each of (1) and (2). Specifically, (1) a thresholded or a proportional selection pressure on agents according to rank by the proxy measure and (2) a prospect or linear utility to the agents of their own rank according to the proxy measure.

Here listed are the main findings of the paper, as I understood them, none of which depend on the choice for (1) or (2):

(A) better proxy measures produce outcomes better aligned with a shared social goal

This is a key finding given that a major contribution of the paper is to put forward a model of cultural evolution where a shared social goal is able to be defined in the first place. That a shared social goal both exists and motivates agents is well-referenced (lines 399-416 and Section 3.4). The notion that prior literature on cultural evolution gives unsatisfactory treatment to the known/stated/implied/emergent goals of competitive social systems is well-argued (Section 3.6 and scattered elsewhere).

(B) moderate levels of competition produce better outcomes with respect to the shared social goal

This arises out of the interaction between two emergent properties of the model:

(B1) higher levels of competition better screen for talent (this is good)

(B2) higher levels of competition better screen for proxy-orientation (this is bad)

These are key findings given that a major contribution of the paper is to put forward a model of signalling/screening that explicitly incorporates selection pressure. That evolutionary selection exists in competitive social systems is well-referenced (lines 116-127). The notion that prior literature on signalling/screening gives unsatisfactory treatment to mechanisms of selection is well-argued (lines 963-72 and scattered elsewhere).

(C) intrinsic motivation bounds evolution but also creates agents unhappy with "locked in" practices

The basic result that intrinsic motivation bounds evolution is borderline tautological. But the *mechanism* whereby a positive g leads to higher equilibrium θ is non-trivial and intriguingly realistic (Section 2.4.5). The notion that defining an intrinsic motivation both bounds the long-term evolution of the system AND makes agents self-aware enough to be discouraged by that equilibrium is really insightful!

Taken together, the findings A, B, and C make for a compelling argument that this paper's combined approach results in a model that is greater than the sum of its parts. The depth of the discussion that can be had by interpreting this model in the context of several real-world systems

further supports the authors' meta-argument: that combining (1) and (2) produces something coherent enough to deserve its own name; proxyeconomics is a bold one.

HOWEVER. Agent-based modelling papers are also very difficult papers to write, and as a reader it took me *a lot of work* to distill A, B, and C from the current version of the paper. This was my job as a reviewer, which I take seriously. But it was a struggle: even though I am well within the target audience for this paper it took me several days to get through it and there was much I had to look up. Many readers might just put the paper down.

My recommendation to the editor is *minor revisions*. Highly effective writing is not listed as a publication criteria on the Royal Society Open Science Guidelines for referees; nor should it be. This research is sound. The paper is fit to be published upon clarification and/or correction of the following:

-- On lines 337-9 it is noted that agents are selected in a random order to update their effort based on their "current" rank. Please clarify that "current" means rank-calculated-in-the-past-timestep. The update order within a timestep should not affect agent calculations (except in the literal sense of the random seed). It may be that this clarification is found on lines 484-6; that is too far away from where the concern arises for me to be confident that the update order doesn't matter for the model as a whole. Alternatively, provide these related technical details in the same section.

-- In the implementation of proportional selection pressure, the appropriate analog to intensity of competition (c) is the *slope* of the proportionality between rank and relative probability of removal. Higher slope gives a larger protective effect to higher ranks, i.e. rank is more important and competition is more intense. The number of agents selected each round is analogous to the absolute selection pressure, very very similar to in the thresholded implementation. This is not clear from the text.

-- If the result in Figure 4 (i.e. result B) requires prospect utility, do note that.

-- In discussing the implications of the model results for market economies (Section 3.9) please clearly state what is being considered a proxy for what. In this sub-section, specifically, we are discussing profit as an imperfect proxy for what, exactly? Perhaps refer back to Table 1.

-- By the authors' own admission in the text of the paper, several sub-sections in the Discussion are highly speculative. Please move the Discussion sub-sections that venture well beyond the present model into a Future Directions section (or something) so it is clear where the science-discussion ends and the meta-discussion begins. Contributing to the literatures on (1) and (2) is science-discussion. Relating specific modelling concepts across fields is science-discussion. Sweeping implications for entire fields of research (biology, economics, sociology, ethics, policy, etc.) is meta-discussion.

My recommendation to the author themselves is *major revisions*. This work is great, so there could be a tremendous payoff from presenting it more effectively. Specifically, re-ordering the paper and modularizing the writing would help it find a larger and more receptive audience. Taking the following steps would make it far easier, and thus far more likely, for the reader to grasp that combining (1) and (2) leads to results A, B, and C.

-- This model combines approaches that are considered standard in two different fields. Right now, the intro is legible only to those with a double background or enough humility to Google dozens of terms; this is a small audience. Most readers would benefit from concise explainers on (1) and (2), separately, where you collect together the known/expected/predicted results of such models, common implementations, and key references for those who are unfamiliar. Present the terms you repeatedly use in their usual context, first, so they avoid being jargon disconnected from their origin.

- An explainer on (1) in the intro would be a great place to put a summary of reference [1] that you can point back to later (e.g. lines 253, 395, 860-80). Note, especially, that their model and ones like it struggle to define a social goal! This is discussed in Section 3.6 as a curious side-note, even though it forms the central contribution of the paper to the literature on (1).

- An explainer on (2) in the intro would be a great place to put technical references on multi-tasking models that are currently scattered through the manuscript, and references from Section 3.4. Note, especially, that these models rely on a notion of "good" or "honest" signalling that this literature struggles to define. This is discussed in Section 3.1 as a curious side-note, even though it forms the central contribution of the paper to the literature on (2).

-- Once you've briefed the reader on (1) and (2), separately, they will be better able to follow a qualitative description of your model (and more likely to believe that the combined model is greater than the sum of its parts). Focus on how this paper combines them: defining a practice as a vector in a proxy-goal space that engages with both (1) and (2). This is a good place for Figure 1, which is very intuitive.

-- When you then go on to describe the model in technical mathematical detail, please just describe the model. Sentences defining core concepts should be early in the Introduction (e.g. "social goal" on line 399). Sentences describing what your model allows you to incorporate should be in the qualitative description (e.g. lines 423-6). Sentences that describe what emerges from the model should be in the Results section (e.g. lines 269-74 and 308-323). Sentences describing the ranges used should be in Implementation (e.g. line 469). Sentences describing what various parameters "reflect", could be "viewed as", or could be "thought of" belong in the Discussion (too many to list). Sentences supporting your meta-argument should be somewhere else. Just define your model! Move all this other stuff out of the technical section.

- Corollary to the above: "attempting to convey the motivation underlying each specific operationalization choice" (lines 325-6) does the paper no favors. The best defence is a good offence---motivate up front. By the time the reader reaches the technical section, they should know to read the model as a neat way to synthesise (1) and (2).

-- The paper would benefit from a Results section because there are important results to convey: A, B, and C.

- Put main results first; that's how readers interpret what's important!

- Result A is hidden in Figure 6A. This plot shows that your model works as advertised: a model of cultural evolution where a shared social goal is well-defined. To make this point cleanly, perhaps set g to zero?

- Result B is very nicely captured in Figure 4; it has one panel conveying B and two panels conveying B2. Perhaps one of those panels could convey B1, instead? It would be very compelling to show neatly that the model both screens for talent and screens for proxy-orientedness, plus that these trade off with rising c . To make this point cleanly, perhaps set g to zero?

- Result C is where g comes in. This is Figure 5; it's excellent. It is worth noting more clearly in the text that, for a system in a "lock in" state, higher competition beyond a certain point produces an equilibrium with similar practices, unhappier agents, and less overall effort. Now that's interesting.

- Then move onto secondary results, transient model behaviour, sensitivity analysis, and robustness checks. This is where the alternative implementations go, also.

-- Keep the discussion and meta-commentary civil, please. The relevant gap in the literature on modelling cultural evolution is arguably just as far-reaching as the relevant gap in the literature on modelling economic behaviour. In your paper, each of these literatures fills the gap in the other.

Decision letter (RSOS-211030.R0)

Dear Dr Braganza

The Editors assigned to your paper RSOS-211030 "Proxycconomics, a theory and model of proxy-based competition and cultural evolution" have now received comments from reviewers and would like you to revise the paper in accordance with the reviewer comments and any comments from the Editors. Please note this decision does not guarantee eventual acceptance.

Please submit your revised manuscript and required files (see below) no later than 21 days from today's (ie 01-Oct-2021) date. Note: the ScholarOne system will 'lock' if submission of the revision is attempted 21 or more days after the deadline. If you do not think you will be able to meet this deadline please contact the editorial office immediately.

on behalf of Marta Kwiatkowska (Subject Editor)
openscience@royalsociety.org

Associate Editor Comments to Author:

Comments to the Author:

Thank you for re-submitting your paper. We've now received two referee reports.

Please ensure that you respond appropriately to their concerns when you submit your revised paper and that you include a point-by-point response to the referees. Please also ensure that you include a version of your manuscript with any tracked changes you've made.

We look forward to seeing your revised paper! Best wishes.

Reviewer comments to Author:

Reviewer: 3

Comments to the Author(s)

Thank you very much for the detailed responses to the comments. Now I fully understood that the results are robust to the selection algorithms, and additional comments on the signaling theory and runaway process made sense to me. Instead of providing a pseudo-code, the author provided the source codes of the model, which is very nice. However, I still believe that providing an overview of the whole model structure in the paper itself is informative to readers, which should make them much easier to understand the results and findings as well as the model definition itself.

Reviewer: 4

Comments to the Author(s)

Agent-based models are difficult to do well, and this paper presents a strikingly good one. The paper models a competitive social system by combining (1) evolutionary selection on rank according to some measure with (2) a multi-tasking utility function for the agents that defines this measure as an imperfect proxy of some shared social goal. This is a major contribution in that it synthesises top approaches from several disciplines into a coherent whole; the resulting model is simple and yet it explains a lot.

Partially in response to prior reviewer comments, the author has included in the paper two implementations for each of (1) and (2). Specifically, (1) a thresholded or a proportional selection pressure on agents according to rank by the proxy measure and (2) a prospect or linear utility to the agents of their own rank according to the proxy measure.

Here listed are the main findings of the paper, as I understood them, none of which depend on the choice for (1) or (2):

(A) better proxy measures produce outcomes better aligned with a shared social goal

This is a key finding given that a major contribution of the paper is to put forward a model of cultural evolution where a shared social goal is able to be defined in the first place. That a shared social goal both exists and motivates agents is well-referenced (lines 399-416 and Section 3.4). The notion that prior literature on cultural evolution gives unsatisfactory treatment to the known/stated/implied/emergent goals of competitive social systems is well-argued (Section 3.6 and scattered elsewhere).

(B) moderate levels of competition produce better outcomes with respect to the shared social goal

This arises out of the interaction between two emergent properties of the model:

- (B1) higher levels of competition better screen for talent (this is good)
 (B2) higher levels of competition better screen for proxy-orientation (this is bad)

These are key findings given that a major contribution of the paper is to put forward a model of signalling/screening that explicitly incorporates selection pressure. That evolutionary selection exists in competitive social systems is well-referenced (lines 116-127). The notion that prior literature on signalling/screening gives unsatisfactory treatment to mechanisms of selection is well-argued (lines 963-72 and scattered elsewhere).

(C) intrinsic motivation bounds evolution but also creates agents unhappy with "locked in" practices

The basic result that intrinsic motivation bounds evolution is borderline tautological. But the *mechanism* whereby a positive g leads to higher equilibrium θ is non-trivial and intriguingly realistic (Section 2.4.5). The notion that defining an intrinsic motivation both bounds the long-term evolution of the system AND makes agents self-aware enough to be discouraged by that equilibrium is really insightful!

Taken together, the findings A, B, and C make for a compelling argument that this paper's combined approach results in a model that is greater than the sum of its parts. The depth of the discussion that can be had by interpreting this model in the context of several real-world systems further supports the authors' meta-argument: that combining (1) and (2) produces something coherent enough to deserve its own name; proxyeconomics is a bold one.

HOWEVER. Agent-based modelling papers are also very difficult papers to write, and as a reader it took me *a lot of work* to distill A, B, and C from the current version of the paper. This was my job as a reviewer, which I take seriously. But it was a struggle: even though I am well within the target audience for this paper it took me several days to get through it and there was much I had to look up. Many readers might just put the paper down.

 My recommendation to the editor is *minor revisions*. Highly effective writing is not listed as a publication criteria on the Royal Society Open Science Guidelines for referees; nor should it be. This research is sound. The paper is fit to be published upon clarification and/or correction of the following:

-- On lines 337-9 it is noted that agents are selected in a random order to update their effort based on their "current" rank. Please clarify that "current" means rank-calculated-in-the-past-timestep. The update order within a timestep should not affect agent calculations (except in the literal sense of the random seed). It may be that this clarification is found on lines 484-6; that is too far away from where the concern arises for me to be confident that the update order doesn't matter for the model as a whole. Alternatively, provide these related technical details in the same section.

-- In the implementation of proportional selection pressure, the appropriate analog to intensity of competition (c) is the *slope* of the proportionality between rank and relative probability of removal. Higher slope gives a larger protective effect to higher ranks, i.e. rank is more important and competition is more intense. The number of agents selected each round is analogous to the absolute selection pressure, very very similar to in the thresholded implementation. This is not clear from the text.

-- If the result in Figure 4 (i.e. result B) requires prospect utility, do note that.

-- In discussing the implications of the model results for market economies (Section 3.9) please clearly state what is being considered a proxy for what. In this sub-section, specifically, we are discussing profit as an imperfect proxy for what, exactly? Perhaps refer back to Table 1.

-- By the authors' own admission in the text of the paper, several sub-sections in the Discussion are highly speculative. Please move the Discussion sub-sections that venture well beyond the present model into a Future Directions section (or something) so it is clear where the science-discussion ends and the meta-discussion begins. Contributing to the literatures on (1) and (2) is science-discussion. Relating specific modelling concepts across fields is science-discussion. Sweeping implications for entire fields of research (biology, economics, sociology, ethics, policy, etc.) is meta-discussion.

My recommendation to the author themselves is *major revisions*. This work is great, so there could be a tremendous payoff from presenting it more effectively. Specifically, re-ordering the paper and modularizing the writing would help it find a larger and more receptive audience. Taking the following steps would make it far easier, and thus far more likely, for the reader to grasp that combining (1) and (2) leads to results A, B, and C.

-- This model combines approaches that are considered standard in two different fields. Right now, the intro is legible only to those with a double background or enough humility to Google dozens of terms; this is a small audience. Most readers would benefit from concise explainers on (1) and (2), separately, where you collect together the known/expected/predicted results of such models, common implementations, and key references for those who are unfamiliar. Present the terms you repeatedly use in their usual context, first, so they avoid being jargon disconnected from their origin.

- An explainer on (1) in the intro would be a great place to put a summary of reference [1] that you can point back to later (e.g. lines 253, 395, 860-80). Note, especially, that their model and ones like it struggle to define a social goal! This is discussed in Section 3.6 as a curious side-note, even though it forms the central contribution of the paper to the literature on (1).

- An explainer on (2) in the intro would be a great place to put technical references on multi-tasking models that are currently scattered through the manuscript, and references from Section 3.4. Note, especially, that these models rely on a notion of "good" or "honest" signalling that this literature struggles to define. This is discussed in Section 3.1 as a curious side-note, even though it forms the central contribution of the paper to the literature on (2).

-- Once you've briefed the reader on (1) and (2), separately, they will be better able to follow a qualitative description of your model (and more likely to believe that the combined model is greater than the sum of its parts). Focus on how this paper combines them: defining a practice as a vector in a proxy-goal space that engages with both (1) and (2). This is a good place for Figure 1, which is very intuitive.

-- When you then go on to describe the model in technical mathematical detail, please just describe the model. Sentences defining core concepts should be early in the Introduction (e.g. "social goal" on line 399). Sentences describing what your model allows you to incorporate should be in the qualitative description (e.g. lines 423-6). Sentences that describe what emerges from the model should be in the Results section (e.g. lines 269-74 and 308-323). Sentences describing the ranges used should be in Implementation (e.g. line 469). Sentences describing what various parameters "reflect", could be "viewed as", or could be "thought of" belong in the Discussion (too many to list). Sentences supporting your meta-argument should be somewhere else. Just define your model! Move all this other stuff out of the technical section.

- Corollary to the above: "attempting to convey the motivation underlying each specific operationalization choice" (lines 325-6) does the paper no favors. The best defence is a good offence---motivate up front. By the time the reader reaches the technical section, they should know to read the model as a neat way to synthesise (1) and (2).

-- The paper would benefit from a Results section because there are important results to convey: A, B, and C.

- Put main results first; that's how readers interpret what's important!

- Result A is hidden in Figure 6A. This plot shows that your model works as advertised: a model of cultural evolution where a shared social goal is well-defined. To make this point cleanly, perhaps set g to zero?

- Result B is very nicely captured in Figure 4; it has one panel conveying B and two panels conveying B2. Perhaps one of those panels could convey B1, instead? It would be very compelling to show neatly that the model both screens for talent and screens for proxy-orientedness, plus that these trade off with rising c . To make this point cleanly, perhaps set g to zero?

- Result C is where g comes in. This is Figure 5; it's excellent. It is worth noting more clearly in the text that, for a system in a "lock in" state, higher competition beyond a certain point produces an equilibrium with similar practices, unhappier agents, and less overall effort. Now that's interesting.

- Then move onto secondary results, transient model behaviour, sensitivity analysis, and robustness checks. This is where the alternative implementations go, also.

-- Keep the discussion and meta-commentary civil, please. The relevant gap in the literature on modelling cultural evolution is arguably just as far-reaching as the relevant gap in the literature on modelling economic behaviour. In your paper, each of these literatures fills the gap in the other.

===PREPARING YOUR MANUSCRIPT===

If you have been asked to revise the written English in your submission as a condition of publication, you must do so, and you are expected to provide evidence that you have received language editing support. The journal would prefer that you use a professional language editing

service and provide a certificate of editing, but a signed letter from a colleague who is a native speaker of English is acceptable. Note the journal has arranged a number of discounts for authors using professional language editing services (<https://royalsociety.org/journals/authors/benefits/language-editing/>).

===PREPARING YOUR REVISION IN SCHOLARONE===

<https://royalsociety.org/journals/authors/author-guidelines/#supplementary-material> to include a suitable title and informative caption. An example of appropriate titling and captioning

may be found at https://figshare.com/articles/Table_S2_from_Is_there_a_trade-off_between_peak_performance_and_performance_breadth_across_temperatures_for_aerobic_sc_ope_in_teleost_fishes_/3843624.

Author's Response to Decision Letter for (RSOS-211030.R0)

See Appendix C.

RSOS-211030.R1

Review form: Reviewer 4

Is the manuscript scientifically sound in its present form?

Yes

Are the interpretations and conclusions justified by the results?

Yes

Is the language acceptable?

Yes

Do you have any ethical concerns with this paper?

No

Have you any concerns about statistical analyses in this paper?

No

Recommendation?

Accept as is

Comments to the Author(s)

The manuscript has greatly improved---well done! The organization of the paper now makes it crystal clear where the model ends and the interpretation begins, making the message that much stronger. The key concepts are explained before they are used and the model is motivated up front. The description of the model itself is technical and concise. The results are listed and much easier to follow. The contributions to cultural evolution and the modelling of economic behavior are both clear and so, hopefully, legible to researchers within the respective communities. The work itself is far-reaching and fascinating; a strikingly good agent-based model.

I am thrilled to recommend the paper for publication in its current form.

Some typos:

* closing parenthesis in line 183

- * genuing in line 314
- * highly x 2 in line 412
- * after this my printer refused to print the pages, so I was reading on the computer and less able to note typos. perhaps a note for the typesetting team?

Decision letter (RSOS-211030.R1)

Dear Dr Braganza,

It is a pleasure to accept your manuscript entitled "Proxycconomics, a theory and model of proxy-based competition and cultural evolution" in its current form for publication in Royal Society Open Science. The comments of the reviewer(s) who reviewed your manuscript are included at the foot of this letter.

Kind regards,
Royal Society Open Science Editorial Office

on behalf of Marta Kwiatkowska (Subject Editor)
openscience@royalsociety.org

Reviewer comments to Author:

Reviewer: 4

Comments to the Author(s)

The manuscript has greatly improved--well done! The organization of the paper now makes it crystal clear where the model ends and the interpretation begins, making the message that much stronger. The key concepts are explained before they are used and the model is motivated up front. The description of the model itself is technical and concise. The results are listed and much easier to follow. The contributions to cultural evolution and the modelling of economic behavior are both clear and so, hopefully, legible to researchers within the respective communities. The work itself is far-reaching and fascinating; a strikingly good agent-based model.

I am thrilled to recommend the paper for publication in its current form.

Some typos:

* closing parenthesis in line 183

* genuing in line 314

* highly x 2 in line 412

* after this my printer refused to print the pages, so I was reading on the computer and less able to note typos. perhaps a note for the typesetting team?

Appendix A

Dear Editor, dear Reviewers,

Thank you for the opportunity to revise my manuscript. In response to the reviewers, I have made substantial changes, trying to make the manuscript more clear and readable in general, but also addressing each individual point raised specifically. Most importantly, I have added/separated-out two new dedicated sections (§ 1.3 and 3.6) to introduce the concept of cultural evolution as used here and to discuss in detail the relation to signalling and runaway in this context. I have also made substantial modifications to section 3.1, which previously contained this general point, but was not clear and explicit enough.

In the following I respond point by point, referring to the line numbers in the manuscript with tracked changes.

Reviewer comments to Author:

Reviewer: 1

Comments to the Author(s)

This paper though interesting, was difficult to review. The thrust of the paper is that competition and cultural evolution often occur not on the actual trait of interest, but on a proxy for this trait - hence “proxyeconomics”. It presents a model to show how this might play out.

The paper draws on many disciplines to frame the model, but is patchy and somewhat haphazard in this. For example, the present work overlaps with signalling theory in both evolution and economics, which can lead to runaway selection (classic peacock’s tail or tattooing in cultural evolution) or green beard type phenomenon. But none of this is discussed.

Thank you for this comment. I was also intrigued by the theories of signalling in economics and biology when I first started exploring this subject, thinking they would prove crucial in fleshing out the theory. However, I was somewhat disappointed to find that they simply assume the signal is constrained to be honest. That is, in both cases, signals are conceived in reference to an overarching selection mechanism which makes sure that the signal cannot stray too far from the goal (which is assumed to be identical to the overarching selection mechanism).

For instance, a foundational assumption of economic signalling models is that a negative correlation exists between signal cost and talent (Connelly et al., 2011; Spence, 1973). While not made explicit, this seems to be based on the notion that a market or some other mechanism would eliminate dishonest signals, and thus they need not be studied. In biology, the situation is similar, though the constraint there seems more plausible, namely overarching genetic selection. In the present theory, by contrast, we believe it is not warranted to assume the ‘abstract societal goal’ is automatically enforced by an overarching, independent selection mechanism. Positing e.g. some kind of genetic fitness as the goal of academic competition seems implausible, both from an explanatory and a normative perspective. Similarly, the assumption that what markets enforce through selection by definition corresponds to the abstract societal goal is receiving increasing scrutiny from within economics (Mirowski and Nik-Khah, 2017). Instead, we posit that the competitive societal system (i.e. the proxy) is our best attempt at translating an abstract goal into a selection mechanism, and that this is then itself not automatically enforced through a higher level selection. This complex but crucial point is now argued more extensively in the manuscript (§1.3, 3.1, 3.6).

The present model thus diverges from signalling models in this basic assumption, so in the results there was not much overlap that could be discussed. Due to the length of the manuscript, furthermore, I did not previously elaborate on this (though the general point had been made in section 3.1). However, I now clarify the relation to signaling theories more explicitly. I have thus substantially modified the manuscript accordingly (e.g. lines 222ff, 284ff), adding two new sections explicitly devoted to cultural evolution (§ 1.3 & 3.6) and clarifying the terms throughout the manuscript.

More generally, I am painfully aware that drawing in so many disciplines limits the depth to which any individual discipline can be treated and in particular that it precludes the situation of the model in any conventional discipline. I now explicitly acknowledge this in line 27ff.

However, I hope the revised manuscript makes more clear that each discipline I cover contributes crucial aspects, and furthermore that it is essential to investigate both discrepancies and relations between disciplines. For instance, the fact that the foundational assumptions of different disciplines (e.g. agency in economic vs. no agency in evolutionary models) differ so starkly should cause concern if we are really interested in a substantive scientific question (e.g. can proxy-based competition lead to lock-in). Indeed, the families of models emerging within a particular literature generally share basic assumptions, and the instance of signalling models discussed above illustrates how in our case rejecting one of these basic assumptions renders our model quite separate. The same holds for the incongruity of the agency vs no agency assumptions mentioned above. I would therefore argue that drawing together accepted formalizations from different disciplines is a strength, even if it means the resulting model is difficult to relate to any of the individual, established disciplines.

While I can see how the impression of patchiness and superficiality may have arisen, I would disagree that this holds for the underlying theory. Instead, I believe the impression is a result of the difficulty in integrating vastly different disciplinary terminologies, methodologies and assumptions into a single comprehensible text. As mentioned in the cover letter, the theory was presented at numerous conferences, where I received feedback from a highly multidisciplinary audience (admittedly mostly economists, complexity scientists, psychologists, sociologists and philosophers and no evolutionary scientists). While this substantially improved and sharpened the key ideas, the criticisms raised almost invariably revealed themselves as a problems in transdisciplinary communication rather than substance. That said, I can only hope the manuscript will trigger more in-depth discussions both amongst and within specific disciplines.

Finally, I have revised the whole manuscript with respect to language and hope this effort made it more readily understandable for a broad audience.

To be most helpful, I've written my reactions to the paper as I read it:

The paper is about formally modeling Goodhart's Law (when a measure becomes a target, it's no longer a good measure) and the related Campbell's Law (that measures will be subject to corruption as people condition on the measure rather than the underlying trait). Table 1 is a nice illustration of this, though I'm less convinced by the politics and markets columns (that populism and lobbying is a good example in politics or markets are optimizing wellbeing).

I would maintain that proponents of the respective systems almost invariably invoke the purported goal as the purpose or justification of the system. However, the reviewer may be suggesting that in some of these systems, this 'purported goal' may be little more than a

superficial public justification. This is a valid view, and I have added this qualification to the table legend, referring to the respective discussion in § 3.1 and §3.6. To which degree a ‘public justification’ can become an actual goal is in fact the very topic of the theory, as noted at the end of § 3.1.

The paper then makes the case for selection on the proxy, which becomes the measure of “fitness” being optimized. To make the case that the proxy can be corrupted, the author points out that the proxy cannot contain all information in the target and that there are mechanisms that undermine the information content of the proxy. This section is mostly asserted (rather than argued) with some citations, but was largely unclear. It was also unclear why the proxy (which is really a signal) doesn’t eventually get recognized as a poor signal.

I have now substantially modified this paragraph to make a more convincing argument. For instance, I have added several examples to more clearly convey why proxy-measures tend never to be perfect, including a citation (Koretz, 2008, line 87) outlining general challenges of measurement. I have also pointed to the distinction to conventional signals in lines 102ff, which references the novel section (§ 3.6) explicitly discussing the relation to evolutionary signalling models. I have further substantially modified § 3.1 to more clearly convey the difference to economic signalling models and point to this in lines 222ff.

The final paragraph (p4 of 37) was highly abstract and more confusing. Proxyeconomics studies the “emergent information strengths and weaknesses” by investigating “time-courses and mechanisms of aggregation and use”. Then the proxy’s psychological impact and salience to neuroscientifically realistic individual decision-makers. All in all, I only had a vague sense for what the author was trying to achieve and the connections to other fields seemed superficial at best.

I have attempted to make this more clear. The paragraph was originally deliberately written to describe the concepts in the most general manner possible, but I can see how the lack of specific example might make it confusing. I have thus now added a few carefully placed examples, and hope this will help to make the message more concrete. In addition to the specific examples in the paragraph, I have also added a reference to table 1, to try to convey the generality of the statements. I hope by referring to the specific examples in table 1, the reader can see for themselves how every proxy is associated with a particular competitive mechanism (e.g. peer review for high impact journals) with particular strengths and weaknesses.

Sections 1.3 and 1.4 continued this trend and did a poor job of specifically situating this work in any literature.

see above

Section 2 introduces the model, which is agent-based. The explanation for the model was again confusing, so I focused on the specifics. As a small aside, it might be better to subscript the variable names than use two letters which may be misunderstood as the product of two variables. I have attempted to make the description of the model clearer throughout its introduction, presentation and discussion (§2, e.g. lines 283ff, 310ff, 431ff, 478ff, 521ff, 529ff, 624ff, 759ff, 836ff). I have also changed the variable names to avoid two-letter variables replacing g with γ , g s with g and p s with p . Thank you for this helpful suggestion.

The model sets up the deviation between the proxy and the goal as an angle between two vectors, which seems reasonable. θ is an evolving angle for effort between these two. Selection pressure is adjusted via two variables – fraction of losers and probability a loser dies. I'm not sure why these weren't combined. An attempt is also made to incorporate a psychological goal scale (g), though why this doesn't evolve was unclear, but it seems like some kind of individual decision to deviate from the norm.

The reason to separate the fraction of losers (c) and probability a loser dies (p) is to respect the different timescales of psychological and cultural-evolutionary mechanisms. Specifically, the agents' psychological response to c (adjust effort) is done continuously in every round to reflect the speed of individual agency. By contrast cultural evolution happens on a slower time scale, i.e. only when an agent who has been competing for a while is selected. Furthermore this allowed us to independently vary the cultural evolutionary pressure and the psychological incentivization, relating the model more clearly to previous models in which only one of the mechanisms exists. I have modified the appropriate section to make this clearer (lines 529ff).

The psychological goal scale (now g) needs to be included because there is no natural relation between the scale of prospect utility and the utility derived from the goal. For instance (Baker, 1992) elaborate on why this necessitates a normalization factor. While the prospect function is empirically justified, there is no good indication as to how goal valuation should be scaled. As such g simply allowed to explore a plausible range (where goal motivation is substantially smaller to substantially larger than competitive motivation).

Intuitively, g can be understood as a scaling of the neural value signal. E.g. a larger g means that the neural value signal of a particular goal performance is comparatively greater, leading to a more powerful incentive effect. As such it would be implausible to simply let it evolve, because this would lead to an unbounded expansion of total value gained by an agent, making agents just progressively increase effort throughout evolution. To be completely sure, I have verified that this happens in an appropriately modified model (where g mutates with standard deviation 0.1 during inheritance, not shown). I have added some explanation in lines 476ff, to make the role of g clearer.

That said, some mechanism to evolve the relation between g and prospect-value would of course be interesting, for instance by adding some kind of constraint (e.g. total value signal available in the brain). However, this constraint would have to be thoroughly thought out and justified, and probing potential implications for the sensitivity and robustness of various plausible implementations would no doubt substantially lengthen the manuscript. Given, the length of the manuscript as is, and the ambitious scope of the present model, (as acknowledged by the reviewer below), this seems too much.

Overall, like the paper in general, the model seemed to be trying to do too much at once, making it difficult to generalize the findings. Section 3 continued the trend of trying to connect the model to too much, but only doing so superficially. Unfortunately, I was not impressed and I recommend major revisions to more clearly explain the idea and connect it to specific existing models in any of the fields the author is trying to draw in. I don't want to be too harsh – I think there is something here regarding how proxies can deviate from the underlying goal, but I think the question itself that the model is trying to shed light on needs to be better articulated and then more clearly presented.

As mentioned above, I have revised the modeling sections in order to be more clear. For instance, I have added the distinction to previous signalling models at an early point (lines 283ff), as well as at the appropriate places in the results and discussion (lines 624ff, 8361ff), making clear that in our context the signal captures both proxy-orientation and talent (i.e. proxy performance reports an interaction between both). The manuscript with tracked changes shows the numerous additional modifications, made to more clearly explain and situate the present model. However, as explained above, the early divergence of foundational assumptions in the present model and other models (e.g. moral hazard models such as (Baker, 2002, 1992) or cultural evolutionary models (Rendell et al., 2011; Smaldino and McElreath, 2016)) meant there was actually very little specific overlap. The main feat of the present model is to provide a first attempt at integration, yielding something quite new. The central finding that the economic multitasking implementation can constrain the evolutionary force could not have been asked without such an integration, but also necessarily led to limited overlap with previous models.

Reviewer: 2

Comments to the Author(s)

Dear Dr. Braganza,

thank you for the opportunity to read this paper.

Before I give my comments, allow me to situate the position from which I speak. I am a social scientist mainly working on the social dynamics of academia and their epistemic consequences. As such, your paper and its arguments touch my research, which is also highly concerned with the social and epistemic effects of competition. However, my expertise is mainly in qualitative methods. Thus I lack the expertise to fully judge the methodological details of your model.

Overall, this paper was a pleasure to read. Though as mentioned I cannot judge the methodological details, I was impressed with the clarity with which you describe your model and its implications. This allowed me as a more generalist than specialist reader to follow your argument and also scrutinize the underlying assumptions of your model to a degree that is appropriate for a scientific publication.

I highly appreciate the ambition of your paper to propose a general model that can stimulate thought and discussion on similar dynamics in very different social contexts and systems. In relation to the literature I have expertise on, on proxies and metrics in science, both your assumptions and your findings resonate very well with what has been described in social science analysis. However, your arguments also have the potential to add to this literature, on at least two levels. First, particularly your arguments about the respective capacities of qualitative and quantitative approaches are highly novel and valid. Second, the level of your theorizing which allows to compare dynamics to related phenomena in other social systems is highly inspiring. That said, as a qualitative social scientist oriented towards the micro, I could of course easily produce a long list of points along which your model is not fully realistic e.g. in its application to science (e.g. talent to satisfy the proxy might be different from talent to meet the societal goal). But I recognize that it is the approach of every model to be reductive and to reduce to key

elements of the dynamics addressed. Further studies might build on this, focus on more specific empirical domains and work with even more refined assumptions.

Thank you for this comment. The reviewer is of course correct that the model is a dramatic simplification. I have added a sentence explicitly acknowledging that in particular a wealth of individual level factors, such as diverging talent for proxy and goal, could play a role in future models and research (lines 9427ff).

Some minor suggestions for improvement:

- For a social science readership, your paper would benefit from more clearly outlining its concept of cultural evolution earlier in the paper. As a reader, this was unclear to me for some time. And as there are very different concepts of cultural evolution, it is important to know what you mean precisely.

This point is well taken, and overlaps with reviewer one's comments. I have now added a dedicated section in the introduction (§ 1.3) to introduce the cultural evolutionary concepts, I later draw on. I also discuss some additional implications from cultural evolution theory in a novel section (§ 3.6).

- In figure 3, I found it confusing that while lines A-D have the same column logic relating to different levels of competition, E does not follow this logic. I would recommend either visually flagging this more or moving E to a separate Figure.

I have separated Fig3, E out into a separate Figure (new Fig. 4).

In my reading, your paper has high potential to spark new interdisciplinary collaborations and conversations of high theoretical relevance to very important contemporary social phenomena. Thank you again for the opportunity to read it.

Many thanks for the interest and enthusiasm. The theory and manuscript do lead to the interface of qualitative and quantitative sciences, and I have now also tried to make this more explicit (lines 431ff). While I realize this makes it exceptionally difficult to place the current work within a particular literature, I thank the reviewer for pointing out that the transdisciplinarity in itself has value.

References

- Baker G. 2002. Distortion and Risk in Optimal Incentive Contracts. *J Hum Resour* **37**:728–751.
- Baker GP. 1992. Incentive Contracts and Performance Measurement. *J Polit Econ* **100**:598–614. doi:10.1086/261831
- Connelly BL, Certo ST, Ireland RD, Reutzel CR. 2011. Signaling Theory: A Review and Assessment. *J Manage* **37**:39–67. doi:10.1177/0149206310388419
- Koretz DM. 2008. *Measuring up : what educational testing really tells us*. Harvard University Press.
- Mirowski P, Nik-Khah E. 2017. *The Knowledge We Have Lost in Information*. Oxford University Press. doi:10.1093/acprof:oso/9780190270056.001.0001
- Rendell L, Fogarty L, Laland KN. 2011. Runaway cultural niche construction. *Philos Trans R Soc B Biol Sci* **366**:823–835. doi:10.1098/rstb.2010.0256
- Smaldino PE, McElreath R. 2016. The natural selection of bad science. *R Soc Open Sci* **3**:160384. doi:10.1098/rsos.160384

Spence M. 1973. Job Market Signaling. *Q J Econ* **87**:355. doi:10.2307/1882010

Appendix B

Associate Editor Comments to Author (Professor Bart de Moor):

Comments to the Author:

Regrettably, one of the reviewers sought has found substantial concerns with the paper - given the journal's general policy of only permitting one round of major revision, we must decline the paper at this stage; however, if you wished to submit a re-worked version of the paper for consideration for publication, you would be most welcome to do so. If, however, you felt your efforts may be better rewarded elsewhere, this decision allows you this flexibility (though, as I'm sure you can imagine, we hope you'll resubmit to RSOS!).

Dear Editor,

Thank you for the invitation to resubmit a revised manuscript. I have responded in detail to the points raised by the reviewer, specifically as recommended by the editor. However, please note again that ‘a differentiation of the population into two groups: motivated and discouraged groups’ is **not the focus of the manuscript** and is e.g. not even mentioned in the abstract. Furthermore, it **does not depend on the evolutionary mechanism but rather the economic mechanism**, as apparently assumed by the reviewer. This detailed in the respective methods (2.3.6, 2.3.7) and results sections (2.4.1 & 2.4.2, e.g. line 536ff, also see point to point). What is explicitly stated as ‘main question’ and ‘most pertinent result’ within the abstract and throughout the manuscript (see abstract, l18; model synopsis, l308ff; or model conclusion, 928ff) is: *How might an economic incentivization mechanism and a cultural evolutionary mechanism interact?* The most pertinent *result was that the economic incentive “can constrain a cultural evolutionary pressure, which would otherwise enforce fully proxy-oriented practices”* (e.g. abstract, l18).

I have nevertheless added additional analysis showing the robustness of both the main result, and the secondary result focused on by the reviewer to replacing the original selection algorithm with fitness-proportionate selection as requested (novel supplementary Figs. S4 and S2, respectively). I have also made more clear and added additional supplementary analysis showing, that this secondary result is not a result of the evolutionary selection (novel supplementary Fig. S1). The main result (see above) was also robust to a model modification which eliminated this secondary result and implements the reviewer’s preferred evolutionary selection algorithm (novel supplementary Fig. S4).

Please note that the model code and code to reproduce all figures was provided as supplementary material at submission. I have now further uploaded the code to https://github.com/oliverbraganza/Proxyeconomics_original

Line numbers refer to the version with tracked changes.

Reviewer comments to Author:

Reviewer: 2

Comments to the Author(s)

Dear Dr. Braganza,

thank you for your detailed response and revisions. They address the points I have raised very well. Again, I highly appreciate the interdisciplinary quality and ambition of your paper, and would be very happy to see it in print in this present form.

Thank you for the valuable comments and criticisms. I highly appreciate the substantial effort and scientific rigor evidenced by the review of the present manuscript, which I realize was both uncommonly extensive and complex in its transdisciplinary ambition.

Reviewer: 3

Comments to the Author(s)

This paper proposed a model of proxy-based competition and cultural evolution, particularly focusing on a differentiation of the population into two groups: motivated and discouraged groups.

Before the detailed discussions and relationships with interdisciplinary topics, I doubt that this is merely due to a particular type of cultural selection algorithm adopted in this model. In the model, a fixed prop. of the (motivated) best agents are secured and survive, and the remaining (discouraged) agents have randomly chosen with a fixed probability and replaced with a copy (with a mutation) of a randomly chosen agent in the former group. This extreme selection algorithm (i.e., elitism of adaptive agents + neutral selection on others) appears to be a major reason why the population evolved to be two different types of agents.

However, surprisingly, there are no clear discussions on the validity of using this type of algorithm to discuss this kind of phenomenon. I believe that this is necessary to understand the significance of the findings. After that, the authors need to show why such a phenomenon obtained from such a strong assumption is significant in this context and not merely due to artifacts caused by this algorithm. In other words, if a fitness proportional and stochastic selection (i.e., so-called roulette wheel selection), used often in evolutionary computations, is adopted, how the results will change and what does it mean in this context?

Dear Reviewer,

Please note that the 'differentiation of the population into two groups: motivated and discouraged' is **not the focus of the present manuscript and does not depend on the evolutionary mechanism**. The main focus, which is explicitly introduced as the 'main goal' throughout abstract, introduction and results, is *combining an economic with an evolutionary mechanism (e.g. 110ff of the abstract, 1246ff, 1251ff, line numbers refer to the revised manuscript with tracked changes)*. The central finding, as explicitly stated in every single summary paragraph including the abstract, is that the economic mechanism (specifically an intrinsic incentive towards the goal) can bound the evolutionary pressure towards the proxy (line 17 of the abstract, 1308ff, 1764ff, 1928ff, described in sections 2.2, 2.5, 2.6). The differentiation of agents into motivated and discouraged groups was *not mentioned in the abstract*, and not suggested as main result anywhere in the manuscript. I have nevertheless added additional analysis showing the robustness of this secondary result to implementing fitness proportional selection (novel Supplementary Fig. S2). I additionally confirmed that the main result of the manuscript was robust to i)

the separation into motivated and discouraged agents and ii) fitness-proportional selection (see below and novel Supplementary Fig. S4).

Given the interest in the specific secondary finding of the separation into motivated and discouraged agents, I have made more clear, and added additional figures showing, that this is **not a result of the evolutionary selection**. I now additionally show that it occurs similarly with no evolutionary selection (novel Supplementary Fig. S1), or fitness-proportional evolutionary selection (novel Supplementary Fig. S2, described in line 745ff).

1. Effort does not evolve, but emerges from the economic utility maximization, i.e. active agent choices in every round. The results describing how this occurs are elaborated in detail in section 2.4.1 & 2.4.2 (e.g. l599ff, or indicated by the black and red arrows in Fig.3A.) and further discussed in 2.5.2 (the algorithms are described in sections 2.3.6 and 2.3.7). For motivation, i.e. effort expenditure, to be inherited and selected would be implausible, so *this was not modeled or suggested anywhere in the manuscript*. By contrast, the complex effect of effort choices on the evolutionary mechanism is described in section 2.4.3. The differentiation into motivated and discouraged agents thus does not arise from, or reflect an evolutionary mechanism. To make this more clear I have now included an additional supplementary Fig S1, in which selection pressure (p) is set to 0, fully disabling the evolutionary mechanism (described in l568ff, l745ff).
2. That being said, the economic mechanism does play out the reviewer's intuition that competitive intensity drives the separation into motivated and discouraged agents, and *this is explicitly reported (throughout sections 2.4.1, 2.4.2, e.g. line556ff) and displayed (Figs 3B, Fig5B)*. Specifically, for very low competition (Fig 3B, leftmost column) the separation is almost fully absent. However, the separation into motivated and discouraged agents itself, while an interesting secondary finding, did not drive the main result (see below).

The claim that our evolutionary algorithm per se, or it's interaction with the economic mechanism, is 'extreme' could nevertheless call into question our main conclusions and it is thus important to address this point in detail. Briefly (1), investigating the effect of varying the intensity of competition and selection was the main organizing principle of the figures and study and (2), the main result of the evolutionary mechanism reproduces previous findings, and (3) the main result of the manuscript is robust to removing effort bifurcation and using fitness proportionate selection.

1. I explore and explicitly report the effect of different intensities of competitive pressure (and the resulting different degrees of effort bifurcation, see Fig.3B, 5B) throughout the manuscript (e.g. l310, table2, l556ff, l688ff, Fig3D, Fig5D). Notably, decreasing competition not only leads to less separation between motivated and unmotivated agents, it also directly decreases the number of selection events (e.g. line 569). The consistent result is that higher competition increases the speed of corruption, but leaves the equilibrium corruption level unaffected (l688ff, l710, l766, Fig.5D). This specific effect on system dynamics, but not equilibrium, was further confirmed in extensive sensitivity and robustness analysis (section 2.4.6, Fig6C, D).
2. The main result of the evolutionary mechanism, i.e. the evolution of practices towards the proxy, is thus robust to decreasing the competitive selection pressure and thus also to decreasing the degree of separation into motivated and discouraged agents. This result is corroborated by, and

indeed merely reproduces, a previous purely evolutionary model of proxy based cultural evolution (3). The latter is the first citation of the manuscript and discussed throughout (cited 18 times in total).

3. The main result of the manuscript (that the economic mechanism can constrain the evolutionary mechanism), is similarly robust to the separation into motivated and discouraged agents. E.g. Fig. 5D shows that an equilibrium is reached independent of the degree of separation (seen in Fig.5B). I have now added an additional analysis in which the separation into motivated and discouraged agents is eliminated fully by linearizing the economic prospect function and with fitness proportional selection. The results show that the intrinsic economic incentive still constrains the evolution to fully proxy-oriented practices (novel supplementary Fig. S4). However, a number of interesting and empirically observed, meso-level phenomena (such as the hill-shaped effort to competition relation, compare Fig. 5E) disappear.

Finally, the reviewer appears to think that the implementation of competition via a salient competition threshold separating winners from losers is unjustified. As this is indeed a novelty of the current model, I feel it is important to respond to this point. In our view this implementation of competition i) has substantial empirical justification and ii) elegantly allows the integration of the economic and the evolutionary mechanism (as carefully argued in l257ff). Specifically, it is based on the most empirically validated utility function presently known (namely that of prospect theory) and in fact represents an even wider confluence of transdisciplinary research on decision making under risk, relating it to its evolutionary origin (1). Briefly, organisms from plants to humans display sigmoidal effort functions, arising from a salient 'survival-threshold' value. Furthermore, the observed arising bifurcation of effort is well supported by empirical evidence from the economic contest literature (2). I would argue this reflects a substantially greater justification (an empirical grounding on two levels) than competing implementations such as fitness-proportional selection. Nevertheless, the main results were robust to implementing a fitness-proportional selection algorithm as requested.

Another factor is the difference in the talents among agents. This appears to simply determine whether the agents will be fall into the motivated or discouraged groups, under this extreme selection pressure. This is not surprising to me.

Talent was added to the model, in order to afford the competitive economic mechanism the opportunity to fulfil one of its prime functions, namely the screening for talent (e.g. l233, 251ff, l465ff). The reviewer correctly observes that this roughly corresponds to the self-separation into motivated and discouraged agents (as stated in e.g. l603ff). Accordingly, our model captures this desirable economic function (e.g. l582ff), which is indeed not surprising but rather reassuring. However, Fig. 3B shows that this separation is conditional on competitiveness, i.e. it does not 'simply' determine motivation status, and is e.g. less pronounced for low competition, where almost no screening for talent occurs. Furthermore, the relation between talent and motivation is to some degree mediated by proxy-orientation as for instance described in line 614: "Simultaneously, some highly talented agents are 'losing' in proxy-competition", owing to the fact that talent determines the cost of effort, but actual effort also reflects the distinctive contributions of the intrinsic and the competitive incentive. Finally, to reiterate, the separation into

motivated and discouraged agents (as well as talent), both have nothing to do with the evolutionary selection pressure (only the practice angle θ is inherited, as stated in l493ff).

The description of the model is not clear and the important procedures are described verbally. Thus, I recommend the authors to describe the whole algorithm by using a pseudo-code. I also recommend them to use capital letters to represent a parameter (i.e., a fixed value through a trial), which is one of the standard ways to discriminate between the fixed parameters and changeable variables, as far as I know.

I have made every attempt to make the description of the model as clear as possible, and e.g. Reviewer 2 explicitly praised the clarity of description. However, it is of course correct that code (or pseudocode) is the clearest way to communicate algorithms. I had therefore supplied the full original code including both the model as well as the code to generate all the figures, as supplemental material. This code is i) carefully annotated to maximize comprehensibility and ii) written in python according to the maxim of maximum readability (e.g. variables are named by their actual names, wherever possible). Such python code is designed to be as readable as pseudocode, and indeed I have so far only received praise as to the clarity of the code. While I agree that generally a pseudocode description can be helpful (particularly if the reader does not want to take the time to inspect the actual model code, or this code is in a less comprehensible language), I believe that in this instance a pseudocode would add very little value over the actual code, but not provide the full transparency awarded by this full code. Please note that I have now further uploaded the code to https://github.com/oliverbraganza/Proxyeconomics_original

Since each parameter is systematically varied to explore the model and results (as summarized in table 2), I think switching between capital and lower case letters between Figures and Figure panels would be confusing.

In the results, I do not understand why the authors do not directly show the changes in γ and e , which are the actually evolving and changing variables. The authors need to show this and discuss what happened through the trial more clearly.

The actually evolving variable is θ (i.e. the practice angle), and I must assume the reviewer mistyped, as this is made extremely clear throughout the manuscript (e.g. 2.3.7). The population mean of θ [mean practice ($^\circ$)] is directly shown in every single results figure (Fig3D, 4B, 5D, 5E middle panel, as well as Fig6A-F and Fig7A-F). It is color-coded as a black line with grey SEM-area throughout the figures such that the reader would be able to instantaneously recognize it, even if having read only one Figure legend. Furthermore, the θ of individual agents is plotted as the angle of a vector in Figs. 3B and 5B. While the plotting of vector ends as indicating vector length and angle should be intuitive given basic trigonometry, I have nevertheless tried to make this type of display exceedingly clear by providing Fig. 2A-C.

γ , as described in section 2.3.2 is a system level constant, which does not evolve or change, determining the degree to which the proxy captures incorruptible information about the goal (l357ff). I thus assume the reviewer mistyped.

Effort (e), which the reviewer rightly notes is the second directly changing variable, is directly plotted in Fig3A and Fig5A as a color-coded trajectory over time for each agent in a population. The types of dynamics (namely motivation leading to increasing effort and discouragement leading to decreasing effort) are described in detail in section 2.4.1 and indicated with arrows in Fig3A. Furthermore the effort distribution for agents within a population is plotted as the vector lengths, of the vectors mentioned above in Fig3B and Fig5B. Figures 4C and 5E rightmost panel, further show the population means of the theta-effort vector.

Of course, the degree to which the combination of effort and practice orientation (theta) is ultimately of interest is in how they determine the contributions to the proxy and the societal goal. Thus a minority of Fig. panels (Fig3C, 4A, 5C) within the above cited Figures is dedicated to directly showing this rather than the variables theta and e .

In addition, while other reviewers pointed out the relationship between this model and signaling theory and sexual selection, I do not understand why this is related to them. No signal is used in the proposed model in that agents do not exchange information about others. Also, a runaway process never occurs because there is no sexual selection in the model.

This is an issue of the framing of concepts within individual literatures/disciplines (as clarified e.g. in l586ff, l805ff, l1152ff). I agree with the previous reviewers that the proxy can valuably be understood as a signal of the goal, even though the precise modeling definitions vary. Agents signal their goal-contribution via the proxy, leading to their competitive selection. This can be seen as a signaling towards an assumed regulator of principal as in the economic literature or towards a female as in the biological literature. At the same time the proxy is the actually visible information between agents, underlying competitive motivation. Agents can thus also be assumed to infer the other agent's goal-contribution from their proxy performances, making the proxy a signal between agents, though here the inferred information is not relevant to any other modeled mechanisms. It is thus as possible to focus on the similarities as on the differences between the proxy, as introduced here, and traditional biological signaling models, and this is discussed in the dedicated section 3.6.

A runaway process by the traditional sexual selection mechanism also does not occur here, as the reviewer rightly observes. However, the population of agents produces a similar runaway phenomenon because the proxy performance (trait) and the competitive threshold (preference) are associated by assumption. This is simply because agents are judged by relative rather than absolute proxy-performance, i.e. an increasing population-mean proxy performance will shift the goal post. To get the faculty position, an academic must be better than the competition and thus the required absolute number of publications (or similar) will depend on the competition. Finally, note the relation to runaway cultural niche construction, in which a similar positive feedback-loop occurs between a niche and a trait. As described in lines 1166ff, such a dynamic is plausible in the general context of the present question, if the implementation of competition were assumed to be done by present proxy-winners. Note that this is a hypothetical higher level dynamic, which was not modeled in the present manuscript.

Due to these points, I do not understand the significance of the model and results, and how the finding can be related to the interdisciplinary phenomena discussed in this paper, from the current manuscript.

I have responded in detail to each point in order to clarify the significance and results of the model. In particular I hope to have made more clear the distinction between the economic and the evolutionary mechanism, and the justification and consequences of each. I further provide several additional analyses to demonstrate the robustness of the main result as well as the result highlighted by the reviewer to fitness-proportionate selection, and hope this addresses the reviewer's main substantive concern.

Minor:

Line 455: "In a subset of models": What do you mean by "subset"?

The subset of models reported in Fig7A-C as part of the robustness analysis to our economic effort choice implementation. We now state this explicitly (l457ff).

Appendix C

Dear Editor, dear Reviewers,

Thank you very much for the opportunity to revise the manuscript. Given the scope of revisions (including much text restructuring according to the detailed suggestions of Reviewer 4) I have highlighted only major changes in the main pdf (ending with `_highlight.pdf`). Line numbers below will refer to this highlight.pdf. In addition the pdf with all changes tracked is also attached (ending on `_diff.pdf`).

Reviewer comments to Author:

Reviewer: 3

Comments to the Author(s)

Thank you very much for the detailed responses to the comments. Now I fully understood that the results are robust to the selection algorithms, and additional comments on the signaling theory and runaway process made sense to me. Instead of providing a pseudo-code, the author provided the source codes of the model, which is very nice. However, I still believe that providing an overview of the whole model structure in the paper itself is informative to readers, which should make them much easier to understand the results and findings as well as the model definition itself.

Thank you for this comment. I have now added a pseudocode section at the beginning of the model description, in order to communicate the overall structure of the model in a concise way (p13, referenced in line 476).

Reviewer: 4

Comments to the Author(s)

Agent-based models are difficult to do well, and this paper presents a strikingly good one. The paper models a competitive social system by combining (1) evolutionary selection on rank according to some measure with (2) a multi-tasking utility function for the agents that defines this measure as an imperfect proxy of some shared social goal. This is a major contribution in that it synthesises top approaches from several disciplines into a coherent whole; the resulting model is simple and yet it explains a lot.

Partially in response to prior reviewer comments, the author has included in the paper two implementations for each of (1) and (2). Specifically, (1) a thresholded or a proportional selection pressure on agents according to rank by the proxy measure and (2) a prospect or linear utility to the agents of their own rank according to the proxy measure.

Here listed are the main findings of the paper, as I understood them, none of which depend on the choice for (1) or (2):

(A) better proxy measures produce outcomes better aligned with a shared social goal

This is a key finding given that a major contribution of the paper is to put forward a model of cultural evolution where a shared social goal is able to be defined in the first place. That a shared social goal both exists and motivates agents is well-referenced (lines 399-416 and Section 3.4). The notion that prior literature on cultural evolution gives unsatisfactory treatment to the known/stated/implied/emergent goals of competitive social systems is well-argued (Section 3.6 and scattered elsewhere).

(B) moderate levels of competition produce better outcomes with respect to the shared social goal

This arises out of the interaction between two emergent properties of the model:

(B1) higher levels of competition better screen for talent (this is good)

(B2) higher levels of competition better screen for proxy-orientation (this is bad)

These are key findings given that a major contribution of the paper is to put forward a model of signalling/screening that explicitly incorporates selection pressure. That evolutionary selection exists in competitive social systems is well-referenced (lines 116-127). The notion that prior literature on signalling/screening gives unsatisfactory treatment to mechanisms of selection is well-argued (lines 963-72 and scattered elsewhere).

(C) intrinsic motivation bounds evolution but also creates agents unhappy with "locked in" practices

The basic result that intrinsic motivation bounds evolution is borderline tautological. But the *mechanism* whereby a positive g leads to higher equilibrium θ is non-trivial and intriguingly realistic (Section 2.4.5). The notion that defining an intrinsic motivation both bounds the long-term evolution of the system AND makes agents self-aware enough to be discouraged by that equilibrium is really insightful!

Taken together, the findings A, B, and C make for a compelling argument that this paper's combined approach results in a model that is greater than the sum of its parts. The depth of the discussion that can be had by interpreting this model in the context of several real-world systems further supports the authors' meta-argument: that combining (1) and (2) produces something coherent enough to deserve its own name; proxyeconomics is a bold one.

HOWEVER. Agent-based modelling papers are also very difficult papers to write, and as a reader it took me *a lot of work* to distill A, B, and C from the current version of the paper. This was my job as a reviewer, which I take seriously. But it was a struggle: even though I am well within the target audience for this paper it took me several days to get through it and there was much I had to look up. Many readers might just put the paper down.

First, I must thank the reviewer for putting in such substantial effort. It has indeed been extremely difficult to write this paper, in particular since many assumptions and terms of the different disciplines do not translate easily to each other and are often only implicit. The reviewer's detailed suggestions for major revisions to make the manuscript more digestible have been extremely helpful. I have attempted to follow both suggestions for minor and major revisions closely, working through the entire manuscript for clarity and reordering sections for a more modular structure (see diff.pdf for all tracked changes). Line numbers below will refer to highlight.pdf, in which only major changes/ revised sections are highlighted.

My recommendation to the editor is *minor revisions*. Highly effective writing is not listed as a publication criteria on the Royal Society Open Science Guidelines for referees; nor should it be. This research is sound. The paper is fit to be published upon clarification and/or correction of the following:

-- On lines 337-9 it is noted that agents are selected in a random order to update their effort based on their "current" rank. Please clarify that "current" means rank-calculated-in-the-past-timestep. The update order within a timestep should not affect agent calculations (except in the literal sense of the random seed). It may be that this clarification is found on lines 484-6; that is too far away from where the concern arises for me to be confident that the update order doesn't matter for the model as a whole. Alternatively, provide these related technical details in the same section.

Thank you for this comment. 'Current' in the indicated context indeed means at the time of the random draw. This seemed more realistic than taking the values from the previous time-step (modeling random noise and uncertainty) but does not lead to outcomes different from when using the previous time step. To clearly show this I have run additional analyses using the previous time-step (see RevFigs 1,2. below). Also note that in dynamic equilibrium, the overall distribution of proxy-values remains stable, i.e. the distribution in the previous time-step is statistically indistinguishable from the present time-step.

I have now made this fully clear in the manuscript in the same section (page 12, footnote 3), but given the length of the manuscript decided against including additional supplementary Figures. If the editors or reviewer think I should include them, I will be happy to do so.

-- In the implementation of proportional selection pressure, the appropriate analog to intensity of competition (c) is the *slope* of the proportionality between rank and relative probability of removal. Higher slope gives a larger protective effect to higher ranks, i.e. rank is more important and competition is more intense. The number of agents selected each round is analogous to the absolute selection pressure, very very similar to in the thresholded implementation. This is not clear from the text.

The slope of the proxy-rank function would indeed be an intuitive analogue to competition. However, this is not possible because the normalization necessary to calculate fitness-proportionate selection, removes any change in slope.

Also note that the present implementation, where competition determines the absolute frequency of selection events ($c \cdot p \cdot N$) achieves precisely what the reviewer asks for: The relative probability of an agent to be selected, i.e. the *slope*, is multiplied by c to determine the actual selection probability. Thus the slope between agents with different proxy performance is proportional to c .

I've now made clearer in the text that c codetermines the number of agents selected each round (line 527, 534). I have further stated in the results section (line 834ff), that 'as in the previous simulations the frequency of such selection events is proportional to competition' to make clear that this is the same as in the previous thresholded implementation.

-- If the result in Figure 4 (i.e. result B) requires prospect utility, do note that.

Since all figures except for Robustness analysis (Fig8A-C, S3 and S4) rely on prospect utility it seems most appropriate to state the use of it centrally, as is done in lines 507. Adding an explicit mention in exclusively Fig. 4 would no doubt raise questions about the other main Figs. (3 & 5-7). However, the reviewer likely wanted to draw attention to the fact that the hill-shaped competition-to-effort relation arises already simply due to prospect utility (though with a slightly altered optimum). This is an important point, and I have thus drawn more explicit attention to it in the main text (lines 701ff). Note that it is also explicitly stated in what is now line 838ff.

-- In discussing the implications of the model results for market economies (Section 3.9)

please clearly state what is being considered a proxy for what. In this sub-section, specifically, we are discussing profit as an imperfect proxy for what, exactly? Perhaps refer back to Table 1.

I now clearly state this in line 1525, also adding some citations.

-- By the authors' own admission in the text of the paper, several sub-sections in the Discussion are highly speculative. Please move the Discussion sub-sections that venture well beyond the present model into a Future Directions section (or something) so it is clear where the science-discussion ends and the meta-discussion begins. Contributing to the literatures on (1) and (2) is science-discussion. Relating specific modelling concepts across fields is science-discussion. Sweeping implications for entire fields of research (biology, economics, sociology, ethics, policy, etc.) is meta-discussion.

I have now restructured the previous section '3. Beyond the model' into two sections namely to '3. General discussion' and '4. Future directions'. As suggested, the general discussion section contains what is most directly pertinent to the disciplines invoked in the model, i.e. the discussions on complex systems, information-theory, and cultural evolution theory. Also as suggested, the sections on i) epistemology, ii) psychology, iii) ethics, iv) sociology, v) policy, vi) markets, and vii) behavioral economics, have been separated into the 'Future directions' section.

I have further added some more clarification in the introductory lines of this section (lines 1321ff), to make clear that some of these sections 'venture well beyond the direct implications of the agent-based model' but that they are intended primarily to convey the 'full potential scope and relevance' of a larger underlying theory.

My recommendation to the author themselves is *major revisions*. This work is great, so there could be a tremendous payoff from presenting it more effectively. Specifically, re-ordering the paper and modularizing the writing would help it find a larger and more receptive audience. Taking the following steps would make it far easier, and thus far more likely, for the reader to grasp that combining (1) and (2) leads to results A, B, and C.

Again thank you for the detailed suggestions. I have attempted follow them closely. E.g. I have modularized the writing in such a way as to allow readers to potentially skip to the sections most relevant to them (see below). This is now noted early on (lines 69ff). I have also added a salient overview section early on, to allow readers to more easily navigate the manuscript (lines 51ff), and revised the text in the entire manuscript according to the reviewers comments.

-- This model combines approaches that are considered standard in two different fields. Right now, the intro is legible only to those with a double background or enough humility to Google dozens of terms; this is a small audience. Most readers would benefit from concise explainers on (1) and (2), separately, where you collect together the known/expected/predicted results of such models, common implementations, and key references for those who are unfamiliar. Present the terms you repeatedly use in their usual context, first, so they avoid being jargon disconnected from their origin.

- An explainer on (1) in the intro would be a great place to put a summary of reference [1] that you can point back to later (e.g. lines 253, 395, 860-80). Note, especially, that their model and ones like it struggle to define a social goal! This is discussed in Section 3.6 as a curious side-note, even though it forms the central contribution of the paper to the literature on (1).

- An explainer on (2) in the intro would be a great place to put technical references on multi-tasking models that are currently scattered through the manuscript, and references from Section 3.4. Note, especially, that these models rely on a notion of "good" or "honest" signalling that this literature struggles to define. This is discussed in Section 3.1 as a curious side-note, even though it forms the central contribution of the paper to the literature on (2).

I have now added brief descriptions of (1) and (2) into the respective sections §1.3 and §1.4. (I would like to keep the general sections, as they have been added at the explicit request of previous reviewers). The issue of discipline specific jargon is of course intensely appreciated, and I have done my very best to address it.

In §1.3 I have attempted to make the overall text more clear and more stringently lead up to the unresolved problem of how goals are thought of in this literature (in particular lines 220-233). This leads up to the requested brief summary of reference [1] (lines 234-250). For clarity, I then added a paragraph detailing how the present model differs from this (lines 251-268).

Similarly, I have restructured and amended §1.4. First, I have attempted to make the section introducing Goodhart's law and its analogues more clear (line 278-307), and in particular more clearly separate it from the introduction of multitasking.

This literature and its central results are then introduced in line 308-324. Line 325-339 makes the important point that the 'organizational goals' typically considered in this literature and those considered here must be distinguished, pointing to the more in depth discussion of the issue in § 3.1.

-- Once you've briefed the reader on (1) and (2), separately, they will be better able to follow a qualitative description of your model (and more likely to believe that the combined model is greater than the sum of its parts). Focus on how this paper combines them: defining a practice as a vector in a proxy-goal space that engages with both (1) and (2). This is a good place for Figure 1, which is very intuitive.

As the general introduction (including the clarifications and references added at the request of previous reviewers) remains somewhat long, I have also included extremely brief descriptors of (1) and (2) in the dedicated model introduction (§2.1, lines 361-376). This modularization should allow readers, who might be most interested in the technical details of the model, to skip directly from §1.1 to §2.1.

This revised model introduction (§2.1) now motivates i) the practice space including Fig.1 and ii) how competition is operationalized in two dedicated subsections (2.1.1, 2.1.2, lines 392ff), as suggested.

-- When you then go on to describe the model in technical mathematical detail, please just describe the model. Sentences defining core concepts should be early in the Introduction (e.g. "social goal" on line 399). Sentences describing what your model allows you to incorporate should be in the qualitative description (e.g. lines 423-6). Sentences that describe what emerges from the model should be in the Results section (e.g. lines 269-74 and 308-323). Sentences describing the ranges used should be in Implementation (e.g. line 469). Sentences describing what various parameters "reflect", could be "viewed as", or could be "thought of" belong in the Discussion (too many to list). Sentences supporting your meta-argument should be somewhere else. Just define your model! Move all this other stuff out of the technical section.

- Corollary to the above: "attempting to convey the motivation underlying each specific operationalization choice" (lines 325-6) does the paper no favors. The best defence is a good offence---motivate up front. By the time the reader reaches the technical section, they should know to read the model as a neat way to synthesise (1) and (2).

I have substantially shortened the technical section, following the above suggestions. The concepts of proxy and goal have been moved to the introduction (§1.2.1, §1.2.2). The motivation of operational choices should now be clear from a revised §2.1. Other aspects have been distributed into the results, implementation and discussion section as suggested. In particular, the discussion of what parameters mean (e.g. gamma, g or m) are moved to a novel discussion section 2.4.4 (line 1016ff), where their interpretation is directly related to their potential role in practical mitigation.

-- The paper would benefit from a Results section because there are important results to convey: A, B, and C.

- Put main results first; that's how readers interpret what's important!

- Result A is hidden in Figure 6A. This plot shows that your model works as advertised: a model of cultural evolution where a shared social goal is well-defined. To make this point cleanly, perhaps set g to zero?

- Result B is very nicely captured in Figure 4; it has one panel conveying B and two panels conveying B2. Perhaps one of those panels could convey B1, instead? It would be very compelling to show neatly that the model both screens for talent and screens for proxy-orientedness, plus that these trade off with rising c. To make this point cleanly, perhaps set g to zero?

- Result C is where g comes in. This is Figure 5; it's excellent. It is worth noting more clearly in the text that, for a system in a "lock in" state, higher competition beyond a certain point produces an equilibrium with similar practices, unhappier agents, and less overall effort. Now that's interesting.

- Then move onto secondary results, transient model behaviour, sensitivity analysis, and robustness checks. This is where the alternative implementations go, also.

Again, thank you for these extremely helpful suggestions. I have substantially revised the model results section (§ 2.3, lines 561ff) attempting to more clearly convey the most important results.

Since the most important emergent results (A and C) build on the smaller mechanistic results (B), I found it difficult to begin with the former. To nevertheless convey the relative importance of respective points I now begin the results section with an overview of four main results (R1-4), where R1,2 correspond to the reviewers result B. This simultaneously conveys how they build on each other:

R1 (screening for talent and proxy + effort incentivization) describes the mechanistic foundation of the emergent phenomena (corresponding to B1,2 + effort incentivization)

R2 (optimal level of competition) emerges from R1 (corresponding to B)

R3 (better proxies and more intrinsic motivation produce better outcomes) emerges from R1,2 as implemented in the practice space (corresponding to A).

R4 (bounded proxy-evolution) emerges from R1-3 in the long run (corresponding to C).

In other words, the emergent results R2-4 are introduced in sequence as arising from R1.

- Result A is now prominently features as R3 (lines 729ff, with an additional figure Fig. 5). As suggested, to make the points in R1 to R3 more cleanly, I've reproduced Figs.

3,4 with model runs where $g=0$ (i.e. no intrinsic incentive). Fig.5 then introduces $g>0$ showing how it affects A.

- Screening for i) talent and ii) proxy orientation can clearly be seen from Fig. 3B. i) is visible from the interaction of the color code with the x-axis. ii) is visible from the y-axis distribution. I have now made this more clear in the text (§ 2.3.2, e.g. lines 659ff). As mentioned, g is now set to 0 for these figures (Fig. 3,4)
- I now more clearly describe the intriguing combination of results for lock-in in lines 779 and 1088, using the concise and clear formulation suggested by the reviewer. I've also added the analogous emphasis for the short term outcomes (line 725)
- As mentioned above, I've retained the structure, beginning with a detailed description of the short-term model behavior, for two practical and one conceptual reasons: First, it is most useful to describe the model mechanics such as effort dynamics (Fig. 3A), i.e. it clarifies how the model works. Second, the long term outcomes literally build on and extend short term outcomes. Third, societal systems are arguably typically far from equilibrium, such that it is conceivable at least as important to investigate emergent dynamic pressures as opposed to eventual equilibria. Fortunately, in the present case both show the interesting lock-in properties.

-- Keep the discussion and meta-commentary civil, please. The relevant gap in the literature on modelling cultural evolution is arguably just as far-reaching as the relevant gap in the literature on modelling economic behaviour. In your paper, each of these literatures fills the gap in the other.

I have revised and shortened potentially controversial sections pertaining in particular to the notion of economy-wide lock-in (§4.6, 4.7). Specifically, I clarify that the present approach is not pro- or anti-economics per se, by elaborating more clearly why the notion of proxy-economics actually bears out the standard economic assumption that markets are far superior mechanisms when compared to e.g. central planning (lines 1523ff). I've also added an important clarification early on, namely that 'the present argument is *not* that proxy measures should, or can, be avoided' but rather that 'it is precisely because they can frequently not be avoided, that the arising dangers merit special attention.' (lines 24ff)

RevFig 1, Data from new Figs. 3,4 (with $g=0$) were reproduced with a change in effort-choice computation: effort was computed by individual agents based on the proxy-performances of all other agents at the previous time step, rather than at the time of their random draw.

RevFig 2, Data from the old Figs. 3,4 (with $g=1$) were reproduced with a change in effort-choice computation: effort was computed by individual agents based on the proxy-performances of all other agents at the previous time step, rather than at the time of their random draw.

===PREPARING YOUR REVISION IN SCHOLARONE===

-- If you are requesting an article processing charge waiver, you must select the relevant

waiver option (if requesting a discretionary waiver, the form should have been uploaded at Step 3 'File upload' above).
